# Change detection in the primate auditory cortex through feedback of prediction error signals

Keitaro Obara[1,2], Teppei Ebina [1], Shin-Ichiro Terada [1], Takanori Uka [3], Misako Komatsu [4], Masafumi Takaji[4,5], Akiya Watakabe [4,5], Kenta Kobayashi [6], Yoshito Masamizu[2], Hiroaki Mizukami[7], Tetsuo Yamamori[4,5,8], Kiyoto Kasai[9,10] & Masanori Matsuzaki [1,2,10] ✉

Although cortical feedback signals are essential for modulating feedforward processing, no feedback error signal across hierarchical cortical areas has been reported. Here, we observed such a signal in the auditory cortex of awake common marmoset during an oddball paradigm to induce auditory duration mismatch negativity. Prediction errors to a deviant tone presentation were generated as offset calcium responses of layer 2/3 neurons in the rostral parabelt (RPB) of higher-order auditory cortex, while responses to non-deviant tones were strongly suppressed. Within several hundred milliseconds, the error signals propagated broadly into layer 1 of the primary auditory cortex (A1) and accumulated locally on top of incoming auditory signals. Blockade of RPB activity prevented deviance detection in A1. Optogenetic activation of RPB following tone presentation nonlinearly enhanced A1 tone response. Thus, the feedback error signal is critical for automatic detection of unpredicted stimuli in physiological auditory processing and may serve as backpropagation-like learning.

The ability to detect salient (or unpredicted) stimuli is crucial for allowing animals to escape from predators and to forage in the wild. This ability is thought to be strongly related to top-down (feedback) transmission of higher-order information through hierarchical cortical areas[1,2]. In the visual cortex, attention changes the gain and selectivity of visual responses[1]. Feedback connections from the higher-order visual cortex to the primary visual cortex amplify the activity associated with differentiating a figure from the background and are thought to be required for visual awareness[3–5]. In addition, in neural network learning theory, feedback signals play a critical role in propagating the error calculated in the upper layer to the lower layer (backpropagation), so that the synaptic weights can be adjusted to minimize the error[6]. However, it remains unknown whether biological error signals extracted in a higher-order area can be explicitly back-projected through the hierarchical cortical architecture.

To search for potential evidence of such error feedback, we focused on auditory mismatch negativity (MMN; abbreviations are listed in Supplementary Table 1), an event-related potential that represents the

[1]Department of Physiology, Graduate School of Medicine, The University of Tokyo, Tokyo 113-0033, Japan. [2]Brain Functional Dynamics Collaboration Laboratory, RIKEN Center for Brain Science, Saitama 351-0198, Japan. [3]Department of Integrative Physiology, Graduate School of Medicine, University of Yamanashi, Yamanashi 409-3898, Japan. [4]Laboratory for Molecular Analysis of Higher Brain Function, RIKEN Center for Brain Science, Saitama 351-0198, Japan. [5]Laboratory for Haptic Perception and Cognitive Physiology, RIKEN Center for Brain Science, Saitama 351-0198, Japan. [6]Section of Viral Vector Development, National Institute for Physiological Sciences, Aichi 444-8585, Japan. [7]Division of Genetic Therapeutics, Center for Molecular Medicine, Jichi Medical University, Tochigi 329-0498, Japan. [8]Central Institute of Experimental Animals, Kanagawa 210-0821, Japan. [9]Department of Neuropsychiatry, Graduate School of Medicine, The University of Tokyo, Tokyo 113-8655, Japan. [10]International Research Center for Neurointelligence (WPI-IRCN), The University of Tokyo Institutes for Advanced Study, Tokyo 113-0033, Japan. ✉e-mail: mzakim@m.u-tokyo.ac.jp

difference between "rare deviant stimuli" (with deviant frequency, duration, or intensity) and "repetitive standard stimuli". MMN includes two components: adaptation to frequently presented stimuli and deviance detection in response to infrequent stimuli[7–13]. The peak amplitude of MMN in the auditory cortex occurs from 150–250 ms after a deviant stimulus[7]. This slow response is assumed to reflect the time required for the error signal to be generated by cortical processing[7,10].

The error signal in MMN is important for human brain physiology because patients with schizophrenia show a selective impairment in the deviance detection component, but not the adaptation component, of duration MMN (dMMN)[13]. Meta-analyses revealed that a reduction in the amplitude of dMMN, particularly dMNN with a long deviant duration, shows a larger effect size than a reduction in the amplitude of frequency MMN[14–16]. Patients with schizophrenia show impairment in tone discrimination, and this impairment is related to a reduction in the amplitude of dMMN[17]. In an analogy to the top-down influences in the visual cortex[4], this impairment may be related to dysfunctional feedback of auditory error signals.

Human electroencephalogram, electrocorticography, and functional magnetic resonance imaging studies suggest that the primary

auditory cortex (A1) and the lateral superior temporal gyrus, which include the higher-order auditory cortex, are involved in generation of auditory MMN[8,10,11,18]. However, these areas might be mainly related to generation of the adaptation component, and it is difficult to examine neuronal circuits at a single-neuron resolution in human studies. In non-human primates and rodents, many studies using electro-encephalogram, electrocorticography, and other electrical recording methods presented oddball paradigms with deviant tone frequency and showed MMN-like activity and deviance detection in the auditory cortex and auditory thalamus[19–24]. However, it remains unknown whether a pure deviance detection signal with minimal contamination of sensory signals is induced in auditory cortical neurons and is pro-pagated backward through the hierarchical auditory cortex. Further-more, if such a signal is back-projected, it is not known which sub-region and what type of neurons show it.

In this study, we examined A1 (core) and higher-order auditory cortex (lateral belt and parabelt) in the common marmoset (Fig. 1a), whose cytoarchitecture is similar to that of macaques and humans[25,26]. The core is surrounded by the belt, and the lateral belt is laterally surrounded by the parabelt (Fig. 1a). In macaques and humans, most of

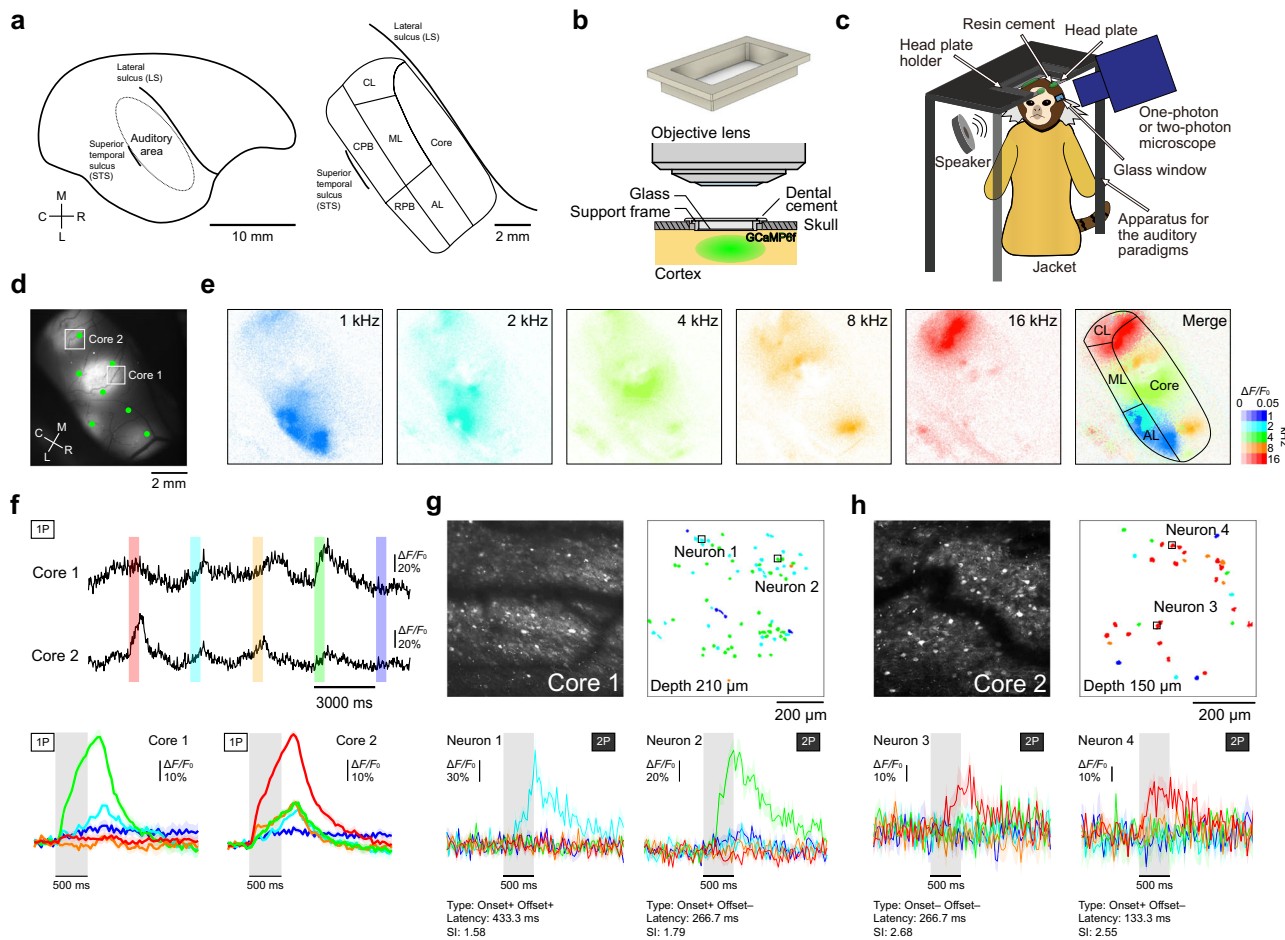

**Fig. 1 | Tonotopic mapping of the core in awake marmosets. a** Schematic right-side view of marmoset brain (left), and an enlarged view of the auditory cortex in the superior temporal gyrus (right). AL, anterolateral belt; ML, mediolateral belt; CL, caudolateral belt; RPB, rostral parabelt; CPB, caudal parabelt. **b** Schematic illustration of the imaging window frame (top) and imaging system (bottom). **c** Schematic illustration of the experiment. **d** Representative epi-fluorescence image of the core in the right hemisphere. Green dots indicate AAV-injected sites. Similar images were obtained in the total 25 sessions from three animals. **e** Calcium response maps for five pure-tone stimuli (1, 2, 4, 8, and 16 kHz). Rightmost image, the best frequency map. Black lines indicate the outer lines of an imaging window or contours of the sub-regions of the auditory cortex that were inferred from an

atlas. For each pixel of 256 × 256-pixel regions, the color of the corresponding BF and amplitude was assigned. **f** Top and middle, example calcium traces from two boxed regions shown in (**d**) (cores 1 and 2). Each color bar indicates the pre-sentation timing of the corresponding tone stimulus. Bottom, trial-averaged response to each tone in cores 1 and 2. Responses are aligned to the tone onset. Shading on each line indicates SEM. **g**, **h** Top left, two-photon images of cores 1 (**g**) and 2 (**h**) shown in (**d**). Top right, maps of BF of tone-responsive neurons in cores 1 (**g**) and 2 (**h**). Bottom, trial-averaged responses of four neurons labeled in the top images. Responses are aligned to the tone onset. The type, latency, and selectivity index of the sound frequency (SI) of each neuron are also shown (see Supple-mentary Table 2). Shading on each line indicates SEM.

the core is buried in the lateral sulcus, but this is not the case in marmosets[8,25,26], allowing the activity of the core, lateral belt, and parabelt to be imaged by fluorescence microscopy through a cranial window[27–30]. We therefore conducted one-photon and two-photon calcium imaging of the core, lateral belt, and parabelt of marmosets during a dMMN paradigm.

## Results

### One-photon and two-photon imaging of pure-tone responses in the core area in awake marmosets

First, we conducted wide-field one-photon imaging to examine tone responses over the marmoset core area (Fig. 1a). Using adult marmosets, we injected adeno-associated viruses (AAVs) carrying a tetracycline-inducible GCaMP6f expression system[31] into the core area and placed a wide-field (9 mm × 5 mm or 10 mm × 6 mm) cranial window (Fig. 1b). GCaMP expression became apparent after 4 to 6 weeks, and we then imaged the core during presentation of 500-ms pure tones at five different frequencies (1, 2, 4, 8, and 16 kHz) with the animals in an awake head-fixed condition (pure-tone paradigm; Fig. 1c, d and Supplementary Fig. 1a). When the best tone frequency (BF) was defined as the frequency with the maximum calcium response amplitude averaged over 1.5 s after the tone onset, each local region had similar BF, with the BF changing from low frequency to high frequency along a rostral-to-caudal axis (Fig. 1e, f). Similar maps were obtained across animals (Supplementary Fig. 1b). These results are consistent with the tonotopic organization of the marmoset auditory core area that was previously detected by extracellular recording[32], intrinsic signal imaging[29], voltage-signal imaging[27], and calcium imaging in anesthetized marmosets[30].

Next, we conducted two-photon calcium imaging of layer 2/3 (L2/3) neurons in the core area to determine whether the BF of individual neurons corresponded with the tonotopic organization. Many tone-responsive neurons in L2/3 had a sharp tuning property and their BF was consistent with the tonotopic organization (Fig. 1g, h). Furthermore, the BFs of neurons within the same local region at different imaging depths (150–350 μm) were also similar (Supplementary Fig. 1c). This result suggests columnar organization of tone frequency responses in the marmoset core area, and is consistent with that of a recent study[30]. Thus, we conclude that our calcium imaging method can detect neuronal tone responses in the marmoset auditory cortex.

### One-photon imaging and LFP recording of deviance detection in the core area

We next conducted one-photon calcium imaging of the core area to determine whether deviance detection occurred in an oddball paradigm in which the duration of repetitive tones of the same frequency was 50 ms, but the duration of a deviant (10% of tones) tone at the same frequency was 100 ms. The stimulus onset asynchrony (SOA) was 550 ms (Fig. 2a). Using this paradigm, dMMN was previously observed in both humans and macaques[33]. We also used the many-standards control paradigm[13], in which tones with 10 different durations (10–225 ms) at the same frequency are presented in a random order (Fig. 2a). We defined the difference in calcium responses between the deviant tone in the oddball paradigm and the 100-ms duration tone in the many-standards paradigm as deviance detection (see Methods for details). We refer to this paradigm, consisting of the oddball paradigm and the many-standards paradigm at the same tone frequency, as the dMMN paradigm.

The calcium response to the 100-ms stimulus in the oddball paradigm was larger than that to the 50-ms stimulus in the oddball paradigm (Fig. 2b) and the 100-ms stimulus in the many-standards paradigm (Fig. 2b–e and Supplementary Fig. 2a–c). Adaptation to the repetitive 50-ms stimuli after the deviant tone in the oddball paradigm was not apparent, and the amplitudes of the calcium responses to the 50-ms stimulus in the trial immediately before the deviant tone were

similar to those to the 50-ms stimulus in the many-standards paradigm (Supplementary Fig. 2d, e). This suggests that there was no apparent adaptation component, such as that defined by Koshiyama et al.[13] as a difference in response to the 50-ms tone between the oddball and many-standards paradigms. The deviance detection became apparent approximately 150 ms after the deviant tone onset in a part of the core area whose BF was the tone frequency used for the oddball paradigm ($F_{odd}$). The deviance detection was mainly observed in the middle region of the core when $F_{odd}$ was 2 or 4 kHz, whereas it was observed in the caudal region when $F_{odd}$ was 16 kHz (Fig. 2c and Supplementary Fig. 2a). The responding area at the time the peak amplitude of deviance detection was shown was similar to that at the time the peak amplitude to the 100-ms tone was shown in the many-standards paradigm, and the amplitude of the deviance detection gradually decreased until 500 ms after the tone onset (Fig. 2c, d and Supplementary Fig. 2a, b). These results suggest that the spatial distribution of deviance detection in the core depends on the tonotopic organization: deviance detection mainly occurs in the region where the BF is $F_{odd}$. The deviance detection amplitude (averaged over 200–400 ms after the tone onset) and the amplitude of the 100-ms tone response (averaged over 0–200 ms after the tone onset) in the many-standards paradigm did not differ among the five tone frequencies (Supplementary Fig. 2f, g).

We also inserted a microelectrode into the superficial layer (<300 μm from the cortical surface) to record the local field potential (LFP) in a site where the BF determined by one-photon imaging was $F_{odd}$. The deviance detection was significant around a time window (200–400 ms after the tone onset) similar to that of the calcium imaging (Fig. 2f). As in the calcium imaging, the amplitude of the LFP response (0–200 ms after the tone onset) to the 50-ms stimulus did not differ between the oddball and many-standards paradigms and did not show adaptation after the deviant tone in the oddball paradigm (Fig. 2g, h). Taken together with the fact that calcium imaging can also detect the deviant response to visual deviant stimuli that is detected by electrical responses[34,35], we consider that the calcium imaging-detected deviance detection and electrically-recorded dMMN reflect very similar neuronal deviant responses.

However, in almost all MMN studies, sensory-specific adaptation was observed in A1[20,23,24,36]. Thus, we further examined the cause of this discrepancy. We introduced repetitive (10 times) 50-ms standard tones with SOA of 550 ms (a standard-tone block). We repeated standard-tone blocks with an inter-block interval of 10 s (standard paradigm) while conducting LFP recording. If the response was suppressed in the first few trials of each session of the oddball and many-standards paradigms and a low steady-state response remained in the subsequent trials, the response to the first tone in the standard-tone block would be higher than the responses to the following tones in the same block. However, such attenuation was not prominent (Fig. 2i). Then, to examine whether the SOA of 550 ms was too long to induce adaptation in the marmoset core, we shortened the SOA to 350 ms in the standard paradigm. This change clearly induced response attenuation after the second tone presentations (Fig. 2i). Thus, the adaptation time course might be slower in the marmoset core than in other examined species[36,37]. This might be due to some different properties of short-term synaptic depression and suppression of excitatory neurons by inhibitory neurons[20,34].

The amplitude of dMMN is defined as the response to the deviant (100 ms) tone minus the response to the standard (50 ms) tone in the oddball paradigm. dMMN consists of the deviance detection component, tone difference component (the response to the 100-ms tone in the many-standards paradigm minus the response to the 50-ms tone in the many-standards paradigm), and adaptation component[13] (Supplementary Fig. 2h). In the core dMMN, the deviance detection component was similar to the tone difference component (Supplementary Fig. 2i).

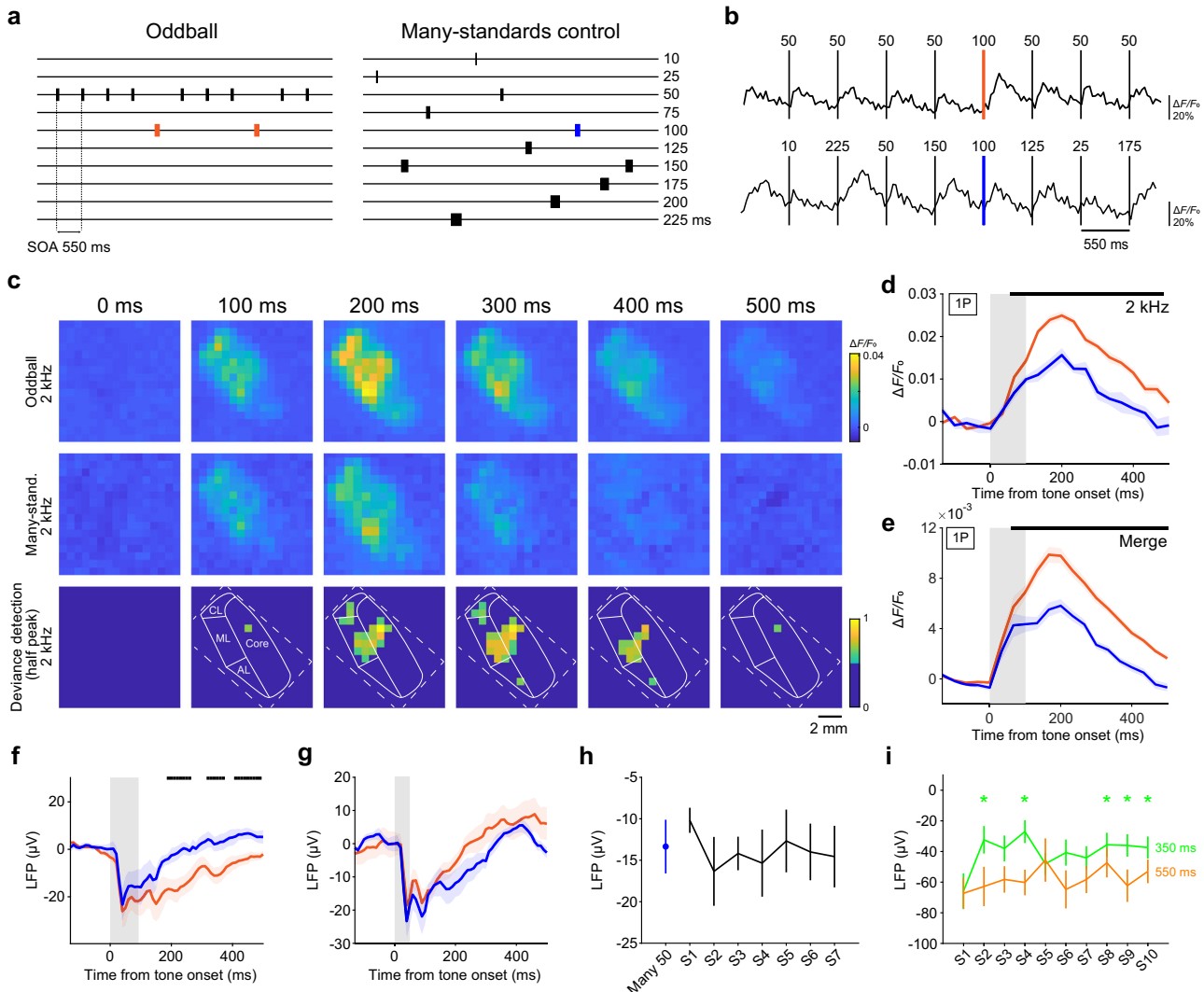

**Fig. 2 | One-photon imaging and LFP recording of deviance detection in the core area. a** Time sequence of the oddball paradigm and many-standards control paradigm in the dMMN paradigm. **b** Representative calcium traces of a core region in the oddball (top) and many-standards (bottom) paradigms. **c** Representative trial-averaged response maps of the core and belt in oddball (top) and many-standards (middle) paradigms with a 2 kHz tone (upper and middle row) and corresponding deviance detection (bottom). The pixel size was down-sampled to 16 × 16 pixels. In the deviance detection maps, pixels with less than half of the maximum response are set to the same deep blue color. Broken white lines indicate outer lines of the imaging window. Solid white lines indicate the putative contours of the sub-regions of the right auditory cortex. **d** Trial-averaged time course of the responses of deviant tone-responsive core regions to the deviant 2 kHz tone (100 ms; orange, $n = 29$ trials) and the 100 ms 2 kHz tone in the many-standards paradigms (blue, $n = 8$ trials) that were shown in (**c**). Shading on each line indicates SEM. The black lines indicate the period in which deviance detection was significant ($P < 0.05$, Wilcoxon rank sum test, two-sided, false discovery rate (FDR)-adjusted). **e** Session-averaged time courses of calcium responses to the 100-ms tone in the oddball (orange) and many-standards (blue) paradigms in the deviant tone-responsive core regions. Black lines indicate the timepoints at which the response amplitude was significantly different ($P < 0.05$, Wilcoxon rank sum test, two-sided,

FDR-adjusted; $n = 25$; 3 for 1 kHz, 11 for 2 kHz, 4 for 4 kHz, 4 for 8 kHz, and 3 for 16 kHz, from three animals in right image). Shading on each line indicates SEM. **f, g** Averaged LFP responses from the superficial layer in the core to the 100-ms (**f**) and 50-ms (**g**) tone in the oddball (orange) and many-standards (blue) paradigms ($n = 13$ penetrations from three animals). In (**g**), only the tone stimulus immediately before the deviant tone stimulus was chosen in the oddball paradigm. The averaged amplitude during 0–200 ms or 200–400 ms after the tone onset was not significantly different (0–200 ms, $P = 0.38$; 200–400 ms, $P = 0.23$, paired $t$-test, two-sided). Shading on each line indicates SEM. **h** The left-most blue dot shows the averaged LFP amplitude to the 50-ms tone in the many-standards paradigm in the core over 0–200 ms after the tone onset. The other points represent the averaged LFP amplitudes in the core to the $i$th 50-ms tone (labeled by S$i$, $i$ = 1, 2, to 7) after the deviant tone stimulus in the oddball paradigm ($n = 13$ penetrations from the same three animals as in **f**). The baseline amplitude was subtracted, and vertical lines indicate SEMs. **i** Averaged LFP amplitude (during 0–200 ms after the tone onset) in the standard paradigm with SOA of 550 ms (orange) or 350 ms (green). $n = 18$ and 16 sessions, respectively, from two animals. SOA 550 ms: $n = 2$ at 1 kHz, 3 at 2 kHz, 3 at 4 kHz, 6 at 8 kHz, and 4 at 16 kHz. SOA 350 ms: $n = 2$ at 1 kHz, 3 at 2 kHz, 2 at 4 kHz, 5 at 8 kHz, and 4 at 16 kHz. *$P < 0.05$, the first tone vs. the second and subsequent tones (Wilcoxon signed-rank test, two-sided, FDR-adjusted).

## Pure-tone responses in the lateral belt and parabelt neurons

Next, we conducted wide-field one-photon imaging of the lateral belt and parabelt. To reduce contamination of the activity of the core neurons, we injected the AAVs into multiple sites around the border between the lateral belt and parabelt, and into the parabelt itself (Fig. 3a). We then examined the calcium responses from fluorescent

regions: the middle lateral (ML) and caudolateral (CL) belt, rostral parabelt (RPB), and caudal parabelt (CPB). All regions appeared to respond to all tone frequencies to some extent, and the presence of tonotopic organization was unclear, although the response to the 16-kHz tone in the anterior part of the RPB was relatively strong (Fig. 3b and Supplementary Fig. 3a).

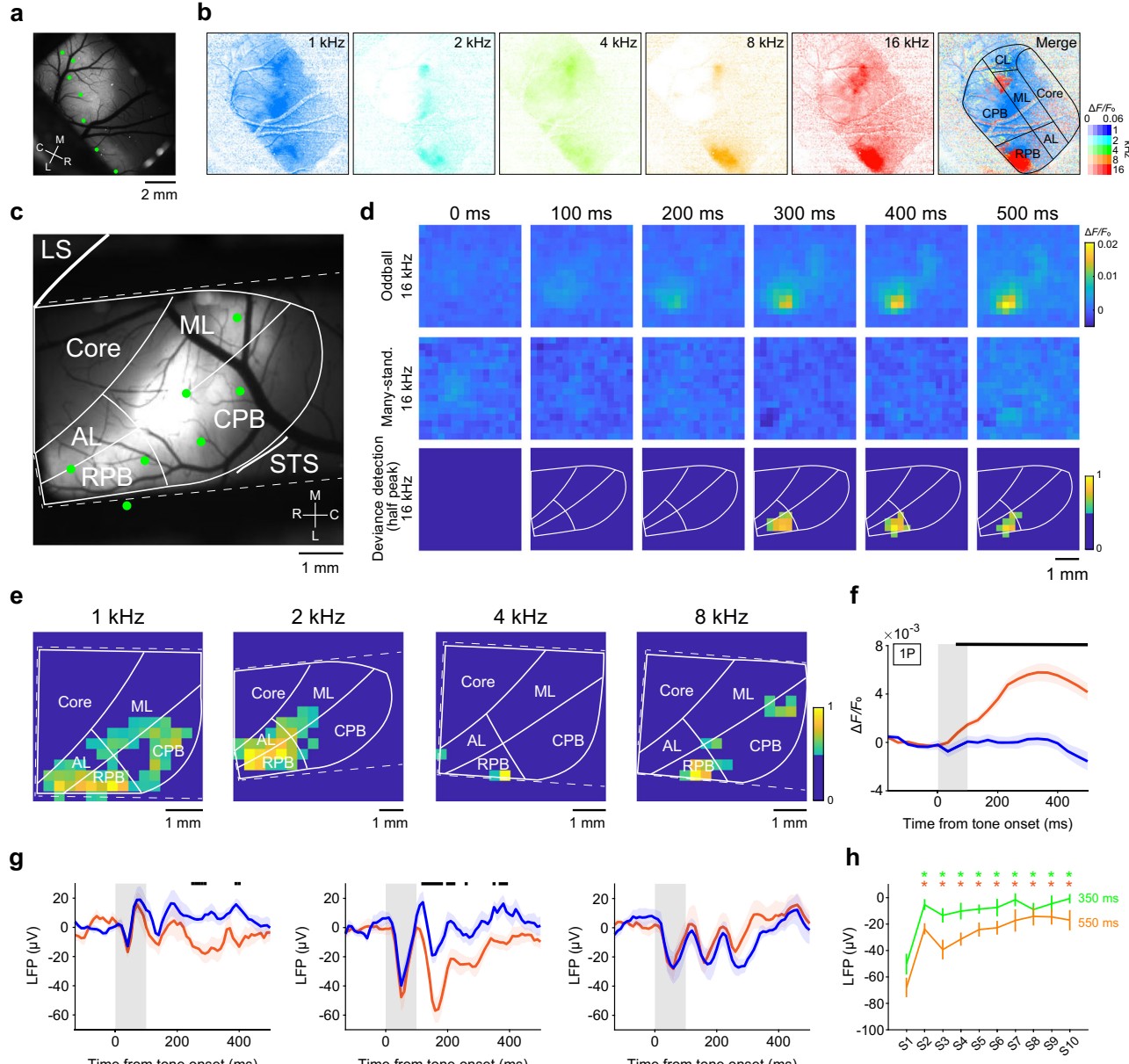

**Fig. 3 | Pure error signals in higher auditory RPB. a** Representative epi-fluorescence image of the lateral belt and parabelt in the right hemisphere. Green dots indicate AAV-injected sites. Similar images were obtained in the total 25 sessions from two animals. **b** Response maps for five pure tones (1, 2, 4, 8, and 16 kHz). Rightmost image, the best frequency map. For each pixel of 256 × 256-pixel regions, the color of the corresponding BF and amplitude was assigned. Black lines indicate the outer lines of an imaging window or putative contours of the sub-regions of the auditory cortex. **c** Representative epi-fluorescence image of the left auditory cortex including the lateral belt and parabelt. Green dots indicate AAV-injection sites. White lines indicate the outer lines of the imaging window or putative contours of the auditory sub-regions. Similar images were obtained in the total 25 sessions from two animals. **d** Representative trial-averaged response maps of the lateral belt and parabelt in oddball (top) and many-standards (middle) paradigms with a 16 kHz tone (upper and middle row) and corresponding deviance detection (bottom). The pixel size was down-sampled to 16 × 16 pixels. In the deviance detection map, the pixels with less than half of the maximum response are set to the same deep blue color. White lines indicate the outer lines of an imaging window or putative

contours of the sub-regions of the auditory cortex. The animal was the same as in (**c**). **e** Representative trial-averaged deviance detection map at 400 ms after the tone onset for 1, 2, 4, and 8 kHz tone frequencies recorded on different days. The animal was the same as in (**c**). **f** Session-averaged time courses of calcium responses to the 100-ms tone in the oddball (orange) and many-standards (blue) paradigms in the deviant tone-responsive RPB regions (*n* = 25 sessions from two animals). Shading on each line indicates SEM. **g** Session-averaged time courses of LFP responses to the 100-ms tone in the oddball (orange) and many-standards (blue) paradigms that were recorded from the superficial layer in the dMMN paradigm in each sub-region (left; *n* = 9 penetrations [sessions] for ML/CL, middle; 9 for RPB, and right; 10 for CPB). The animal was the same as in (**c**). Shading on each line indicates SEM. **h** Averaged LFP amplitude (during 0–200 ms after the tone onset) in the standard paradigm with SOA of 550 ms (orange) and 350 ms (green). *P < 0.05, the first tone vs. the second and subsequent tones (Wilcoxon signed-rank test, two-sided, FDR-adjusted). The baseline amplitude was subtracted, and vertical lines indicate SEMs. For each SOA of 550 and 350 ms, *n* = 15 from two animals; *n* = 3 at 2 kHz, 4 at 4 kHz, 6 at 8 kHz, and 2 at 16 kHz.

When two-photon imaging was conducted, individual tone-responsive neurons showed the BF, and many of them showed sharp tone-tuned properties (Supplementary Fig. 3b–d). This is consistent with findings from electrical recording of the macaque parabelt[38]. The

width of the frequency tuning (selectivity index, SI) was similar among auditory cortical regions (Supplementary Table 2, Fig. 1g, h, and Supplementary Fig. 3b–d). As shown in Supplementary Fig. 3b–d (neurons 6, 8, and 10), a subpopulation of neurons showed responses after the

end of the BF tone presentation (offset responses). We found that 14–23% of tone-responsive neurons in the core, lateral belt, and parabelt showed such offset responses (offset+; Supplementary Table 2). Among offset+ neurons, 50–59% also showed responses during the tone presentation (onset+ offset+; Supplementary Table 2). These results are consistent with previous studies that performed electrical recording in A1 of macaques, marmosets, and guinea pigs[39–41].

In the imaging fields, the highest proportion of neurons had a BF of 8 kHz in the RPB (45.1%) and 1 kHz in the CPB (37.7%) (Supplementary Table 2). This difference was consistent with the result of one-photon imaging (Fig. 3b). Although the rostral-caudal gradient of high-to-low BF in the RPB and low-to-high BF in the CPB has been reported in electrical recording of macaques[38], we did not detect a low-to-high BF gradient in the CPB. The proportion of nearest neighbors having the same BF was slightly higher in the core and ML/CL than in RPB and CPB (core, 44.5% ± 2.1%, $n = 32$ fields; ML/CL, 45.1% ± 3.9%, $n = 14$; RPB, 37.7% ± 4.3%, $n = 11$; CPB, 42.5% ± 4.8%, $n = 11$), but not significantly so ($P > 0.05$, unpaired $t$-test with Bonferroni correction, two-sided). Thus, the reason why the tonotopy in the belt and parabelt was not clear in one-photon imaging might be because the responses from the neuropiles (axons and dendrites) with different BF were more intermingled in the belt and parabelt than in the core, rather than because of a local difference in the proportion of neurons that possessed the same BF. To clarify the tone frequency representation in more detail, it is necessary to record the BF of individual neurons in superficial and deep layers over a broad area including the belt and parabelt.

### One-photon imaging and LFP recording of deviance detection in the lateral belt and parabelt

Next, we conducted one-photon imaging of the lateral belt and parabelt to determine whether these areas showed deviance detection. In these areas, responses to the control stimulus were very weak, whereas responses to the deviant stimulus were clearly detected (Fig. 3c–f). Deviance detection 200–400 ms after the tone onset was strongly induced in the RPB, irrespective of $F_{odd}$ (Fig. 3d, e and Supplementary Fig. 3e). Averaged over sessions, the RPB and anterolateral belt (AL) showed strong deviance detection in one marmoset, while the RPB and ML/CL showed strong deviance detection in the other marmoset (Supplementary Fig. 3f). Thus, the RPB showed strong deviance detection in both animals. When the LFP was recorded in the superficial layer, significant timepoints for deviance detection were detected in ML/CL after approximately 200 ms after the tone onset, and in RPB after approximately 100 ms after the tone onset, but not in CPB (Fig. 3g). These results suggest that RPB was the origin of the error signal in the auditory cortex.

In contrast to the LFP response in the core, the LFP response in the RPB rapidly attenuated in the standard paradigm with a tone duration of 50 ms and SOA of 550 ms (Fig. 3h). In addition, the response was much reduced during the many-standards control paradigm (Supplementary Fig. 3g). The response to the 50-ms tone was similarly weak in the oddball and many-standards paradigms in both LFP recording and calcium imaging, and the attenuated amplitude of the LFP response to the 50-ms stimulus did not recover immediately after the deviant tone nor did it then adapt again until the next deviant tone in the oddball paradigm (Supplementary Fig. 3h–j). These results suggest that the repetitive stimulation with the same tone frequency strongly attenuated the RPB response to a low steady-state level, regardless of the duration of sound presentation, and that no apparent adaptation component was observed in the RPB as well as in the core, at least under the present stimulus conditions. No apparent adaptation component in the core was likely due to no apparent adaptation in responses to the 50-ms tone in the oddball and many-standard control paradigms, while that in the RPB was likely due to similarly strong adaptation in responses to the 50-ms tone in both paradigms. In

dMMN in the RPB, the deviance detection component was much larger than the tone difference component (Supplementary Fig. 3k).

### Two-photon imaging of deviance detection in the core, lateral belt, and parabelt

Next, we conducted two-photon calcium imaging of L1 and L2/3 in the core, ML/CL, RPB, and CPB. If a L2/3 neuronal soma showed significantly high activity in at least one of the oddball and many-standards paradigms, it was defined as a dMMN-responsive neuron (see "Methods" for details). The proportion of dMMN-responsive neurons was approximately 60%; in the core it was 1694/2771 in 34 sessions from three animals, in the ML/CL it was 712/1176 in 16 sessions from two animals, in the RPB it was 516/791 in 11 sessions from two animals, and in the CPB it was 482/791 in 11 sessions from two animals. We separately analyzed calcium responses of neuropils in L1 and L2/3, and dMMN-responsive neurons in L2/3 (Fig. 4 and Supplementary Fig. 4). Deviance detection was clearly detected in both L1 and L2/3. The first timepoint with significant deviance detection was detected 100–200 ms from the tone onset in the neuropils, but was around 300 ms from the tone onset in the L2/3 neuronal somata. However, responses to the control stimulus were detected in both layers (Fig. 4a–e and Supplementary Fig. 4a). In ML/CL, deviance detection was detected in both L1 and L2/3, but in contrast to the core, responses to the control stimulus were weak in the neuropils (Fig. 4f). In RPB, significant deviance detection was detected in L2/3 neuropils and neuronal somata immediately after tone cessation. The response to the control stimulus after the tone end was weak, even in L2/3 neuronal somata (Fig. 4g and Supplementary Fig. 4b). In CPB, deviance detection was not detected in either layer (Fig. 4h). Thus, the deviance detection calcium response was strongly correlated with the deviance detection electrical response, and the pure error signal originated from the RPB neurons in these regions.

### Pure error signals in RPB neurons with offset responses

We classified dMMN-responsive neurons that were also identified in the pure-tone paradigm into those neurons whose BF was $F_{odd}$ and those whose BF was otherwise. Although the BF determined in the pure-tone paradigm was not necessarily the best frequency for the shorter tone stimulation, the deviance detection was larger in those neurons whose BF was $F_{odd}$ than in those whose BF was otherwise, and the deviance detection in the former neurons was larger in the RPB than in the core and ML/CL (Fig. 5a and Supplementary Fig. 5a, b).

The RPB neurons with offset responses showed larger deviance detection than those without offset responses, while some significant timepoints for the deviance detection were detected immediately after the tone end in the latter neurons (Fig. 5b, c and Supplementary Fig. 5c). The deviance detection immediately after the tone end (averaged over 0–67 ms) was significantly higher than zero in core and RPB neurons without offset responses, but not in those with offset responses (Fig. 5d). In contrast to the core and ML/CL neurons, the RPB neurons showed a deviance detection amplitude that was almost the same as that of calcium responses to the deviant stimulus because the response to the 100-ms tone in the many-standards control paradigm was subtle (Fig. 5b, c and Supplementary Fig. 5d, e). Thus, RPB neurons represented the pure error signal when the deviant tone was presented. The RPB neurons with offset responses but without onset responses showed the largest deviance detection (Supplementary Fig. 5f). In the core and ML/CL, neurons with both onset and offset responses showed the largest deviance detection (Supplementary Fig. 5f).

We further examined the properties of those neurons with offset responses whose BF was $F_{odd}$. The responses of these neurons to each tone in the many-standards paradigm (averaged over 100–300 ms after the tone end) correlated strongly with the tone duration in the core, but not in the other areas (Fig. 5e–h). RPB neurons with offset

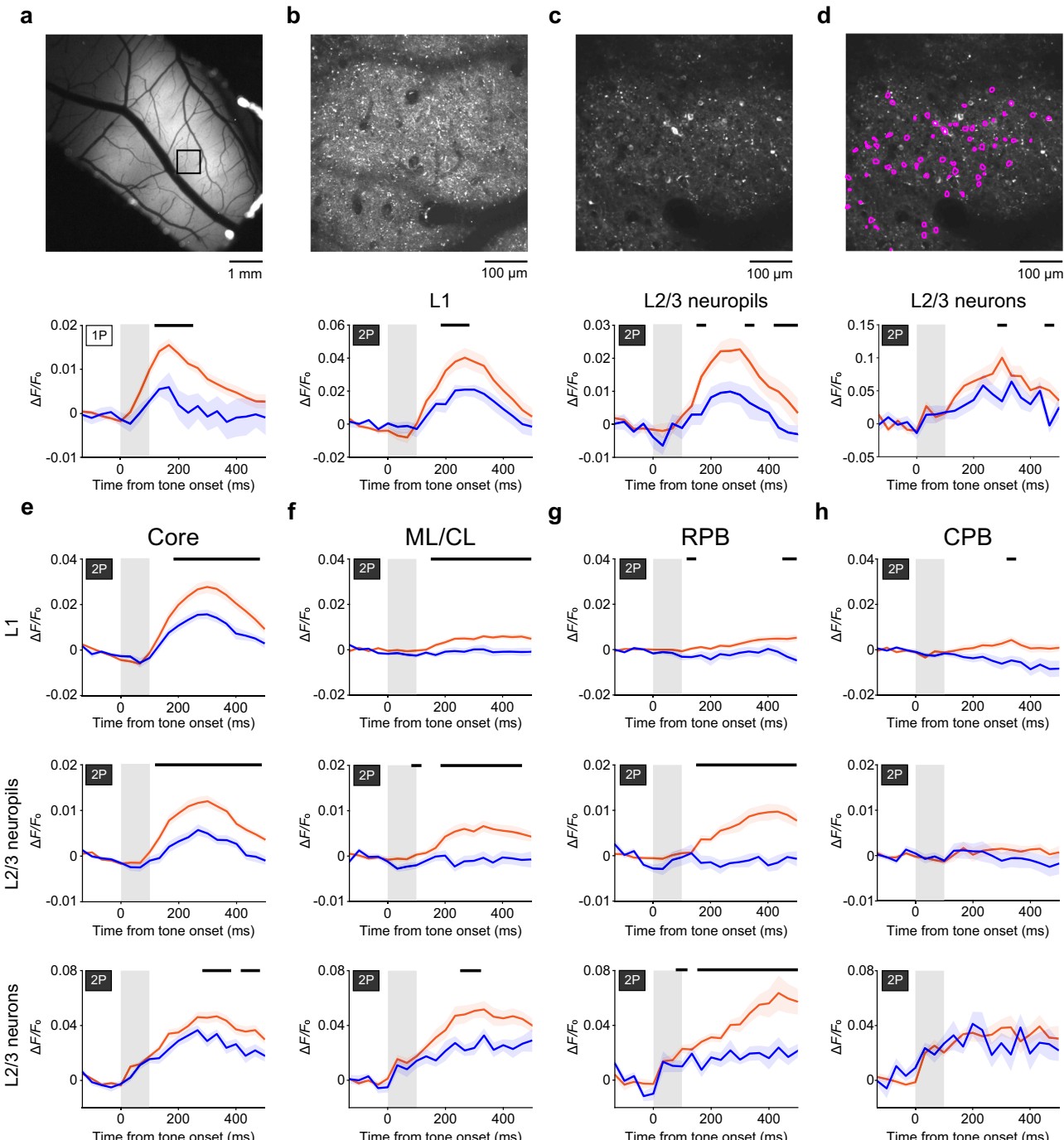

**Fig. 4 | Two-photon calcium imaging of deviance detection in the core, lateral belt, and parabelt. a** Top, representative epi-fluorescent image of the right auditory cortex including the core. Bottom, trial-averaged calcium responses of the boxed region in the top image. The tone frequency was 2 kHz. Shading on each line indicates SEM. $n = 31$ (orange) and 7 (blue) trials. **b–d** Top, two-photon images of the boxed region in A in L1 (**b**) and L2/3 (**c**, **d**). The tone frequency was 4 kHz. Shading indicates SEM. The number of trials with the 100-ms tone in the oddball and many-standards paradigms was 30 and 14, respectively (**b**), and 29 and 10, respectively (**c**, **d**). In (**d**), contours of 64 dMMN-responsive neurons are overlaid on the image. Bottom, trial-averaged calcium responses of the field in the top image (**b**), the field except for the regions corresponding to neuronal somata in the top

image (**c**), and the dMMN-responsive neurons in the top image (**d**). Orange, calcium responses to the deviant stimulus. Blue, calcium responses to the control stimulus. **e–h** Calcium responses in the core (**e**), ML/CL (**f**), RPB (**g**), and CPB (**h**). Black lines indicate the period in which deviance detection was significant ($P < 0.05$, Wilcoxon rank sum test, two-sided, FDR-adjusted). Shading on each line indicates SEM. In L1, $n = 21$ ROIs for the core from one animal; and $n = 18$ for ML/CL, 14 for RPB, and 11 for CPB from two animals. In L2/3 fields except for the regions corresponding to neuronal somata, $n = 34$ ROIs for the core from three animals; and $n = 16$ ROIs for ML/CL, 11 for RPB, and 11 for CPB from two animals. In L2/3 dMMN-responsive neuron-averaged responses, $n = 1694$ neurons for the core from three animals; and $n = 712$ for ML/CL, 516 for RPB, and 482 for CPB from two animals.

responses showed significant activity after the end of 10-, 200-, and 225-ms tone stimulations (Fig. 5g). In the ML/CL and CPB, such prominent responses were not apparent. In the RPB, the amplitude of the response to the longest (225 ms) tone in the many-standards control

paradigm was the highest in the neurons with offset responses whose BF was $F_{odd}$ (Supplementary Fig. 5g), as in the case of the response to the 100-ms tone in the oddball paradigm. The amplitude of the deviance detection and SI were negatively correlated in RPB, but not in

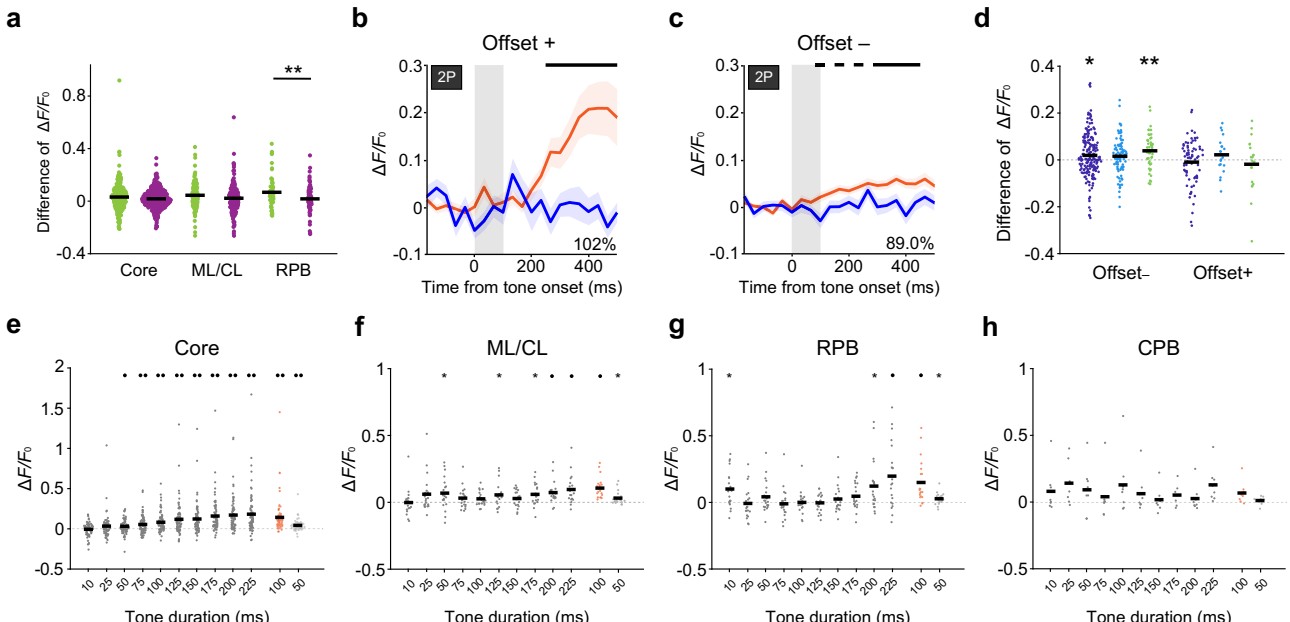

**Fig. 5 | Deviance detection is strongly induced in RPB neurons that show strong offset responses to the tone frequency used in the oddball paradigm.**
**a** Averaged deviance detection amplitude (200–400 ms after the tone onset) of dMMN-responsive neurons whose BF was $F_{odd}$ (green; $n = 251$ in core, 112 in ML/CL, and 68 in RPB) or not (purple; $n = 381$ in core, 174 in ML/CL, and 68 in RPB) in each region (unpaired $t$-test, two-sided, $**P = 0.0092$, $d = 0.45$ [RPB]). **b, c** Time course of calcium responses of RPB neurons whose BF were $F_{odd}$ with offset responses (**b**), and without offset responses (**c**) to the deviant stimulus (orange) and the control stimulus (blue). The neurons that showed significant responses in the oddball and/or many-standard paradigm were chosen from those whose BF was $F_{odd}$. $n = 21$ (**b**), $n = 47$ (**c**). Shading indicates SEM. The black line indicates the timepoints at which the difference was significant ($P < 0.05$, Wilcoxon rank sum test, two-sided, FDR-adjusted). The proportion of the deviance detection amplitude to the amplitude of

responses to the deviant tone in the oddball paradigm is also shown. **d**, Averaged early deviance detection amplitude (100–167 ms after the tone onset) of neurons whose BF was $F_{odd}$ without and with offset responses in the core (dark blue), ML/CL (light blue), and RPB (green). $*P = 0.0246$, $**P = 0.0029$, one-sample $t$-test, two-sided, with Bonferroni correction. Offset–, $n = 180$ (core), 92 (ML/CL), and 47 (RPB). Offset+ , $n = 71$ (core), 20 (ML/CL), and 21 (RPB). **e–h** Averaged response amplitudes (100–300 ms after the tone end) of the neurons whose BF was $F_{odd}$ with offset responses in the core (**e**), ML/CL (**f**), RPB (**g**), and CPB (**h**) to each of the 10–225-ms tones in the many-standards paradigm and to the 100-ms (orange) and 50-ms tones in the oddball paradigm in each region. $*P < 0.05$, $\cdot$ $P < 0.01$, $\cdot \cdot$ $P < 0.001$, one-sample $t$-test, two-sided, FDR-adjusted, $n = 71$ for core, 20 for ML/CL, 21 for RPB, and 8 for CPB.

the core or ML/CL (Supplementary Fig. 5h). The proportions of offset neurons were similar between the core (23%) and RPB (22%) (Supplementary Table 2). These results suggest that the prominence of the deviance detection in the RPB was related to the specific properties of the RPB neurons with offset responses.

The strong pure error signal that was detected in the RPB might originate from the dorsolateral prefrontal cortex (dlPFC) because (1) auditory MMN is also detected in the dlPFC in the macaque, although it is weaker than in the auditory cortex[42]; (2) transcranial direct current stimulation of the dlPFC reduces the amplitude of dMMN at frontal recording sites in humans[43]; and (3) the dlPFC shows auditory memory activity and interconnects with the RPB in the macaque[44]. To test whether the pure error signal can also be detected in the dlPFC, we injected AAV-GCaMP6f into the dlPFC (Supplementary Fig. 6a) and conducted one-photon imaging of the area. Auditory responses in the pure-tone paradigm were very weak, and the dlPFC imaging field did not show the BF (Supplementary Fig. 6b, c). In the dMMN paradigm, there was no clear calcium response to the deviant stimulus or the control stimulus (Supplementary Fig. 6d, e). These results suggest that the dlPFC was not the origin for the generation of strong deviance detection in the current dMMN paradigm.

**Feedback spreading of pure error signals from RPB to the core**
From the results so far described, we suspected that deviance detection induced in RPB neurons might affect deviance detection in the core and lateral belt. If so, it would be possible to detect deviance detection in axons spreading from the RPB to the core and lateral belt because there are axonal projections from the RPB to these areas[26,45].

Indeed, we did find that there were axonal projections from the RPB to L1 in the core and lateral belt (Fig. 6a). We injected AAV carrying GAP43-GCaMP6f into the RPB (Supplementary Fig. 7a) and then conducted one-photon imaging of the core (Supplementary Fig. 7b). GAP43-GCaMP6f is distributed specifically within the axonal compartment because of its signal peptide[46]; therefore, fluorescent signals far away from the injection sites should mainly reflect signals of axons originating from neurons around the injection sites. During the pure-tone paradigm, fluorescent change was observed in the core, and all core regions showed similar responses across different tone frequencies, although the fluorescence intensity was low (Fig. 6b and Supplementary Fig. 7b, c). This suggests that neurons with different BFs in RPB project their axons in a mixture over the core. During the dMMN paradigm, the response to the deviant stimulus was also small, but clearly detected in both rostral and caudal parts of the core, whereas the response to the control stimulus was hardly detected in the core (Fig. 6c, d). Regardless of whether the tone frequency was high or low, the deviance detection spread over the rostral and caudal parts of the core (Supplementary Fig. 7d, e). These results suggest that the back-projection of the pure error signal from RPB would continue to depolarize the broad auditory cortex for a few hundred milliseconds.

**RPB activity affects core responses**
Next, we determined whether RPB activity underlies induction of deviance detection in the core by injecting muscimol into RPB in animals in which GCaMP6f was expressed over the auditory cortex (Supplementary Fig. 8a, b) and conducting one-photon calcium

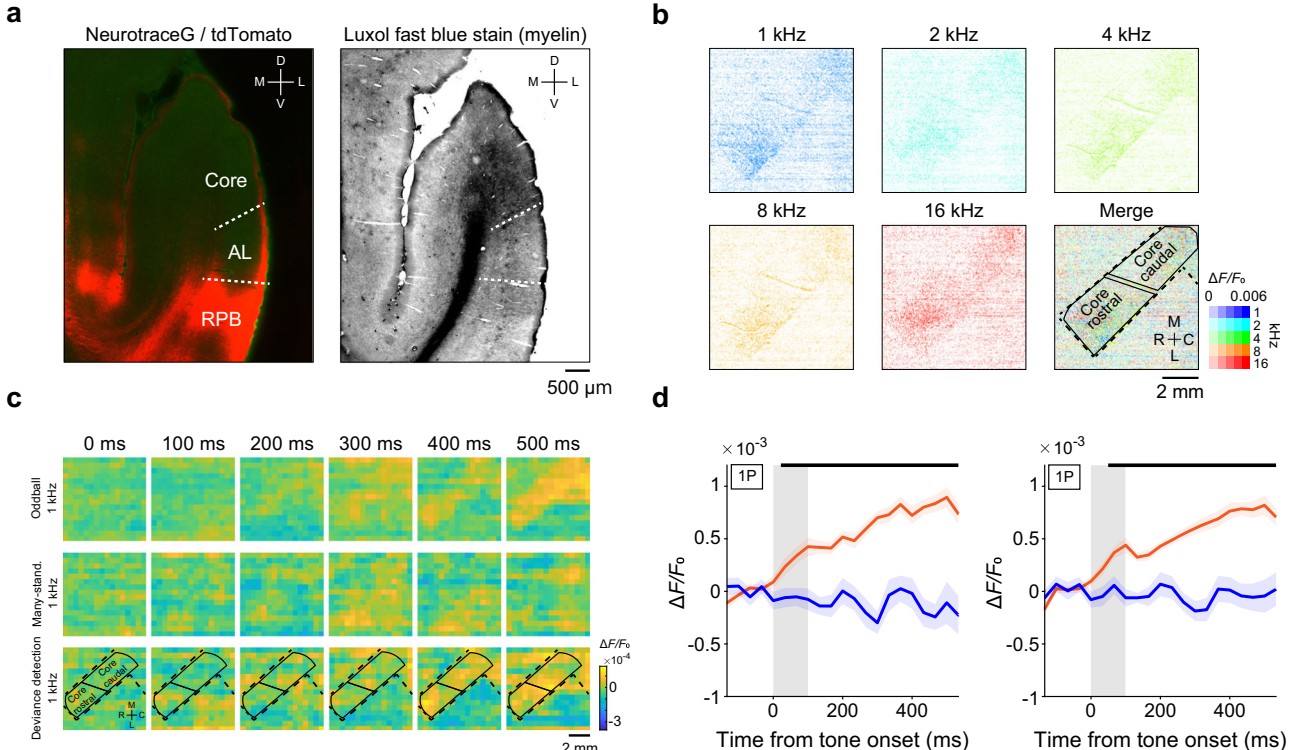

**Fig. 6 | Feedback projection of deviance detection from RPB to the core.**
**a** tdTomato expression in the coronal sections including the auditory cortex after
AAV-tdTomato injection into RPB. Left, tdTomato expression and Nissl staining.
Right, myelin staining. Dense myelin staining indicates the core area. A similar
image was obtained in one other slice. The boundary between the auditory cortical
sub-regions was decided on the basis of myeloarchitecture and cytoarchitecture. D,
dorsal; V, ventral; M, medial; L, lateral. **b** Calcium response maps for five pure tones
(1, 2, 4, 8, and 16 kHz). Bottom right, the best frequency map. For each pixel of the
256 × 256 pixel region, the color of the corresponding BF and amplitude was
assigned. Broken lines indicate the outer lines of the imaging area and solid lines
indicate the contours of rostral and caudal parts of the core that was inferred from
an atlas. To avoid fluorescence emission from the vicinity of the RPB injection site, a

black sheet was placed on the cover glass above the lateral belt and parabelt. The
vasculature image and epi-fluorescence image are shown in Supplementary Fig. 7a,
b. **c** Representative trial-averaged response map of the right core and corre-
sponding deviance detection (bottom) in the same animal as (**b**). The pixel size was
down-sampled to 16 × 16 pixels. **d** Session-averaged time courses of calcium
responses to the 100-ms tone in the oddball (orange) and many-standards (blue)
paradigms in the deviant tone-responsive regions in rostral (left) and caudal (right)
parts of the core (left, $n = 23$ sessions from one animal, 5 for 1 kHz, 4 for 2 kHz, 1 for
4 kHz, 6 for 8 kHz, and 7 for 16 kHz; right, $n = 22$ sessions from one animal, 5 for
1 kHz, 3 for 2 kHz, 1 for 4 kHz, 7 for 8 kHz, and 6 for 16 kHz). Shading on each line
indicates SEM.

imaging during the dMMN paradigm. RPB inactivation eliminated the
statistically significant deviance detection in the core, although the
response to the 100-ms tone in the many-standards paradigm was not
different between sessions with or without muscimol (Fig. 7a–d and
Supplementary Fig. 8c–e). By contrast, when the CPB was inactivated,
deviance detection in the core was clearly detected (Fig. 7b, d). In this
experiment, the core response should largely reflect the activity of the
core neurons. Thus, these results suggest that the error signal gener-
ated in RPB neurons was necessary for robust induction of deviance
detection in the core.

Finally, we determined whether RPB activity can enhance the core
response to the standard-tone stimulus. Marmosets were prepared to
express GCaMP6f in the core area and ChrimsonR[47] in the RPB (Sup-
plementary Fig. 8f), and we conducted one-photon calcium imaging
during a tone paradigm in which only the 50-ms tone was repeated
with an SOA of 550 ms. For 10% of the tones (randomly chosen), 300-
ms red laser illumination on the RPB immediately followed the end of
the tone (Fig. 8a and Supplementary Fig. 8g). This photostimulation
enhanced the response in the core region in which the activity was not
evoked by only photostimulation (Fig. 8b and Supplementary Fig. 8h,
i), and the response remaining after the end of the photostimulation
was significantly enhanced (Fig. 8c). In this calculation, we removed
the core region in which the fluorescence intensity was increased by
only photostimulation to remove the contribution of some red laser-
induced light artifacts and RPB activation-dependent, but tone

response-independent, activity (Supplementary Fig. 8h, j, k). Thus, RPB
activation immediately after the tone end can enhance the tone
response in core neurons in a non-linear manner.

## Discussion
In the current study, we used one-photon and two-photon calcium
imaging methods to demonstrate the feedback propagation of pre-
diction error signals from the higher-order auditory cortex to the
primary auditory cortex in the awake marmoset. Single-neuron-
resolution imaging allowed us to reveal that deviance detection was
strongly induced in those neurons whose BF was $F_{odd}$ over the core,
lateral belt, and parabelt. It has so far been unclear whether deviance
detection induction is prominent in neurons with a BF of $F_{odd}$ because
MMN has mostly been detected from population-averaged activity.
Even in previous experiments recording single-neuron activity, the
tone frequency used for frequency MMN was chosen to be near the BF
of the recorded neurons in mice[20,24], and the BFs of recorded neurons
in the macaque have not been examined[42].

The deviance detection in RPB neurons with BF of $F_{odd}$ and offset
responses was prominent, and its amplitude was much larger than that
of the response to 100-ms tones in the many-standards control para-
digm, even though the presentation probability of the 100-ms tone
was the same (0.1) in both paradigms. The RPB responses to 50-ms
tones in both paradigms were similarly suppressed to a great extent,
although the presentation probability of the 50-ms tone differed

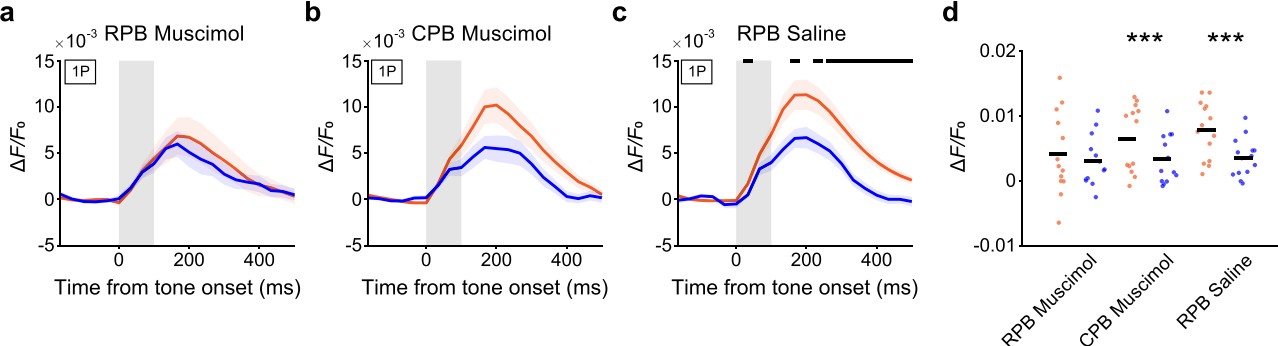

**Fig. 7 | RPB inactivation suppresses the deviant detection in the core. a–c** Session-averaged time courses of calcium responses to the 100-ms tone in the oddball (orange) and many-standards (blue) paradigms in the core regions responsive to the deviant tone after muscimol injection into RPB (**a**) ($n = 13$ from two animals: 3 for 1 kHz, 3 for 2 kHz, 2 for 4 kHz, 3 for 8 kHz, and 2 for 16 kHz), muscimol injection into CPB (**b**) ($n = 13$ from two animals: 2 for 1 kHz, 3 for 2 kHz, 2 for 4 kHz, 4 for 8 kHz, and 2 for 16 kHz), or saline injection into RPB (**c**) ($n = 14$ from two animals: 3 for 1 kHz, 4 for 2 kHz, 2 for 4 kHz, 2 for 8 kHz, and 3 for 16 kHz). The black lines indicate the period in which deviance detection was significant ($P < 0.05$, Wilcoxon rank sum test, two-sided, FDR-adjusted). Although there was no timepoint with significant deviance detection in (**b**), the averaged values over the 200–400 ms

period were different (**d**). This discrepancy might be because there was a larger amount of measurement variation and more data points in the former test than in the latter one. **d** Mean amplitude (averaged over 200–400 ms after the tone onset) of calcium responses to the 100-ms tone in the oddball (orange) and many-standards (blue) paradigms in each condition (paired $t$-test, two-sided, \*\*\*$P = 5.4 \times 10^{-4}$, $d = 0.73$ [CPB muscimol, $n = 13$ sessions], \*\*\*$P = 1.9 \times 10^{-4}$, $d = 1.12$ [RPB saline, $n = 14$ sessions]). The core response to the 100-ms tone in the many-standards paradigm was not significantly different among the conditions with RPB muscimol injection (unpaired $t$-test with Bonferroni correction, $P = 0.85$ for RPB muscimol ($n = 13$ sessions) vs. CPB muscimol, $P = 0.70$ for RPB muscimol vs. RPB saline, $P = 0.85$ for RPB saline vs. CPB muscimol).

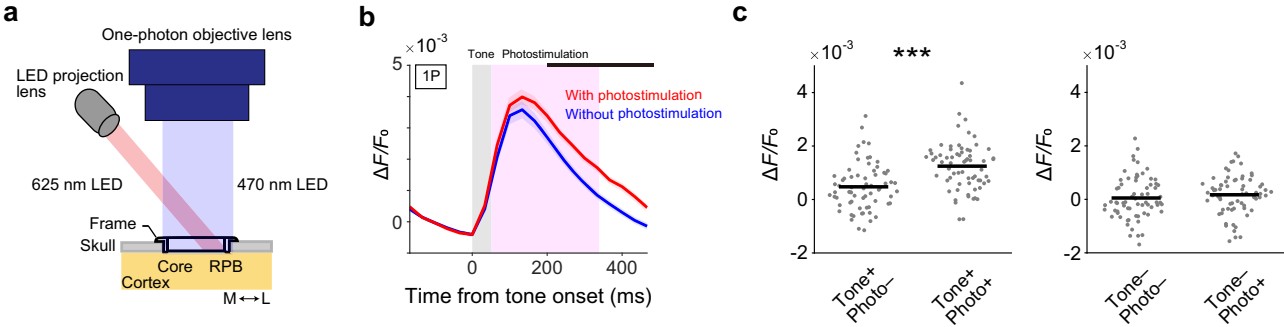

**Fig. 8 | RPB activation nonlinearly enhances the tone response in the core. a** Schematic illustration of simultaneous one-photon imaging (light blue) of the auditory cortex and red-light illumination (light red) on the RPB. **b** Calcium responses in the core area to tone and photostimulation of RPB (red), and to only tone (blue). The responses of the core region that showed significant responses to the tone followed by the photostimulation, but did not show significant responses to only photostimulation, were averaged. The black line indicates the period in which the amplitude was significantly different between the red and blue traces ($P < 0.05$, Wilcoxon rank sum test, two-sided, FDR-adjusted). $n = 68$ from three

animals; 19 for 1 kHz, 16 for 2 kHz, 15 for 4 kHz, 13 for 8 kHz, and 5 for 16 kHz from three animals. Shading on each line indicates SEM. **c** Left, calcium response amplitude (average from 333 ms after the tone onset to 500 ms after the onset) in the conditions with only tone, and in those with tone and photostimulation ($n = 68$ from three animals). Right, the amplitude (average from 333 ms after the tone onset to 500 ms after the onset) in the conditions with the virtual tone (no tone) and with only photostimulation ($n = 68$ from three animals). Wilcoxon signed-rank test, two-sided, \*\*\*$P = 3.0 \times 10^{-5}$, $d = 0.88$ (left).

substantially between these paradigms (0.9 vs. 0.1). Thus, it is rational to assume that the auditory cortex detected that the 100-ms tone deviated from the repetitive 50-ms tone, and that it was not from random-order tones with 10 different tones (10–225 ms). This property of RPB neurons might also be related to the finding that they significantly responded to 10-, 200-, and 225-ms tones whose durations were relatively different from the averaged tone duration (113.5 ms) in the many-standards paradigm. To understand the generic role of the RPB in dMMN, it is necessary to examine the different combinations of deviant and standard-tone durations including short deviant and long standard tones[7,11,48].

In the rat, attenuation of the auditory response to the repetitive stimulus is stronger in the higher-order auditory cortex than in A1, and stronger in the medial prefrontal cortex than in the higher-order auditory cortex[36,37]. Thus, it may be a cross-species commonality that the higher the hierarchy of the cortical areas, the weaker the response

to repeated stimuli. However, it remains unknown how the tone duration-independent attenuation occurred in RPB but not in the core, and how the deviant tone resumed the suppressed activity.

This attenuation mechanism may prevent the predictable signal from penetrating higher-order cortical regions, while the deviance detection function may allow unpredictable signals to be conveyed to these regions as error or salient signals and evoke cognitive and/or attention processes in a context-dependent manner[49], consistent with hierarchical predictive coding theory[10,11]. Although the dlPFC rarely responded to standard or deviant tones in the current oddball paradigm, the orbitofrontal cortex and area 10 might process the error signal because RPB has strong connections with these areas[44]. The induction of the strong deviance detection signal in the RPB could be evidence for the hierarchical predictive coding theory[10,11].

When LFP was recorded in the core, the significant deviance detection appeared around 200 ms after the tone onset and was still

detected after 400 ms (Fig. 2f). In the RPB, the significant deviance detection appeared around 120 ms after the tone onset and was still detected at around 400 ms (Fig. 3g). In a similar oddball paradigm for duration MMN in human and macaque (50 and 100 ms for the standard and deviant tones, respectively, and 500 ms for the SOA), the latency of the dMMN is around 130 ms after the tone onset and the dMMN is still detected after 200 ms[13,33]. Thus, the time course appears to be similar among these primate species. However, in the dMMN paradigm, the differences in the responses between different auditory cortical areas have not been examined. In the frequency MMN paradigm, deviance detection was measured in rat A1 and higher-order auditory cortex with LFP recording[24]. The deviance detection amplitude was larger in higher-order auditory cortex than in A1, and significant deviance detection was detected for 60–100 ms after the tone onset in A1, and for 25–200 ms in higher-order auditory cortex. As the deviant tone can be detected as the deviance immediately after the tone onset in the frequency MMN paradigm, the time window would be shifted to earlier than that in the current study. The difference in the time window length might reflect the difference in the size or structure of the auditory cortex between the rat and marmoset. In both the dMMN paradigm for the marmoset and the frequency MMN paradigm for the rat, the latency was earlier in the higher-order auditory cortex than in A1. Thus, in both rodents and primates, the higher-order auditory cortex may rapidly detect the prediction error signal and then feed it back to A1. In addition, in the frequency MMN paradigm of the rat, the medial prefrontal cortex neurons respond to the deviant tone with a latency of 120–180 ms from the tone onset in the frequency MMN paradigm[37]. In the marmoset, the pure deviance detection that is embodied in the rodent medial prefrontal cortex may be generated in the RPB, part of the higher-order auditory cortex, which may reflect more elaborate cortical sensory processing than in rodents.

Overall, we propose that the strong error signal generated in the higher auditory cortex during the dMMN paradigm explicitly propagates backward into the primary auditory cortex and alters deviant tone-specific activity (Fig. 9). The pure error signal may be sequentially extracted by RPB neurons without offset responses and then by those with offset responses. As the feedback projections originated mainly from layer 5 in the auditory cortex[50], layer 5 neurons downstream of L2/3 neurons in RPB may broadly back-project to L1 of the core and lateral belt. Based on the optogenetic experiment, even when the core was not sufficiently depolarized to exhibit action potentials upon RPB photostimulation alone, if the core had already been depolarized by the sound stimulation, it could be depolarized to exceed the action potential threshold by the RPB photostimulation. Feedback projection into L1 can increase the dendritic excitability of excitatory neurons[51]. Thus, the top-down feedback from RPB may further depolarize the core and belt neurons that are already depolarized during the tone presentation. In turn, some of these neurons in the local region might trigger action potentials ~200–400 ms after the tone onset, and then amplify RPB activity with offset response in the feedforward direction to maintain deviance detection signaling within the auditory cortex. This is consistent with a previous finding that L1 shows prominent MMN in the macaque primary auditory cortex[23]. The effect of RPB activation on the core might also be mediated by the dorsal and magnocellular divisions of the medial geniculate complex (which receive projections from the parabelt, and in turn have connections with the core L1[26]) because strong deviance detection is detected in these divisions of the auditory thalamus[24].

However, although two-photon calcium imaging has a single-cell resolution and is also used to examine visual MMN[34,35], its time resolution is lower than that of electrical recording. It is therefore necessary to conduct electrical recording to obtain a fine-scale time

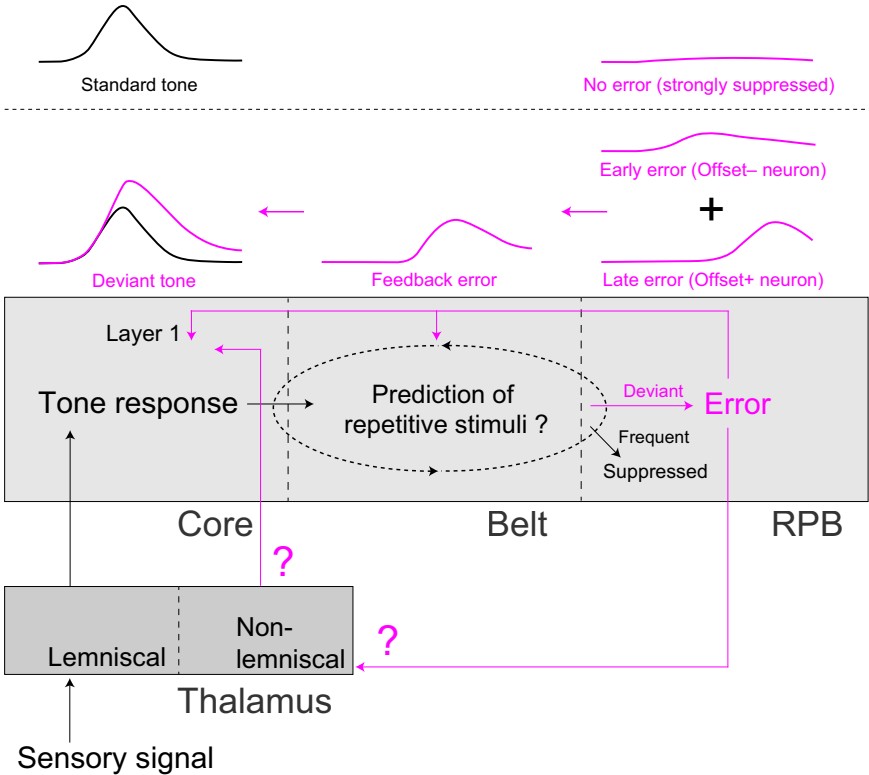

**Fig. 9 | Schematic of possible activity flows related to the deviance detection generation in the auditory cortex.** We propose that the core receives the sensory stimulus as it is, the difference between the repeated and deviant tones is calculated in unknown circuits in the auditory cortex (dotted ellipse), the RPB that highly adapts to the repetitive tone represents the pure error signal, the pure error signal in the RPB propagates backwards (magenta) to the core directly or through the belt and non-lemniscal thalamus, and the response to the deviant tone is enhanced in the core. This enhancement may be amplified within the auditory cortex and auditory thalamus.

sequence to reveal how the onset and offset responses interact with each other to detect a deviant tone and transmit the error signal across the core, belt, and parabelt. In addition, the present oddball paradigm did not induce strong adaptation in the core. It should be clarified whether similar deviance detection is detected in the core and RPB when the adaptation occurs in the core (for example, the SOA of 350 ms is used), and if so, whether the back-projection of the error signal from the RPB is important for the core deviance detection.

In the visual cortex, the feedback connections amplify activity to differentiate figures from the background[4]. Similarly, the rostral belt and RPB, but not the CPB, show their highest activation when macaques segregate acoustic figures consisting of multiple coherent frequency elements from acoustic backgrounds without a coherent element[52]. RPB neurons with offset responses also responded to the relatively deviant tones in the many-standards paradigm. Thus, if certain regularities are present, even if completely matched sounds are not repeated, RPB may be able to generate a signal in response to something unusual during that period of time. It is assumed that the rostral auditory pathway carries information on what the sound is and what the origin of the sound is, and that the caudal (dorsal) pathway carries information on where the sound comes from[53,54]. RPB may detect what the deviant sound is and what causes the deviation, whereas CPB may detect the movement of sound sources. The feedback propagation of the error signals within the auditory cortex and/or through the thalamus would play a critical role in enhancing the perception of the salient features of the sound. This might be one of the reasons why impairments in tone discrimination and everyday functioning in patients with schizophrenia are related to a reduction in the amplitude of the dMMN[17,55]. The sensitivity of error signals to environmental and cognitive contexts[8,56] may render them susceptible to perturbation in psychiatric disorders involving synaptic network communication and prediction, such as schizophrenia. Considering the abundance of feedback connections in the cerebral cortex, the feeding-back of error signals may be common in hierarchical cortical processing.

Although backpropagation is biologically implausible because it requires symmetric forward and feedback connections, feedback error signals transferred via connections with random feedback weights are theoretically able to adjust the weights (feedback alignment) and thereby facilitate learning of a variety of tasks[57,58]. If core and belt neurons can memorize a repeated tone, when a deviant tone is repeated these neurons may update the periodic memory according to the feedback activity in the RPB neurons. This suggests that the feedback error signals arrive at L1 apical dendrites of pyramidal neurons in the core and lateral belt, allowing dendritic compartments to actively induce synaptic plasticity[6,59–61]. Although it is crucial to examine whether the error signals from the RPB induce synaptic plasticity in core and belt neurons, our results suggest that backpropagation-like learning may be possible in the cerebral cortex.

## Methods
### Animals
All experiments were approved by the Animal Experimental Committee of the University of Tokyo and the Animal Care and Use Committees of the RIKEN Center for Brain Science. Eighteen laboratory-bred common marmosets (*Callithrix jacchus*; eleven males and seven females) were used in the present study. They were aged from 1–5 years and their weights ranged from 250–450 g during the experiments. For one-photon imaging of the core, three males (ages 2, 2, and 3 years) and two females (ages 3 and 5 years) were used. Among them, one male (age 2 years) and two females (ages 3 and 5 years) were used for two-photon imaging. For one-photon and two-photon imaging of the belt and parabelt, one male (age 1 year) and one female (age 2 years) were used. For LFP recording, one male (age 2 years) and two females (ages 2 and 5 years) were used. For muscimol inactivation of

the parabelt, two males (ages 3 and 3 years) were used. For structural imaging of the axons from the RPB, one female (age 3 years) was used. For calcium imaging of the axons from the RPB, one male (age 2 years) was used. For one-photon imaging of the dorsolateral prefrontal cortex, one male (age 5 years) was used. For photoactivation of the RPB, one male (age 2 years) and two females (ages 2 and 3 years) were used. For recording of LFP adaptation, two males (ages 1 and 2 years) were used. For estimating the spread of muscimol within the auditory cortex, one female (age 1 year) was used. Due to the low sample size in each experiment and because the sex does not significantly affect the results in the human MMN[62], we did not conduct sex-based analysis. The marmosets were kept on a 12:12 h light–dark cycle in individual cages adjacent to each other in rooms with multiple animals. They were not used for other experiments prior to the present study.

### Virus production
We used capsids of serotype 1 for all the AAV vectors used in this study. The AAV vectors for TRE3-GCaMP6f and TRE3G-tdTomato were described previously[31,63]. The plasmid for TRE3-Gap43-GCaMP6f (or tdTomato)-WPRE was constructed by combining the TRE3 from TRE3-ZsGreen1 (Clontech Laboratories, CA, USA), GCaMP6f from pGP-CMV-GCaMP6 (Addgene plasmid #40755; http://www.addgene.org/40755; RRID: Addgene_40755)[64] (or pAAV-hSyn-tdTomato [Addgene plasmid #51506; http://www.addgene.org/51506; RRID: Addgene_51506]), WPRE from pAAV-Ef1a-DIO hChR2 (E123T/T159C)-EYFP (Addgene plasmid #35509; http://www.addgene.org/35509; RRID: Addgene_3550)[65], and synthetic oligonucleotides encoding the palmitoylation domain derived from Gap43[46]. The plasmid for TRE3-ChrimsonR-mKate2 was constructed by replacing the GCaMP6f in the TRE3-GCaMP6f with ChrimsonR from pAAV-Syn-ChrimsonR-tdT (Addgene plasmid #59171; https://www.addgene.org/59171; RRID: Addgene_59171)[47] and mKate2 from pmKate2-f-mem (Evrogen, Russia). The AAV vectors from these plasmids were produced in HEK293 cells using a helper-virus-free system[66], purified using POROS CaptureSelect AAVX Affinity Resin (ThermoFisher Scientific, MA, USA) according to the manufacturer instructions, dialyzed against PBS using a Slide-A-Lyzer Dialysis Cassette (ThermoFisher Scientific), concentrated using Vivaspin 6-100 K (GE Healthcare, IL, USA), and titrated by qPCR using a KAPA SYBR Fast qPCR Kit (NIPPON Genetics, Tokyo, Japan). The plasmid pAAV-human synapsin I promoter (hSyn)-tTA2 was constructed by replacing the pAAV-thy1S-tTA2 promoter[67], and its AAV was produced and purified as described previously[68].

### Surgery and virus injection
All surgical procedures and AAV injections were performed under aseptic conditions, as described previously[67,69]. Before the virus injection, a universal primer (Tokuyama, Tokyo, Japan) was applied to the surface of the skull, and a head plate (CFR-1; Narishige, Tokyo, Japan) was attached to the skull using dual-cured adhesive resin cement (BistiteII or EstecemII; Tokuyama). A craniotomy was then made over the auditory cortex or dlPFC area and the dura mater was removed. The positions of the auditory sub-regions were determined according to the positions of two sulci (lateral sulcus and superior temporal sulcus) and the coordinates of an atlas map[70]. For virus injections, a pulled glass pipette (broken and beveled to a 30-μm outer diameter; Sutter Instruments, CA, USA) and a 5-μl Hamilton syringe were back-filled with mineral oil (Nacalai Tesque, Kyoto, Japan) and front-loaded with virus solution. A mixture of rAAV2/1-hSyn-tTA2 (final concentration, $3 \times 10^{12}$ vector genomes ml$^{-1}$) and rAAV2/1-TRE3G-GCaMP6f-WPRE (final concentration, $7 \times 10^{12}$ vector genomes ml$^{-1}$) was injected at 0.1 μl minute$^{-1}$ for 10 min with a syringe pump (KDS310; KD Scientific, MA, USA). The pipette was inserted vertically approximately 500 μm ventral from the brain surface.

For one-photon imaging of the axonal activity, a mixture of rAAV2/1-thy1S promoter-tTA2 (final concentration, $1.5 \times 10^{12}$ vector

genomes ml⁻¹) and rAAV2/1-TRE3G-Gap43 promoter- GCaMP6f-WPRE (final concentration, $1 \times 10^{12}$ vector genomes ml⁻¹) was injected at 0.1 µl minute⁻¹ for 7.5 minutes with a syringe pump. The pipette was inserted vertically approximately 500 and 1000 µm ventral from the brain surface.

To confirm the axonal projection from the parabelt to the core, a mixture of rAAV2/1-thy1 promoter-tTA2 (final concentration, $6.0 \times 10^{13}$ vector genomes ml⁻¹) and rAAV2/1-TRE3G-tdTomato-WPRE (final concentration, $3.8 \times 10^{12}$ vector genomes ml⁻¹) was injected at 0.1 µl minute⁻¹ for 5 minutes with a syringe pump. The pipette was inserted vertically approximately 500 and 1000 µm ventral from the brain surface.

For the optogenetics experiment, a mixture of rAAV2/1-thy1S promoter-tTA2 (final concentration, $1 \times 10^{11}$ vector genomes ml⁻¹) and rAAV2/1-TRE3G- ChrimsonR-mKate2-WPRE (final concentration, $1 \times 10^{12}$ vector genomes ml⁻¹) was injected at 0.1 µl minute⁻¹ for 5 minutes with a syringe pump. The pipette was inserted vertically approximately 500 and 1000 µm ventral from the brain surface.

After injection at each depth, the pipette was maintained in place for an additional 10 minutes, before being slowly withdrawn. A custom made polyacetal window consisting of a $9 \times 5$ mm or $10 \times 6$ mm rectangular glass coverslip (for auditory imaging), or a 5.5 mm diameter circular glass coverslip (for dlPFC imaging) with approximately 150 µm thickness (Matsunami Glass No.1, Osaka, Japan), was pressed onto the brain surface, and the edge was sealed with silicone elastomer (Kwik-Sil; World Precision Instruments, Hitchin, UK) and dental adhesive resin cement (Super bond; Sun Medical, Shiga, Japan).

## Pure-tone and dMMN paradigms

Head-fixed jacket-mediated trunk-restrained marmosets[69] were sat in a chair during the tone stimulus paradigms. Their arms and legs were gently wrapped with a cotton cloth over the jacket. Their face and body were monitored online through video by one or two experimenters, and if they demonstrated sleepy or agitated behavior the paradigm was stopped and they were returned to their home cage. All acoustic stimuli were generated using LABVIEW (National Instruments, TX, USA) and were amplified with a pre-main amp (PMA-2000AE; DENON, Kanagawa, Japan). The tone stimuli were presented by a head-phone speaker (K701; AKG, LA, USA) placed 10 cm away from the contralateral ear of the imaging side. The intensity of the tone stimuli was calibrated using a sound level measurement amplifier (NA-42; RION, Tokyo, Japan) around the ear position of the animal, and the difference from the microscope noise was kept at 60 dB by use of a sound-proof board.

In the pure-tone paradigm for tonotopy mapping, five pure-tone stimuli (1, 2, 4, 8, and 16 kHz) were pseudo-randomly presented. The duration of the tone stimulus was 500 ms, and the SOA was 3000 ms. Each stimulus was presented more than 10 times in each session. The duration of one pure-tone paradigm was ~3 minutes. We combined the oddball paradigm and many-standards control paradigm for the dMMN paradigm. In each oddball paradigm, one of five pure-tone stimuli (1, 2, 4, 8, and 16 kHz) was used for standard and deviant tones. The standard duration tone (50 ms) was repeatedly presented and randomly replaced by the deviant duration tone (100 ms) in 10% of trials. The major reason why we used the 100-ms tone as a deviant tone and the 50-ms tone as a standard tone was because we followed recent studies that showed dMMN in humans and macaques with the same combination of the tone duration[13,33]. We also followed a meta-analysis study that showed that the effect size of MMN deficits in schizophrenia is larger in long duration deviants than in shorter duration deviants[16]. In the many-standards paradigm, the same frequency tone used for the oddball paradigm was presented at 10 different durations with equal probability (10, 25, 50, 75, 100, 125, 150, 175, 200, and 225 ms). These tones were pseudo-randomly presented in each session. The SOA was 550 ms in both paradigms. Deviance detection was defined as the difference between the response magnitude to the deviant tone in the oddball paradigm and the response magnitude to the tone of the same duration as the deviant tone in the many-standards paradigm. One dMMN paradigm consisted of one oddball paradigm (~3 minutes) and one corresponding many-standards paradigm (~3 minutes). The average number of deviant tone trials in each oddball paradigm was 30.

For each one-photon imaging session with the dMMN paradigm, the order was generally pure-tone paradigm, oddball paradigm, and many-standards paradigm, although in some cases the order was shuffled. When the dMMN paradigms were conducted multiple times within a session, the tone frequencies were different among the paradigms. For the majority of the two-photon imaging sessions with the dMMN paradigm, the pure-tone paradigm was also conducted in the same L2/3 field of view. The same field of view at different depths (L1 and L2/3) was frequently imaged in a session.

## In vivo imaging

All calcium imaging was performed between 1.5 and 4 months after AAV injection. One-photon imaging was conducted with a variable zoom microscope (Axio Zoom.V16; Carl Zeiss, Jena, Germany) with a 2.3× objective (Plan-NEOFLUAR Z 2.3×/0.57; Carl Zeiss). A monochromic EM-CCD camera with $1024 \times 1024$ pixels (iXon Ultra 888; Andor Technology, Belfast, UK), or an sCMOS camera with $2048 \times 2048$ pixels (SONA 4bv-11; Andor Technology), was used as a photodetector. Images were acquired using a 470-nm blue LED (M470L3; Thorlabs, NJ, USA), and the laser power under the objective was 1.7 mW. The optical axis was adjusted to be nearly perpendicular to the plane of the cranial window on the imaging areas by tilting the animal chair. Images of $512 \times 512$ pixels were acquired at a frame rate of 30 Hz using Andor software (Andor Technology), regardless of the camera used.

Two-photon imaging was conducted with a custom-built two-photon microscope (Olympus, Tokyo, Japan) equipped with a water immersion objective lens (XLPLN10XSVMP, numerical aperture of 0.6, working distance of 8 mm; XLSLPLN25XSVMP2, numerical aperture of 0.95, working distance of 8 mm; Olympus), and Nd-based fiber-delivered femtosecond laser illumination (Femtolite FD/J-FD-500, pulse width of 191–194 fs, repetition rate of 51 MHz; IMRA, MI, USA) at a wavelength of 920 nm, as in a previous study[69]. In this study, the laser power under the objective was 30–40 mW. The optical axis was adjusted to be nearly perpendicular to the plane of the cranial window on the imaging areas by tilting the microscope body (30–45 degrees). Images of $512 \times 512$ pixels were acquired at a frame rate of 30 Hz using FV30S-SW software (Olympus).

## Extracellular recording and the standard paradigm

The LFP was recorded in the imaging areas of the three marmosets using a tungsten microelectrode (500 kΩ; World Precision Instruments). Before the extracellular recording, the glass coverslip of the imaging window was replaced with a silicone sheet (6-9085-12; AS-ONE Corporation, Osaka, Japan). The electrode was inserted perpendicular to the cortical surface through the silicone sheet using a micromanipulator (Narishige), and the recording sites were less than 300 µm from the cortical surface. During the recording, silicone elastomer (Kwik-Sil; World Precision Instruments) was filled into the window to suppress brain pulsation. Neural data were amplified 1000× or 10000×, filtered (1–3000 Hz; DAM80, World Precision Instruments), and sampled at 24 kHz (NI-DAQ, National Instruments).

In two other marmosets, only LFP recording was conducted in the core and RPB. First, auditory responses were measured at each penetration site in the pure-tone paradigm, and one to three tone frequencies that induced apparent responses were determined. Then, we conducted the standard paradigm, in which ten 50-ms tones with each of the responsive frequencies at that site and SOAs of 550 and 350 ms (a standard-tone block) were repeated with an inter-block interval of 10 s for approximately 5 minutes (standard paradigm). The many-standards paradigm with the first tone duration of 100 ms in each paradigm session was also conducted on one of the two marmosets. A

comparison was made between the response to the first 100-ms tone and the average of the responses to the 100-ms tones that were randomly presented afterwards in the same paradigm session (Supplementary Fig. 3g).

## Muscimol-injection test

In two marmosets, muscimol (0.12 µl, 5 µg/µl; SIGMA M1523-5MG, Merck KGaA, Darmstadt, Germany) or a placebo (the same amount of saline) was injected into the parabelt region at a depth of 1 mm from the cortical surface at 0.1 µl minute$^{-1}$ for 1.2 minutes with a syringe pump through a pulled glass pipette. After injection of muscimol or saline, animals were maintained in their home cages for 2 h before the dMMN paradigm. Several injections were conducted using a double-blind test in which the experimenter did not know whether drug or placebo was injected.

To estimate the lateral spread of muscimol in the auditory cortex, as described in our previous study in mice[71], a small volume of muscimol BODIPY TMR-X conjugate (120 nl; 5 µg/µl; M2400, ThermoFisher Scientific, MA, USA) was injected into the RPB in another marmoset under the same conditions as in the pharmacological inactivation experiment. We then conducted one-photon imaging for 3 h after the injection. Since the full width at half maximum remained almost unchanged over time, we assume that some of the injected molecule broadly diffused from the insertion site of the glass pipette. Since the molecular weight and hydrophilicity of the muscimol BODIPY (607) and muscimol (114) differed, and these diffused in the extracellular space in the cerebral cortex, the spread of muscimol could not be accurately estimated in this experiment. However, given that the diffusion constant in aqueous solution is inversely proportional to the square root of the molecular weight, and that the diffusion distance is proportional to the square root of the diffusion constant[71], we supposed that the spread of muscimol was approximately 1.5 times greater than that of muscimol BODIPY, and the distance from the injection point to the point with the half maximum of the intensity would therefore still be less than 1 mm. The distance between the injection site and the putative border between the RPB and AL was more than 1 mm in all muscimol-injection experiments. Thus, we consider the muscimol effect to be mainly due to inactivation of the RPB, but not the AL or core.

## Optogenetic stimulation of the RPB

This experiment was conducted in three marmosets. A 625-nm laser (M625F2 – 625 nm; Thorlabs, NJ, USA) was delivered through a 1000-µm optical fiber (M35L02, Ø 1000 µm, 0.39 NA; Thorlabs) and projection lens (MAP1030100-A, 1:3.33 Matched Achromatic Pair, f1 = 30 mm, f2 = 100 mm; Thorlabs). The diameter of the laser beam illumination was estimated to be 2 mm. The 50-ms-duration tone was repeatedly presented at an SOA of 550 ms, and 10% of these tones (randomly chosen) were immediately followed by the laser illumination (10 lots of 10-ms pulses with an interstimulus interval of 20 ms). The laser power was set to 11 mW. These sessions were conducted two to four times a day. In the without-tone control sessions, a virtual 50-ms tone was presented with an SOA of 550 ms, and 10% (randomly chosen) of these virtual tones were immediately followed by the laser illumination. The without-tone control sessions were conducted one to three times on each day.

## Histology

The marmosets in which tdTomato was expressed were deeply anesthetized by an intraperitoneal injection of sodium pentobarbital and perfused transcardially with 4% paraformaldehyde (PFA) 4 weeks after the AAV injection. The brain was removed and put into 4% PFA for 3 days at 4 °C. The brain was then embedded in 3% agar in 0.1 M phosphate-buffered saline (PBS) and sliced coronally into 50-µm sections with a vibratome (VT-1000S; Leica Biosystems, Nussloch, Germany). The sections were mounted on glass slides with ProLong Diamond Antifade Mountant (Thermo Fisher Scientific). The brain

sections were divided into two series: odd-numbered sections underwent myelin staining using the Kluver-Barrera staining method (Luxol Fast Blue Solution; ScyTek Laboratories, Inc., UT, USA), and even-numbered sections underwent Nissl staining (NeuroTrace 500/525; Invitrogen, MA, USA). All of the mounted section images were acquired using a fluorescence microscope (BZ-X700; Keyence, Osaka, Japan). The areal boundary of the auditory sub-region was decided on the basis of the myeloarchitecture and cytoarchitecture in the coronal sections[29,45,70].

## Analysis of one-photon imaging data

All calcium imaging data were analyzed using MATLAB (R2018a, 9.4.0.949201; and R2020b, 9.9.0.2037887; MathWorks, MA, USA). To make the tonotopic maps, the recorded raw data (512 × 512 pixels) were binned to half size (256 × 256 pixels) and plotted with five gradations of color brightness according to maximum intensity. Analysis of the response to the pure-tone paradigm was performed by binning the recorded raw data into 16 × 16-pixel segments. The calcium response was presented as $(F-F_0)/F_0$ (i.e., $\Delta F/F_0$), where the fluorescence intensity averaged over 10 frames (333 ms) before the tone onset was $F_0$ (baseline activity), and the intensity averaged over 45 frames (1.5 s) after the tone onset was $F$ (evoked activity). The statistical significance of the pure-tone responsiveness was determined according to the following two criteria: (1) the presence of three consecutive timepoints whose averaged $F$ exceeded 1.96 times the standard deviation (SD) after the tone onset; and (2) a mean magnitude of $F$ over 45 frames (1.5 s) after the tone onset that was significantly larger than $F_0$ (Wilcoxon signed-rank test, two-sided). In the one-photon image analysis of the dMMN paradigm, the response amplitude of each binned segment was calculated by $\Delta F/F_0$, where the averaged intensity over five frames (167 ms) before the tone onset was $F_0$, and the averaged intensity over 15 frames (0.5 s) after the tone onset was $F$. The two criteria for statistical significance were similar to the case for the pure-tone paradigm. In the dMMN paradigm, deviance detection responses were defined as the intensity for the deviant tone (100 ms) in the oddball paradigm minus the intensity of the 100-ms tone in the many-standards control paradigm. In this calculation using the calcium responses, 100-ms tones that previously had 10-, 25-, and 50-ms durations in the many-standards paradigm were used to eliminate the effect of the large decay of the calcium response in the previous tones with long duration in the $F_0$ estimation. The slope of the decay of the calcium response before the tone onset was similar between the 50-ms tone stimulus before the deviant tone in the oddball paradigm and the 10–50-ms tone stimuli in the many-standards paradigm (Supplementary Fig. 2c). However, when all 100-ms tones in the many-standards paradigm were used, deviance detection was apparently detected (Supplementary Fig. 2c). The timepoint with a statistically significant deviance detection was defined as the timepoint when the responses to the deviant tone in the oddball paradigm were significantly higher than the 100-ms tone in the many-standards paradigm ($P < 0.05$, Wilcoxon rank sum test, two-sided, FDR-adjusted). In this analysis, the $\Delta F/F_0$ in the pixel segments that showed significant responses to the deviant tone in the oddball paradigm in the 16 × 16-pixel segments were averaged. In the muscimol-injection experiment, all pixels within the core region were used for analysis. In the optogenetic experiment, the statistical significance of the responsiveness to the photostimulation was determined using the same two criteria used for the one-photon image analysis of the dMMN paradigm, but only trials with the photostimulation were used. The response was averaged over the core areas showing significant responses to the tone and photostimulation, except for the area that showed significant response to the photostimulation without the tone in at least one session on each day.

## Analysis of two-photon imaging data

Two-photon images were first realigned with a finite Fourier transform algorithm to remove tangential drifts[69]. Regions of interest (ROIs)

corresponding to neuronal somata were extracted from the time series of the images using a constrained non-negative matrix factorization algorithm (https://github.com/flatironinstitute/CaImAn). The number of ROIs for the search was set at 100–150 for each field of view. The fluorescence of each ROI ($F_{roi}$) was calculated by averaging the fluorescence intensity over all pixels within the ROI in the motion-corrected raw images. The area not labeled as ROI was defined as neuropil within the recorded two-photon images. To correct for neuropil contamination in $F_{roi}$, the fluorescence of surrounding neuropil ($F_{neuropil}$) was calculated. For each ROI, the mean fluorescence of a 2 μm neighborhood in the neuropil was defined as $F_{neuropil}$. Using all the timepoints except for those where $F_{roi}$ exceeded the time-averaged mean plus 1 SD, $F_{roi}$ values were regressed against $F_{neuropil}$ using an iteratively re-weighted least-squares algorithm (robustfit in MATLAB), and the slope of the regression line was defined as a contamination factor for the ROI[72,73]. In each imaging field, the median value of the contamination factors of the ROIs was defined as $r$ (range, 0.1–0.2). Then, we estimated the fluorescence of each ROI without neuropil contamination as $F = F_{roi} − r × F_{neuropil}$.

The calcium response was presented as $(F−F_0)/F_0$ (i.e., $\Delta F/F_0$), where the averaged intensity over 10 frames (333 ms) before the tone onset was $F_0$ (baseline activity). If a neuron showed a significant response to at least one of the five tone frequencies in the pure-tone paradigm ($P < 0.05$, Wilcoxon signed-rank test, two-sided) and $\Delta F/F$ was more than 0.01, it was defined as a tone-responsive neuron. For each tone-responsive neuron, the tone frequency with the maximum response over 30 frames (1 s) after the tone onset was defined as BF. We calculated the fluorescence intensity during 1 s after the tone onset minus the fluorescence intensity at baseline (0.5 s before the tone onset) for each of five sound frequencies in the pure-tone paradigm, and then calculated the selectivity index (SI) as the sum of the five relative intensities obtained by scaling the response to the best frequency to a value of 1. The maximum value of the range was 5 when the neuron responded equally to all frequencies, and the minimum value was 1 when the neuron responded to only one frequency (there was no negative response). Offset+ (or onset+) neurons were defined as those neurons in which the $F$ averaged over 15 frames (0.5 s) after the end (or onset) of the BF tone presentation was significantly larger ($P < 0.05$, Wilcoxon signed-rank test, two-sided) than that over five frames (167 ms) before the end (or onset) of the BF tone presentation in the pure-tone paradigm. Neurons that fulfilled both criteria were also defined as onset+ offset+ neurons. In this classification, onset+ offset+ neurons can show two peaks or sustained responses, depending on the peak timings of the activities during the tone presentation period and during the period after the tone end. If a neuron with BF that was used in the dMMN paradigm showed significantly higher ($P < 0.05$, Wilcoxon signed-rank test, two-sided) fluorescence intensity averaged over 15 frames (0.5 s) after the tone onset ($F$) than the intensity averaged over five frames (167 ms) before the tone onset ($F_0$) in at least one of twenty groups of trials (a group of the deviant tone in the oddball paradigm, nine randomly assigned groups of standard tones in the oddball paradigm, and 10 groups of tones with different durations in the many-standards paradigm), and if $\Delta F/F_0$ in the significant group was more than 0.01, then it was defined as a dMMN-responsive neuron. The definition of deviance detection and the statistical analysis were the same as for the one-photon imaging.

### Analysis of extracellular recording data

Extracellular recording data were analyzed using MATLAB (R2018a, 9.4.0.949201, and R2020b, 9.9.0.2037887; MathWorks). The raw extracellular recording data were compared within non-overlapping 10-ms bins from 100 ms before the tone onset to 500 ms after the onset. The timepoint showing a statistically significant deviance detection was defined as the timepoint when the responses in the

deviant tone in the oddball paradigm were significantly higher than the same duration tone in the many-standards control ($P < 0.05$, Wilcoxon rank sum test, two-sided, FDR-adjusted), as in the calcium imaging.

### Statistical analysis

Statistical analyses were performed using MATLAB (R2018a, 9.4.0.949201, and R2020b, 9.9.0.2037887; MathWorks). The Wilcoxon signed-rank test, Wilcoxon rank sum test followed by false discovery rate (FDR), and Student's $t$-test followed by Bonferroni correction were used as appropriate. All these tests were two-sided. Cohen's $d$ was used to calculate the effect size. No statistical tests were run to pre-determine the sample size. Data are presented as mean ± SEM or swarm plots with the means unless otherwise noted.

### Reporting summary

Further information on research design is available in the Nature Portfolio Reporting Summary linked to this article.

## Data availability

All data are available from the corresponding author. The data used in the graphs in this study are provided in the Supplementary Information/Source Data file. Source data are provided with this paper.

## Code availability

All computer codes are available from the corresponding author.

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

## Acknowledgements
We thank M. Kondo for technical support with the tone presentation devices. We thank Y. Hirayama, M. Hirokawa, Y. Takahashi, and A. Yamamoto for animal handling. We thank S. Koike, M. Tada, and T. Toyoizumi for helpful discussion, and C. Yokoyama for helpful commenting and editing of the manuscript. We thank K. Deisseroth for providing pAAV-Ef1a-DIO hChR2 (E123T/T159C)-EYFP, E. Boyden for pAAV-Syn-ChrimsonR-tdT, Douglas Kim and the GENIE Project for providing pGP-CMV-GCaMP6f, and Hongkui Zeng for providing pAAV-hSyn-tdTomato. This work was supported by Grants-in-Aid for Scientific Research on Innovative Areas (17H06309 to M.M. and 21H00302 to T.E.), for Transformative Research Areas (A) (23H04977 to T.E., 23H04674 to T.U., and 22H05160 to M.M.), for Scientific Research (A) (19H01037 and 23H00388 to M.M.), for Scientific Research (B) (20H03546 to T.E.), and for Young Scientists (A) (17H04982 to Y.M.) from the Ministry of Education, Culture, Sports, Science, and Technology, Japan; AMED (JP19dm0207069 to T.U., M.K., K.Ka., and M.M.; JP15dm0207001 to M.K., T.Y., and M.M.; JP18dm0207027 to M.M.; JP19dm0107150 to M.M.; and JP19dm0207085 to T.E. and M.M.); RIKEN SPDR program (to K.O.); and the Tokyo Society of Medical Sciences (to T.E.).

## Author contributions
K.Ka. and M.M. designed the study. K.O. and T.E. performed extracellular recording, animal surgery, AAV injection, histology, in vivo imaging, and data analyses. T.E. and Y.M. set up in vivo imaging systems with the tone stimulation device. S.-I.T. designed and made cranial glass windows. T.U. and M.K. optimized the paradigm parameters. Y.M., M.T., A.W., K.Ko., H.M., and T.Y. designed and prepared the AAV vector system. K.O., T.E., and M.M. wrote the paper with comments from all other authors.

## Competing interests
The authors declare no competing interests.
