## [Peer Review File · Nature Communications]

Change detection in the primate auditory cortex through feedback of prediction error signalsREVIEWER COMMENTS

Reviewer #1 (Remarks to the Author):

The paper is generally well written, and on the whole has the beginnings of an interesting argument. However, there are some critical problems with the experimental design and analysis, and these require major adjustments to be addressed.

1. I found surprising that not adaptation for the standard tone was found in the core (P7) even more considering the deviant and standard tone were the same frequency. The paper does not discuss the adaptation that is occurring in repetitive (standard) signals, and assumes that the responses to all presented standards are the same. This is not the case, and would need a complete reanalysis. Without a careful examination of what adaptation is and is not occurring, it is impossible for the authors to rule out that all of their results are simply the result of adaptation.

2. Although the paper does briefly touch on the need for electrophysiological recordings, it does not discuss one of the primary reasons for this: temporal resolution. At no point is the temporal resolution of the imaging setup so much as mentioned. Buried in the methodology, there is a single reference to the imaging being viewed at 30Hz. However, it is extremely unlikely that the imaging is being recorded at 30Hz - the best temporal resolution I have seen in papers using this type of imaging talk about 60ms between images. That is quite a bit slower than 30Hz, and further is a much larger interval than both the differences between different stimuli lengths and the length of the shortest stimulus presented. As such, the findings presented are unconvincing. They honestly need to be backed up with electrophysiology across the board, and not a single experiment where the researchers did an electrophysiological recording, got similar results to their imaging, and called it a day. Fine temporal changes in stimuli require fine temporal measurements.

3. Authors observed that offset neurons have the strongest deviant detection signal (DD) and that this DD was earlier and stronger in the RPB area (figure 4). I think it is important to quantify how was the proportion and latencies of those responses across core, belt and parabelt areas and whether the offset magnitude correlate with the tone duration during the equal probable many-standard protocol. Is the DD stronger in the belt areas because they have more offset neurons? I was wondering whether offset neurons are the detectors for duration deviants or whether a similar trend across neuron types and areas would be expected for frequency deviants as well.

4. I think authors need to consider other scenarios that they are missing due to technical limitations. In the predictive coding literature usually, a canonical model is proposed in which the feedback signals are send from the deep to superficial layers and an extended circuit model in which in addition there is superficial-to-superficial layer feedback projections. Authors mentioned that the majority of the 2-photon imaging and dMMN paradigm were done in the L2/L3 layers which is understood due to the calcium imaging technique. So, it would be welcome authors discuss other possible signal routing that could be occurring during deviance signaling. Is it possible that the lack of DD signal in the CPB is due to the fact that in higher order areas the feedback signaling arise from deep layers rather than superficial?

5. There is evidence (Wang, X.-J. (2022). Theory of the Multiregional Neocortex: Large-Scale Neural Dynamics and Distributed Cognition. Annual Review of Neuroscience 45, 533-560.) that indicates that temporal integration of neurons increases for areas of higher hierarchical order. Is it possible that if testing shorter durations like 25 vs 75 ms, the DD in the core area would be enhanced since that durations better match their time windows.

6. It is clear the changes in the latencies and magnitudes of the DD signals across areas. Do the dynamics of the DD (duration, latency) follow the same as previously described for frequency deviants in the rat or macaque model, for example? It would be nice to include some comparison of the neural dynamics across areas, sound features and animal models since few studies performed multisite recording with the same paradigm.

7. It would be important to include a summary figure that contrast the temporal dynamics of the

feedforward and feedback routing, i.e., how the neural activation occurs across areas and layer (1 vs 2/3) when the sound duration is equal-probable, repetitive and deviant. Maybe using a colormap for tracing the trajectory of the activation appearing first in core and then in the belt when equal-probable and then it switching to the RPB when deviant.

8. In previous studies it has been found that the tuning width and DD magnitude are correlated for other sound features such as frequency. In methods it was indicated that a tuning width index was estimated. It would be interesting to know whether neurons with larger receptive fields also had stronger dMMN in each area?

9. I am concerned regarding any contamination due to the fixed SOA on the magnitude of the DD or standard response. Why to use a fixed SOA = 550 rather than a fixed ISI? How was the variability of the activation across ISIs of different length?

10. It was not clear for me at which intensity the tone was delivered. It is important since looking at the marmoset audiogram there are clear difference all along the tested tones (1,2,4,8,16 kHz).

11. Authors found that the DD mainly occurs in the region where the BF was the deviant/standard stimulus (P8). It is interesting to know whether deviance detection for duration has been done using broad-band noise rather than pure tones for example? Would authors expect a similar trend in the origin and propagation of the feedback error signal?

12. Was the DD signal similar across tone frequencies or there was a bias to any frequency for duration deviance detection?

13. Regarding the activation and deactivation experiments. I have some questions on that:

- Authors injected AAV into the RPB and found that RPB project to L1 in core and belt. Can also authors indicate in which RPB depth the injections were done? Which was the main projection layer authors were manipulating?
- Why authors waited 2 hours after the muscimol injection to perform the image acquisition? How do authors control for muscimol spread across areas along time?
- Why 10 and 45 frames were used for estimating baseline and evoked activity?
- In methods when referring to time points I suggest to indicated also their equivalence in ms.
-

Other minor issues

There is incredibly heavy use of abbreviations and acronyms in this paper. It absolutely needs to have a reference table for them all. Further, when possible, some of the abbreviations/acronyms should be avoided or spelled out. The constant use of them makes the paper more difficult to read.

56: Repetitive standard stimuli induce adaptation. It needs to be discussed. It's entirely possible all results seen are due to adaptation unless it is accounted for.

61. Missing reference

85 - 87: This sentence is unclear. Pretty sure macaques and humans have all three regions of the superior temporal gyrus as well.

126: Discussion of the calcium responses to 100 and 50 ms stimuli does not account for adaptation in the presented standard.

140: What kind of electrical recording? surface electrodes on the cranium? It needs to be specified.

153: RPB is not defined. I assume it means Rostral Parabelt, but I could be wrong and the paper cannot rely on the reader to know what it means. Please never use an acronym without first defining it.

170: This is probably adaptation occurring - the discussion of control stimulus responses being weak without discussing adaptation leaves the conclusion completely unsupported.

177: RES was defined once, two pages earlier, and never mentioned again until here. Again, acronyms need to be used sparingly to increase readability, not to obscure the discussion.

179 - 182: I'm not at all convinced when you have three data points, and are making conclusions on the one data point at the end of your data set. Increased the number randomly presented deviant lengths: 75, 100, 150 are not sufficient by themselves.

188: Clearer and faster aren't objective observations. There needs to be statistical analysis to back it up.

218: Was there an electrophysiological response? With the low temporal resolution of the imaging, no response doesn't necessarily mean there is no response. Even the imaging resolution is 50ms, which I doubt, neurones could fire 40 times in that window, and not be captured.

244: largely reduced needs to be backed up with stats!

245: I like the idea of the many-standard paradigm, but with your limited (again I assume, there has been NO discussion of imaging temporal resolution) temporal resolution, you are very constrained by what you can and cannot observe.

279: This argument seems plausible to me.

291: I am much less convinced by this.

Reviewer #2 (Remarks to the Author):

The manuscript by Obara et al discovered that prediction error signal was generated in the rostral parabelt (RPB) of higher-order auditory cortex. Such error signal was manifested as deviance detection (DD) in the neural signal, which was measured in the current study by using one-photon (1P) or two-photon (2P) calcium imaging or extracellular recordings in the auditory cortical areas of core, belt, and parabelt of awake common marmoset, during presentation of auditory oddball paradigm. DD has been reported in previous studies to be observed in a macroscopic ECoG signal as mismatch negativity (MMN) in the frontal and temporal lobes of the brain, which is an important biomarker in neuropsychiatric disorders. Here the authors examined DD in primary and high-order auditory cortical areas upon presentation of auditory duration oddball paradigm. Distinct types of tone responses in single neurons at these auditory cortical areas were characterized according to their response onsets/offsets. A pure error signal, or residual error signal (RES), was for the first time discovered in RPB neurons, especially for RPB neurons of certain response type, when RPB neurons showed very strong response to oddball tone stimulus and very weak response to control tone stimulus. Using pharmacology and optogenetic stimulation, the authors found evidences supporting that the error signal, here DD, was backpropagated from the higher-order auditory area RPB to primary auditory cortex.

I think this work is original and of significance for understanding cortical processing of error signal. The evidence for supporting main findings are clear, although more experiments and analysis could be performed to make the findings more thorough. I find the description and discussion of results are sometimes too concise. It would be helpful if the authors can be more explicit in the results description and elaborate the meaning of results. In my opinion, the manuscript can potentially be considered for publication after major revision.

My main comments are as follows:

1) I find it surprising that there was no clear simple adaptation of responses to the standard tones in the 1P calcium signal upon duration oddball paradigm stimulation (Fig 2b). What would be the

reason for this?

2) The authors chose to calculate DD as the difference in calcium signal between responses to oddball tone in oddball paradigm and the responses to the same duration tone in the many-standards paradigm. How would the DD be when calculated in a more conventional way, i.e., comparing the calcium responses to oddball tone in the oddball paradigm and those to the standard tone in oddball paradigm?

3) Would the DD be still present when swapping the tone duration in oddball paradigm, i.e., using shorter duration of 50 ms for oddball tone and longer duration of 100 ms for standard tone and compare with responses to the 50-ms tone in the many-standards paradigm?

4) Although the authors found that tonotopy was not clear in lateral belt and parabelt regions when using 1P calcium imaging (Fig 3b, line 154), they identified best frequencies (BF) of single neurons in these areas when using 2P calcium imaging (Ext Fig 3b-3d, line 158). This seems contradictory. Are neurons of different BF intermingled in these areas?

5) The authors further characterized different response types according to the 2P calcium signal from neurons in different auditory cortical areas (Ext Table 1). Some offset responses, e.g., those in Ext Fig 3b-3d, look to me more like onset responses of longer latencies. I also wonder whether responses of onset+/offset+ are of two peaks or sustained responses. It would be helpful to provide exemplary traces and latencies for responses of onset+/offset-, onset+/offset-, onset-/offset+, and onset-/offset-.

6) The calculation of selectivity index (SI) is not clear. It would be helpful if the authors can provide exemplary traces showing similar tuning widths across core, ML/CL, RPB, and CPB (Ext Table 1).

7) The authors observed very strong responses to oddball tone in the oddball paradigm and very weak responses to the control stimulus in the many-standards paradigm using 1P calcium imaging in RPB, suggesting the presence of pure error signal (Fig 3d-de). As RPB neurons can be responsive to pure-tones of longer duration of 500 ms (Ext fig 31-3d, using 2P calcium imaging), I wonder if such pure error signal in RPB exists only for shorter tone duration. Precisely, I wonder what would be the range of oddball tone duration for eliciting pure error signal, or RES, in RPB.

8) The time course of DD appears to be faster when using 1P calcium imaging compared to when using 2P calcium imaging (e.g., Fig 3f vs Fig 4g, and perhaps Fig 4a vs fig 4b-4d). What would be the reason for this? It would be very helpful to label clearly how signal was obtained (e.g., using 1P or 2P imaging) when showing calcium traces.

9) In Fig 5, the authors sought to state that DD was largest in RPB neurons who displayed offset responses and upon oddball stimulation of their BF. I think it has to be clearly put that BF of RPB neurons were determined using 2P calcium imaging upon pure-tone paradigm (Ext Fig 3a-3d, Ext Fig 5c), in which the tone duration and inter-stimulus interval were longer compared to oddball and many-standards paradigms. The onset/offset responses may not be obvious upon oddball and many-standards paradigms of shorter tone duration.

10) The authors found that DD in the core was reduced when RPB was inactivated by injecting muscimol there (Fig 7a-7d). This is a significant finding suggesting that RPB is necessary for DD at the core. In Fig 7b, there may be missing black lines for indicating DD. The description in line 244-245 is confusing; it seems that this result is derived according Fig 7d and it would be helpful that the authors can provide more detailed description. It is however still not clear whether the reduced DD at the core was caused by reduced responses to oddball tone or enhanced responses to many-standards tone. Since inactivating RPB by using muscimol injection could be non-specific, would the effect of reduced DD be still present when using more specific manipulation approach, e.g. optogenetic inhibition, for inactivating RPB?

11) The authors simulated oddball stimulation by presenting a sequence of standard tones and applied optogenetic activation in RPB right after tone stimulation for 10% of the standard tones. The authors found enhanced late responses (after ~250 ms from tone onset) to stimulation of simulated oddball stimulus, resembling DD (Fig 7e-7f, Ext Fig 8e). This is a very interesting finding. It is however not clear how this can be linked to the generation of DD at the core. I think it would be helpful for understanding how RPB activation may contribute to DD generation at the core by plotting the trace of photostimulation only in Fig 7e.

Here are some minor comments:

1) The scale bars of y-axis are missing in Fig 1f, Fig 2b, Extended Fig 3a.

2) The reference for different response types measured using electrophysiological recording in mice A1 in line 166 is missing.

- 3) It's not clear to me the difference between Fig 4b-4d vs Fig 4e.
- 4) In Ext Fig 4a, for the first neuron (top) at the core, the tone-evoked response to standard tones were weak. Are the 2 neurons in Ext Fig 4a both showing DD? If so, it would be helpful to provide the numbers of neurons, showing and not showing DD, in each auditory cortical area; the authors can consider to put line 714-717 in Results.
- 5) It is not clear whether the responses shown in Fig 5 were obtained using 1P or 2P calcium imaging (I suppose 2P imaging).
- 6) It is not clear what are the grey and pink shades in Fig 7e.

We thank all the reviewers for their careful consideration of our manuscript and for making helpful comments. Our detailed responses (in black) to the reviewers' comments (in blue) are provided below.

Reviewer #1 (Remarks to the Author):

The paper is generally well written, and on the whole has the beginnings of an interesting argument. However, there are some critical problems with the experimental design and analysis, and these require major adjustments to be addressed.

1. I found surprising that not adaptation for the standard tone was found in the core (P7) even more considering the deviant and standard tone were the same frequency. The paper does not discuss the adaptation that is occurring in repetitive (standard) signals, and assumes that the responses to all presented standards are the same. This is not the case, and would need a complete reanalysis. Without a careful examination of what adaptation is and is not occurring, it is impossible for the authors to rule out that all of their results are simply the result of adaptation.

First, we backed up the calcium response data with electrical response data. We re-analyzed the LFP response in the core to the 50-ms standard tone in the oddball paradigm and to the 50-ms tone in the many-standards control paradigm (corresponding to Supplementary Fig. 2d). They were not significantly different (Response Fig. 1a).

Response Figure 1. Averaged LFP responses in the core to the 50-ms tone in the oddball paradigm (orange) and 50-ms tone in the many-standards control paradigm (blue) ($n = 13$ penetrations from the three animals that were used in Fig. 2f). There was not any time point

at which the response amplitude was significantly different ($P < 0.05$, Wilcoxon rank sum test, FDR-adjusted). The averaged amplitudes during 0–200 ms or 200–400 ms after the tone onset were also not significantly different (0–200 ms, $P = 0.38$; 200–400 ms, $P = 0.23$, t -test). **b**, The left-most blue dot shows the averaged LFP amplitude to the 50-ms tone in the many-standards control paradigm in the core over 0–200 ms after the tone onset. The other plots show the averaged LFP amplitudes in the core to the i th 50-ms tone (labeled by S_i , $i = 1, 2$, to 7) after the deviant tone stimulus in the oddball paradigm ($n = 13$ penetrations from the same three animals as in **a**). Vertical lines indicate SEMs. The baseline amplitude was subtracted. **c**, Averaged LFP amplitude (during 0–200 ms after the tone onset) in the standard paradigm with SOA of 550 ms (orange) or 350 ms (green). $n = 18$ and 16 sessions, respectively, from two animals. SOA 550 ms: $n = 2$ at 1 kHz, 3 at 2 kHz, 3 at 4 kHz, 6 at 8 kHz, and 4 at 16 kHz. SOA 350 ms: $n = 2$ at 1 kHz, 3 at 2 kHz, 2 at 4 kHz, 5 at 8 kHz, and 4 at 16 kHz. * $P < 0.05$, the first tone vs. the second and subsequent tones (Wilcoxon signed-rank test, FDR-adjusted).

Next, we examined whether the LFP response to the standard 50-ms tone decreased in the repetition after the deviant tone (corresponding to Supplementary Fig. 2e). The LFP amplitude in the core during 0–200 ms after the tone onset did not apparently decrease as the repetition increased (Response Fig. 1b).

Second, we examined whether the tone response was adapted throughout the oddball paradigm, in which the SOA was 550 ms and the same tone frequency was used. As a new paradigm, we introduced repetitive (10 times) 50-ms standard tones with SOA of 550 ms (a standard-tone block). We repeated the standard-tone block with an inter-block interval of 10 s (standard paradigm). During the standard paradigm, we conducted the LFP recording in the core. If the response was to show adaptation in the first few trials in the oddball paradigm of each session and then remain at a steady-state low level, the response to the first tone in the standard-tone block would be higher than the responses to the following tones in the block. However, they were not significantly different (orange line in Response Fig. 1c).

Another possibility is that the adaptation threshold in the marmoset is high compared with that in other species, and that adaptation did not occur under our current condition. To test this, we shortened the SOA from 550 ms to 350 ms in the standard paradigm to make adaptation by repeated stimulation more likely to occur. In fact, in papers that studied stimulus-specific adaptation in the auditory cortex of rodents, an interstimulus interval or SOA of 300 ms was frequently used (Natan et al., eLife 4, e09868, 2015; Nieto-Diego and Malmierca, PLoS. Biol. 14, e1002397, 2016). Shortening of the SOA decreased

the core responses to the second and subsequent tones (green line in Response Fig. 1c). Thus, the reason why the core response adaptation was not apparent in the current study was because the stimulation condition was too weak to induce it in the awake marmoset. We have added these results to Fig. 2g–i.

By contrast, we clearly detected rapid attenuation in the RPB responses in the standard paradigm. We describe the results of the additional experiments and their interpretation on page 20 in our response to a minor comment of Reviewer #1 (170: This is probably adaptation occurring...).

2. Although the paper does briefly touch on the need for electrophysiological recordings, it does not discuss one of the primary reasons for this: temporal resolution. At no point is the temporal resolution of the imaging setup so much as mentioned. Buried in the methodology, there is a single reference to the imaging being viewed at 30Hz. However, it is extremely unlikely that the imaging is being recorded at 30Hz - the best temporal resolution I have seen in papers using this type of imaging talk about 60ms between images. That is quite a bit slower than 30Hz, and further is a much larger interval than both the differences between different stimuli lengths and the length of the shortest stimulus presented. As such, the findings presented are unconvincing. They honestly need to be backed up with electrophysiology across the board, and not a single experiment where the researchers did an electrophysiological recording, got similar results to their imaging, and called it a day. Fine temporal changes in stimuli require fine temporal measurements.

Two-photon microscopy with a combination of the Galvano scanner on the slow axis and a resonant scanner on the fast axis achieves a frame rate of 30 Hz (Ji et al., Nat. Neurosci. 19, 1154–1164, 2016). This frame rate was adopted by some two-photon microscopy systems that are commercially available (Olympus, Sadakane et al., Cell Rep. 13, 1989–1999, 2015; Scientifica, Currie et al., Cell Rep. 39, 110801, 2022; Thorlabs, Liu et al., Cell 185, 3408–3425.e29, 2022; Sutter Instruments, Musall et al., Nat. Neurosci. 22, 1677–1686, 2019). In one-photon calcium imaging of the mouse cortex, a frame rate of 30 Hz is frequently used

(Makino et al., *Neuron* 94, 880–890.e8, 2017; Musall et al., *Nat. Neurosci.* 22, 1677–1686, 2019). In our previous study on behaving head-fixed marmoset, two-photon calcium imaging of the motor cortex was also conducted at a frame rate of 30 Hz (Ebina et al., *Nat. Commun.* 9, 1879, 2018).

As the reviewer pointed out, the temporal resolution of calcium imaging is lower than that of electrical recording. If action potentials occur at intervals shorter than the frame rate, temporal variations in individual neural activity cannot be detected. However, the number (or occurrence probability) of action potentials within a given time period can be estimated because calcium responses have a decay and are accumulated as the number of action potentials increases (Chen et al., *Nature* 499, 295–300, 2013; Pnevmatikakis et al., *Neuron* 89, 285–299, 2016). In the current study, we used a genetically-encoded calcium indicator with fast kinetics (GCaMP6f; Chen et al., 2013) that we previously estimated to have a half-decay time of 0.31 s in the marmoset cortex (Sadakane et al., 2015). This value is comparable with that of the original report for GCaMP6f (~0.2 s; Chen et al. *Nature* 499, 295–300, 2013). The time window to detect deviance detection in electrical recording is frequently set to ~100–200 ms after the tone onset (Koshiyama et al., *Schizophr. Bull.* 46, 937–946, 2020). In the LFP recording in the present study, the deviance detection was detected within 200 ms after the onset (Fig. 2f and 3g). Considering these results, we think that it is rational that the significant deviance detection of the calcium response was detected in the time window 100–400 ms after the tone onset. In another study in which calcium imaging was used to measure visual mismatch negativity in the mouse visual cortex, the time window to show the mismatch negativity was longer in calcium imaging than in electrical recording (Fig. 1E and 1I in Hamm and Yuste, *Cell Rep.* 15, 597–604, 2016). We have also re-examined the extent of the adaptation in the core and RPB by performing LFP recording with a high temporal resolution, as described on pages 1 and 20. However, we agree that electrical recording is necessary to reveal how the deviance detection is generated with a fine-temporal resolution. We have now mentioned this limitation to our method in the Discussion section (lines 397–401).

3. Authors observed that offset neurons have the strongest deviant detection signal (DD) and that this DD was earlier and stronger in the RPB area (figure 4). I think it is important to quantify how was the proportion and latencies of those responses across core, belt and parabelt areas and whether the offset magnitude correlate with the tone duration during the equal probable many-standard protocol. Is the DD stronger in the belt areas because they have more offset neurons? I was wandering whether offset neurons are the detectors for duration deviants or whether a similar trend across neuron types and areas would be expected for frequency deviants as well.

We show the proportions of the four types of responses (onset+ offset-, onset+ offset+, onset- offset+, and onset- offset-) in Supplementary Table 2. We have added the latency (we defined this as the time from the tone onset to the point when the response exceeded half of the maximal amplitude in the pure-tone paradigm) to Supplementary Table 2. The proportion or latency of offset+ neurons in the RPB was not substantially different from that in the core. Even when the analysis was limited to those neurons whose BF was F_{odd} , the proportions of offset+ neurons were 28% (71/251) in the core, 18% (20/112) in ML/CL, and 31% (21/68) in RPB. Thus, the proportion was not substantially different between the core and RPB. Although the proportions between ML/CL and RPB were different (18% vs, 31%), the numbers of offset+ neurons were similar (20 vs. 21), and therefore the statistical power should be similar for these neurons when they are examined in Fig. 5b and Supplementary Fig. 5d. The results suggest that the offset+ neurons showed higher deviance detection in the RPB than in ML/CL, although we do not exclude the possibility that ML/CL offset+ neurons contributed to the generation of deviance detection.

Significant deviance detection was first detected at the 100-ms point after the tone onset in RPB offset- neurons (Fig. 5c). It was secondly detected at a 166-ms point after the tone onset in the core offset- neurons (Supplementary Fig. 5e). We have added a graph that shows that early deviance detection amplitude (averaged over 100–167 ms after the tone onset) of offset- neurons in the core and RPB (Fig. 5d). Thus, the deviance detection might be first detected in RPB offset- neurons, and then might be amplified by RPB offset+

neurons, rather than being first generated by RPB offset+ neurons. However, as the reviewer pointed out, determining the time order to a resolution of around 10 ms is difficult with calcium imaging, and the limited number of neurons detected makes it difficult to draw clear conclusions. We have added a summary schematic of the possible activity flows of RPB neurons with and without offset responses, and added the following sentence to the Discussion section (lines 398–401):

“It is therefore necessary to conduct electrical recording to obtain a fine-scale time sequence to reveal how the onset and offset responses interact with each other to detect a deviant tone and transmit the error signal across the core, belt, and parabelt”.

When we examined the amplitude of the response to each tone duration over 100–300 ms after the tone end in the many-standards control paradigm, the amplitude in offset+ neurons in the core increased as the tone duration increased (Response Fig. 2). By contrast, in the RPB, it was high with the 10-ms tone and rapidly increased with tone duration from ~150 ms to 225 ms (Response Fig. 2). The large response to the 10-ms tone might be related to detection of the tone omission because the current tone stimulation condition was weak for the marmoset, as described in our response to the first comment. We have added these results to Fig. 5e–h. The dependency of the response on the tone duration in ML/CL appeared to be intermediate between those in the core and RPB, and a dependency in CPB neurons was not apparent. These results suggest that the offset+ neurons in the RPB had different properties for detecting the tone duration to those in other auditory cortical areas. On the basis of the present study, the characteristics of cellular activity in each region should be investigated in more detail, as in the visual cortex. We believe that such work is beyond the present study, and that the present study will act to boost future research.

Response Figure 2. Averaged response amplitudes (100–300 ms after the tone end) of the offset+ neurons whose BF was F_{odd} to each of the 10–225-ms tones in the many-standards control paradigm and to the 100-ms and 50-ms tones in the oddball paradigm in each region

(one-sample *t*-test, FDR-adjusted, **P* < 0.05, · *P* < 0.01, ·· *P* < 0.001, *n* = 71 for core, *n* = 20 for ML/CL, *n* = 21 for RPB, *n* = 8 for CPB).

4. I think authors need to consider other scenarios that they are missing due to technical limitations. In the predictive coding literature usually, a canonical model is proposed in which the feedback signals are sent from the deep to superficial layers and an extended circuit model in which in addition there is superficial-to-superficial layer feedback projections. Authors mentioned that the majority of the 2-photon imaging and dMMN paradigm were done in the L2/L3 layers which is understood due to the calcium imaging technique. So, it would be welcome authors discuss other possible signal routing that could be occurring during deviance signaling. Is it possible that the lack of DD signal in the CPB is due to the fact that in higher order areas the feedback signaling arise from deep layers rather than superficial?

We do not exclude the possibility that the deep-layer neurons send the feedback error signal to the core. Even in the RPB, the deep layer neurons might show stronger error signals than L2/3 neurons. The calcium responses of axons extending from RPB to the core and the effects of photoactivation of RPB and RPB muscimol injection on the core response might originate from the deep-layer neurons in the RPB, rather than the RPB L2/3 neurons. Muscimol was injected into the CPB at a depth of 1 mm from the cortical surface so that the deep layer should also be inhibited. Thus, we consider that the CPB was not necessary for the induction of deviance detection. We have added the following sentence to the Discussion section (lines 385–387):

“As the feedback projections originated mainly from layer 5 in the auditory cortex (Galaburda et al., *J. Comp. Neurol.* 184, 169–184, 1983), layer 5 neurons downstream of L2/3 neurons in RPB may broadly back-project to L1 of the core and lateral belt”.

We have also discussed possible different functions between the RPB and CPB as follows (lines 408–412):

“It is assumed that the rostral auditory pathway carries information on what the sound is and what the origin of the sound is, and that the caudal (dorsal) pathway carries information on where the sound comes from (Rauschecker and Tian, *PNAS* 97, 11800–11806, 2000;

Bizley and Cohen, *Nat. Rev. Neurosci.* 14, 693–707, 2013). RPB may detect what the deviant sound is and what causes the deviation, whereas CPB may detect the movement of sound sources”.

5. There is evidence (Wang, X.-J. (2022). *Theory of the Multiregional Neocortex: Large-Scale Neural Dynamics and Distributed Cognition. Annual Review of Neuroscience* 45, 533-560.) that indicates that temporal integration of neurons increases for areas of higher hierarchical order. Is it possible that if testing shorter durations like 25 vs 75 ms, the DD in the core area would be enhanced since that durations better match their time windows.

We agree that it is important to determine the temporal range required to induce the deviance detection by changing the duration of the standard and deviant tones. However, this should also depend on the tone frequency, intensity, duration, and SOA. We think that such comparative measurements are beyond this study, the main aim of which is to show the occurrence of feedback from the RPB to the core following change detection. As shown in our response to the third comment, we detected an increase in the response to the 10-ms tone in the RPB offset+ neurons in the many-standards control paradigm. This suggests that even when the deviant tone is shorter than the standard tone, deviance detection will still occur in the RPB.

6. It is clear the changes in the latencies and magnitudes of the DD signals across areas. Do the dynamics of the DD (duration, latency) follow the same as previously described for frequency deviants in the rat or macaque model, for example? It would be nice to include some comparison of the neural dynamics across areas, sound features and animal models since few studies performed multisite recording with the same paradigm.

We have added the following discussion to line 358:

“When LFP was recorded in the core, the significant deviance detection appeared around 200 ms after the tone onset and was still detected after 400 ms (Fig. 2f). In the RPB, the

significant deviance detection appeared around 120 ms after the tone onset and was still detected at around 400 ms (Fig. 3g). In a similar oddball paradigm for duration MMN in human and macaque (50 and 100 ms for the standard and deviant tones, respectively, and 500 ms for the SOA), the latency of the dMMN is around 130 ms after the tone onset and the dMMN is still detected after 200 ms^{12,31}. Thus, the time course appears to be similar among these primate species. However, in the dMMN paradigm, the differences in the responses between different auditory cortical areas have not been examined. In the frequency MMN paradigm, deviance detection was measured in rat A1 and higher-order auditory cortex with LFP recording²². The deviance detection amplitude was larger in higher-order auditory cortex than in A1, and significant deviance detection was detected for 60–100 ms after the tone onset in A1, and for 25–200 ms in higher-order auditory cortex. As the deviant tone can be detected as the deviance immediately after the tone onset in the frequency MMN paradigm, the time window would be shifted to earlier than that in the current study. The difference in the time window length might reflect the difference in the size or structure of the auditory cortex between the rat and marmoset. In both the dMMN paradigm for the marmoset and the frequency MMN paradigm for the rat, the latency was earlier in the higher-order auditory cortex than in A1. Thus, in both rodents and primates, the higher-order auditory cortex may rapidly detect the prediction error signal and then feed it back to A1. In addition, in the frequency MMN paradigm of the rat, the medial prefrontal cortex neurons respond to the deviant tone with a latency of 120–180 ms from the tone onset in the frequency MMN paradigm⁴⁷. In the marmoset, the pure deviance detection that is embodied in the rodent medial prefrontal cortex may be generated in the RPB, part of the higher-order auditory cortex, which may reflect more elaborate cortical sensory processing than in rodents”.

7. It would be important to include a summary figure that contrast the temporal dynamics of the feedforward and feedback routing, i.e., how the neural activation occurs across areas and layer (1 vs 2/3) when the sound duration is equal-probable, repetitive and deviant. Maybe using a colormap for tracing the trajectory of the activation appearing first in core and then

in the belt when equal-probable and then it switching to the RPB when deviant.

We have added a summary figure as Fig. 8, as follows:

This is a schematic of the possible activity flows related to the deviance detection generation in the auditory cortex. We propose that the core receives the sensory stimulus as it is; RPB, which is highly adapted to the tone repetition, detects the deviant tone stimulus and the error signal in the RPB propagates backwards (red) to the core directly or through the belt and non-lemniscal thalamus; and the response to the deviant tone is enhanced in the core. It is unclear how the auditory cortical circuit determines whether sounds of the same nature are being repeated.

8. In previous studies it has been found that the tuning width and DD magnitude are correlated for other sound features such as frequency. In methods it was indicated that a tuning width index was estimated. It would be interesting to know whether neurons with larger receptive fields also had stronger dMMN in each area?

We estimated the relationship between the amplitude of the deviance detection and the selectivity index of the sound frequency (SI). Deviance detection was strongly detected in

the offset+ neurons whose BF was F_{odd} (Fig. 5b and Supplementary Fig. 5g), so we therefore examined the relationship in these neurons. Among these neurons in the RPB, but not in the core or belt, the amplitude of the deviance detection and SI were negatively correlated (Response Fig. 3). This suggests that in RPB offset+ neurons, if the best frequency was used for the oddball paradigm and the tuning frequency range was narrow, the amplitude of the deviance detection was higher. We have added this result to Supplementary Fig. 5h.

Response Figure 3. The relationship between the mean amplitude of the deviance detection (averaged over 200–400 ms after the tone onset) and SI in offset+ neurons whose BF was F_{odd} . Pearson correlation coefficients are shown below, core, $r = -0.23$, $P = 0.06$. ML/CL, $r = -0.10$, $P = 0.67$. RPB, $r = -0.54$, $P = 0.01$.

9. I am concerned regarding any contamination due to the fixed SOA on the magnitude of the DD or standard response. Why to use a fixed SOA = 550 rather than a fixed ISI? How was the variability of the activation across ISIs of different length?

In the duration mismatch negativity paradigm, SOA is more common than ISI (for example, SOA was used for humans, Koshiyama et al, Schizophr. Res. 195, 387–384, 2018; macaque, Tada et al., Front. Psychiatry 11, 874, 2020; and rat, Nakamura et al., Front. Psychol. 2, 367, 2011). As shown in Supplementary Fig. 2c, the response after a short ISI (that is, a long-duration tone in the previous tone) was slightly contaminated by the decayed response to the previous tone. Brown and green traces represent responses subject to the large decay of calcium responses in the previous tone stimulus and the fluorescence signal averaged for 150 ms before the tone onset being set to zero; these traces for 100 ms before the tone onset

were slightly different from those represented by the orange and blue traces. However, the large response to the deviant tone was apparent compared with the other three responses. In the calculation of the amplitude in the many-standards control protocol, we used only those trials whose previous trials had a short tone duration (10, 25, and 50 ms) to eliminate the effect of the remaining activity after the long-tone presentation.

10. It was not clear for me at which intensity the tone was delivered. It is important since looking at the marmoset audiogram there are clear difference all along the tested tones (1,2,4,8,16 kHz).

The tone stimuli were presented through a headphone speaker placed 10 cm away from the contralateral ear of the imaging side. The intensity of the tone stimuli was calibrated using a sound level measurement amplifier around the ear position of the animal, and the difference from the microscope noise was kept at 60 dB through the use of a sound-proof board. In a frequency MMN experiment for the marmoset, one of the coauthors used 60-dB sound pressure level (SPL) around the animal's ear (Komatsu et al., *Sci. Rep.* 5, 15006, 2015). The 60-dB intensity is within the range used in other studies on auditory responses of the marmoset (30–90-dB SPL in Zeng et al., *PNAS* 116, 3239–3244, 2019; 60–70-dB SPL in Nishimura et al., *Brain Struct. Funct.* 223, 1599–1614, 2017; 80-dB SPL in Tani et al., *eNeuro* 5, ENEURO.0078–18.2018, 2018).

As the reviewer pointed out, the hearing threshold in the marmoset is similar between 1, 2, 4, and 16 kHz, but is lower at 8 kHz than at the other frequencies (Osmanski and Wang, *Hear. Res.* 277, 127–133, 2011). However, in the current study, when comparing the difference in the response between the oddball and many-standards control paradigms, we used the same frequency in both paradigms. Thus, the differences in the tone responsiveness between different tone frequencies should not affect the results. We used the same tone intensity over all frequencies (60 dB). The amplitude (averaged over 0–200 ms after the tone onset) of the core response to the 100-ms tone in the many-standards control paradigm was detected with all five frequencies, including 8 kHz (Response Fig. 4). We

have added this result to Supplementary Fig. 2g. In addition, the deviance detection (averaged over 200–400 ms after the tone onset) was detected with all five frequencies, as described in the response to Reviewer #1’s comment 12.

Response Figure 4. The mean amplitude (averaged over 0–200 ms after the tone onset) of the core one-photon response to the 100-ms tone in the many-standards control paradigm with each tone frequency. There was no significant difference among the five frequencies, unpaired *t*-test with Bonferroni correction ($n = 25$; 5 for 1 kHz, 9 for 2 kHz, 4 for 4 kHz, 4 for 8 kHz, and 3 for 16 kHz, from three animals).

11. Authors found that the DD mainly occurs in the region where the BF was the deviant/standard stimulus (P8). It is interesting to know whether deviance detection for duration has been done using broad-band noise rather than pure tones for example? Would authors expect a similar trend in the origin and propagation of the feedback error signal?

We agree that it is important to examine how different types of tone induce deviance detection. However, broad-band noise is not commonly used in MMN research, and we think that such an examination is beyond the current study. We would like to approach this in the future.

12. Was the DD signal similar across tone frequencies or there was a bias to any frequency for duration deviance detection?

The amplitude of the deviance detection did not differ among the tone frequencies in the core (Response Fig. 5a). In RPB, it tended to be larger for the high frequency tone than for the low frequency tone, but it was not significantly different among the four frequencies (the number of sessions for 4 kHz frequency was small, and it was therefore combined with

that for 2 kHz) (Response Fig. 5b). We have added these results to Supplementary Fig. 2f and Supplementary Fig. 3e.

Response Figure 5. The mean amplitude of the deviance detection for each tone frequency used for the oddball paradigm (averaged over 200–400 ms after the tone onset) of one-photon responses in the core (**a**) and RPB (**b**). There was no significant difference among the frequencies (unpaired *t*-test with Bonferroni correction, $n = 25$; 5 for 1 kHz, 9 for 2 kHz, 4 for 4 kHz, 4 for 8 kHz, and 3 for 16 kHz, from three animals in **a**, and $n = 25$; 6 for 1 kHz, 5 for 2 kHz, 2 for 4 kHz, 4 for 8 kHz, and 8 for 16 kHz, from two animals in **b**). In **b**, the data at 2 and 4 kHz are pooled.

13. Regarding the activation and deactivation experiments. I have some questions on that:
 - Authors injected AAV into the RPB and found that RPB project to L1 in core and belt. Can also authors indicate in which RPB depth the injections were done? Which was the main projection layer authors were manipulating?

In the Methods section, we describe that the pipette was inserted vertically approximately 500 and 1000 μm ventral from the brain surface (lines 681, 686, and 691). Thus, we consider that projecting neurons in both layers 2/3 and 5 were manipulated to a similar extent.

- Why authors waited 2 hours after the muscimol injection to perform the image acquisition?
 How do authors control for muscimol spread across areas along time?

This experiment was based on the fact that marmosets were unable to perform forelimb movements 2 hours after injection of muscimol into the forelimb motor cortex under nearly identical conditions in another experiment using our marmosets. The effect on the forelimb

movement disappeared 6 hours after the injection (Ebina et al., unpublished data). In an experiment with the macaque (Miyamoto et al., *Science* 355, 188–193, 2017), the recording was started 30–90 minutes after muscimol injection into A8/9. In an experiment with the owl monkey (Stepniewska et al., *J. Neurophysiol.* 111, 1100–1119, 2014), the recording was conducted within 4 hours after muscimol injection into certain cortical areas. Thus, we think that it is reasonable to record 2 hours after the injection.

To estimate the spread of muscimol, as described in our previous study on the mouse (Terada et al., *Cell Rep.* 41, 111494, 2022), a small volume of muscimol BODIPY TMR-X conjugate (120 nl; 5 $\mu\text{g}/\mu\text{l}$) was injected into the RPB under the same conditions as used in the pharmacological inactivation experiment. We conducted two-photon imaging for 3 hours after the injection. The full width at half maximum (FWHM) of the fluorescence intensity on a line perpendicular to the putative border between the RPB and AL was 1.1–1.2 mm (Response Fig. 6).

Response Figure 6. Fluorescence of BODIPY conjugated muscimol solution injected into RPB. **a**, The red dot indicates the injection site. The example image was acquired 2 hours after the injection. **b**, Fluorescence profiles along the line crossing near the injection center (yellow line in **a**). The FWHM of the intensity profile at each timepoint is indicated in parentheses.

Since the FWHM remained almost unchanged over time, we assume that some of the injected molecule was broadly diffused from the insertion site of the glass pipette. Since the molecular weight and hydrophilicity of the muscimol BODIPY (607) and muscimol (114) differed, and these diffused in the intercellular space in the cerebral cortex, the spread of muscimol could not be accurately estimated in this experiment. However, given that the diffusion constant in aqueous solution is inversely proportional to the square root of the molecular weight, and that the diffusion distance is proportional to the square root of the diffusion constant (Terada et al., 2022), we supposed that the spread of muscimol was

approximately 1.5 times that of muscimol BODIPY, and the distance from the injection point to the point with the half maximum of the intensity would therefore still be less than 1 mm. The distance between the injection site and the putative border between the RPB and AL was more than 1 mm in all muscimol injection experiments. Thus, we consider the muscimol effect to be mainly due to inactivation of the RPB, but not the AL or core. We have added these results to Supplementary Fig. 8a and discussed the muscimol spreading in the Methods section (lines 789–801).

- Why 10 and 45 frames were used for estimating baseline and evoked activity?

Ten and 45 frames correspond to 333 ms and 1.5 s, respectively. These were arbitrarily determined. The duration (500 ms) and SOA (3 s) in the pure-tone paradigm was longer than those (5 and 15 frames, respectively) in the oddball and many-standards control paradigms. The use of a large number of frames was intended to reduce variation per timepoint by allowing averaging over long time windows.

- In methods when referring to time points I suggest to indicated also their equivalence in ms.

We have added the equivalence in ms or second whenever we have referred to time points in the Methods sections.

Other minor issues

There is incredibly heavy use of abbreviations and acronyms in this paper. It absolutely needs to have a reference table for them all. Further, when possible, some of the abbreviations/acronyms should be avoided or spelled out. The constant use of them makes the paper more difficult to read.

We have added a table of abbreviations as Supplementary Table 1. We have removed the

following abbreviations: DD, RES, STG, V1, mPFC, EEG, ECoG, and fMRI.

56: Repetitive standard stimuli induce adaptation. It needs to be discussed. It's entirely possible all results seen are due to adaptation unless it is accounted for.

We have rewritten the corresponding sentences as follows:

“To search for potential evidence of such error feedback, we focused on auditory mismatch negativity (MMN; abbreviations are listed in Supplementary Table 1), an event-related potential that represents the difference between “rare deviant stimuli” (with deviant frequency, duration, or intensity) and “repetitive standard stimuli”. MMN includes two components: adaptation to frequently presented stimuli and deviance detection in response to infrequent stimuli^{6–12}”.

As described on pages 1 and 20, we have added electrical recording results and discussions on adaptation in the core and RPB.

61. Missing reference

We have now included Naatanen et al. (Clin. Neurophysiol. 118. 2544–2590, 2007) and Chao et al. (Neuron 100, 1252–1266.e3, 2018).

85 - 87: This sentence is unclear. Pretty sure macaques and humans have all three regions of the superior temporal gyrus as well.

We have rewritten this sentence as follows (lines 85–87):

“In macaques and humans, most of the core is buried in the lateral sulcus, but this is not the case in marmosets^{7,23,24}, allowing the activity of the core, lateral belt, and parabelt to be imaged by fluorescence microscopy through a cranial window^{25–28}”.

126: Discussion of the calcium responses to 100 and 50 ms stimuli does not account for

adaptation in the presented standard.

We have clearly described the electrical recording results regarding the adaptation in the core, and discussed their interpretation in the Results section as follows (lines 158–169):

“However, in almost all MMN studies, sensory-specific adaptation was observed in A1^{18,21,22,35}. Thus, we further examined the cause of this discrepancy. We introduced repetitive (10 times) 50-ms standard tones with SOA of 550 ms (a standard-tone block). We repeated standard-tone blocks with an inter-block interval of 10 s (standard paradigm) while conducting LFP recording. If the response was suppressed in the first few trials of each session of the oddball and many-standards control paradigms and a low steady-state response remained in the subsequent trials, the response to the first tone in the standard-tone block would be higher than the responses to the following tones in the same block. However, such attenuation was not prominent (Fig. 2i). Then, to examine whether the SOA of 550 ms was too long to induce adaptation in the marmoset core, we shortened the SOA to 350 ms in the standard paradigm. This change clearly induced response attenuation after the second tone presentations (Fig. 2i). Thus, the adaptation threshold in the marmoset core might be higher than that in other examined species”.

140: What kind of electrical recording? surface electrodes on the cranium? It needs to be specified.

We have rewritten the sentence as follows (lines 148–150):

“We also inserted a microelectrode into the superficial layer (< 300 μm from the cortical surface) to record the local field potential (LFP) in a site where the BF determined by one-photon imaging was F_{odd} ”.

We also describe the methods in detail in the Methods section, as follows (lines 758–764):

“The LFP was recorded in the imaging areas of the three marmosets using a tungsten microelectrode (500 $\text{k}\Omega$; World Precision Instruments). Before the extracellular recording,

the glass coverslip of the imaging window was replaced with a silicone sheet (6-9085-12; AS-ONE Corporation, Osaka, Japan). The electrode was inserted perpendicular to the cortical surface through the silicone sheet using a micromanipulator (Narishige), and the recording sites were less than 300 μm from the *cortical surface*. During the recording, silicone elastomer (Kwik-Sil; World Precision Instruments) was filled into the window to suppress brain pulsation”.

153: RPB is not defined. I assume it means Rostral Parabelt, but I could be wrong and the paper cannot rely on the reader to know what it means. Please never use an acronym without first defining it.

We defined RPB in the abstract, but not in the main text. We have defined it again in the Results section.

170: This is probably adaptation occurring - the discussion of control stimulus responses being weak without discussing adaptation leaves the conclusion completely unsupported.

When we conducted LFP recording of the RPB in the standard paradigm in which the tone duration was 50 ms and SOA was 550 ms, response attenuation rapidly occurred (Response Fig. 7a). This was largely different from the non-adaptive response in the core, as shown in Response Fig. 1c. Furthermore, we found that the RPB response to the 100-ms tone during the many-standards control paradigm adapted to a greater extent than the RPB response to the 100-ms tone in the absence of the preceding tone presentation (Response Fig. 7b). In both LFP recording and calcium imaging, the amplitude of the response to the 50-ms tone (averaged over 0–200 ms after the tone onset) did not differ between the oddball and many-standards control paradigms (Response Fig. 7c, d). These results indicate that when a tone with the same tone frequency was repeated, whether or not the duration of the tone was constant, the RPB response rapidly adapted and remained at a low steady-state level. We have added these results to Fig. 3h and Supplementary Fig. 3g–i.

Response Figure 7. **a**, Averaged amplitude of LFP (during 0–200 ms after the tone onset) from the RPB in the standard paradigm with SOA of 550 ms (orange) or 350 ms (green). $n = 15$ sessions from two animals that were the same as those in Response Fig. 1c. $*P < 0.05$, the first tone vs. the second and subsequent tones (Wilcoxon signed-rank test, FDR-adjusted, SOA 550 ms: $n = 15$; 4 at 2 kHz, 4 at 4 kHz, 6 at 8 kHz, and 2 at 16 kHz from two marmosets. SOA 350 ms: $n = 15$, 4 at 2 kHz, 4 at 4 kHz, 6 at 8 kHz, and 2 at 16 kHz). The animals were the same as those used in Response Fig. 1c. **b**, Averaged amplitude of LFP (during 0–200 ms after the tone onset) from the RPB in the many-standards control paradigm in which the duration of the first tone was 100 ms. The left column is the response to the first 100-ms tone and the right is the average of the response to the 100-ms tones during the paradigm. $P < 0.001$, Wilcoxon signed-rank test ($n = 50$ sessions from one of those used in **a**; $n = 18$ at 1 kHz, 17 at 2 kHz, and 15 at 4 kHz). **c**, Averaged LFP amplitude to the 50-ms tone in oddball (orange) and many-standards control (blue) paradigms in the RPB ($n = 9$ from one animal that was used in Fig. 3g). The amplitude averaged during 0–200 ms after the tone onset was not significantly different ($P = 0.37$, t -test). **d**, Averaged amplitude of the calcium response to the 50-ms tone in the oddball and many-standards control paradigm in the RPB ($n = 25$ from two animals that were used in Fig. 3c). The amplitude averaged during 0–200 ms after the tone onset was not significantly different ($P = 0.13$, t -test).

If the response to the 100-ms tone was large in the oddball paradigm because there were neurons that specifically responded to the 100-ms tone but not to any of the other tones, and they were non-adapted during the repetition of 50-ms tones, then the amplitude of the response should be similar between the oddball and many-standard control paradigms because the presentation probability of the 100-ms tone was 0.10 in both paradigms and the 100-ms tone-specific neurons should not be adapted in either paradigm. However, the response to the 100-ms tone was lower in the many-standards control paradigm than in the oddball paradigm. Thus, it is rational to assume that the response to the 100-ms tone in the oddball paradigm was larger because the auditory cortex predicted the repetition of the 50-ms tones and detected the 100-ms tone as deviant (prediction error). Consistent with this hypothesis, in the many-standards control paradigm, the responses of offset+ neurons were large for the tones with 10, 200, and 225 ms, which deviated considerably from the average

sound duration (~100 ms) (Response Fig. 2). Thus, we speculate that the RPB activity is attenuated during predictable stimuli, and that when an unpredictable stimulus is detected, the attenuation is removed by disinhibition, and/or strong excitatory inputs are newly generated, and the RPB is able to respond, sending error signals back to the core. This signal may also be sent to some higher-order brain areas including PFC, which may trigger attention and some cognitive process. We have added these results to the Results section as follows (lines 212–220):

“In contrast to the LFP response in the core, the LFP response in the RPB rapidly attenuated in the standard paradigm with a tone duration of 50 ms and SOA of 550 ms (Fig. 3h). In addition, the response was much reduced during the many-standards control paradigm (Supplementary Fig. 3g). The response to the 50-ms tone was similarly weak in the oddball and many-standards control paradigms in both LFP recording and calcium imaging, and the difference between these paradigms was not significant (Supplementary Fig. 3h,i). These results suggest that the repetitive stimulation with the same tone frequency strongly attenuated the RPB response to a low steady-state level, regardless of the duration of sound presentation, and that there was no apparent duration-specific adaptation as there was in the core, at least under the present stimulus conditions”.

In the rat, mPFC neurons slowly respond to a deviant tone with a latency of 120–180 ms from the tone onset in the frequency MMN paradigm, and the mPFC more rapidly shows attenuation to repetitive tones than does the auditory cortex (Casado-Roman et al., *PLoS Biol.* 18, e3001019, 2020). Thus, mPFC in the rat shows pure deviance detection, which is similar to the RPB in the marmoset. We have cited this paper and discussed how the pure prediction error might be generated in the Discussion section as follows (lines 335–349):

“The deviance detection in RPB neurons with BF of F_{odd} and offset responses was prominent, and its amplitude was much larger than that of the response to 100-ms tones in the many-standards control paradigm, even though the presentation probability of the 100-ms tone was the same (0.1) in both paradigms. The RPB responses to 50-ms tones in both paradigms were similarly suppressed to a great extent, although the presentation probability

of the 50-ms tone differed substantially between these paradigms (0.9 vs. 0.1). Thus, it is rational to assume that the auditory cortex detected that the 100-ms tone deviated from the repetitive 50-ms tone, and that it was not from random-order tones with 10 different tones (10–225 ms). Consistent with this, the RPB neurons significantly responded to those tones whose durations deviated from the averaged tone duration in the many-standards control paradigms. In the rat, attenuation of the auditory response to the repetitive stimulus is stronger in the higher-order auditory cortex than in A1, and stronger in the medial prefrontal cortex than in the higher-order auditory cortex^{35,47}. Thus, it may be a cross-species commonality that the higher the hierarchy of the cortical areas, the weaker the response to repeated stimuli. However, it remains unknown how the tone duration-independent attenuation occurred in RPB but not in the core, and how the deviant tone resumed the suppressed activity”.

Thanks to the reviewer’s comments, we have revealed the difference in the adaptation property between the core and RPB. We believe that our revised manuscript now gives more fruitful insights into the neuronal mechanism behind the generation of deviance detection in the auditory cortex, with different response properties of subareas.

177: RES was defined once, two pages earlier, and never mentioned again until here. Again, acronyms need to be used sparingly to increase readability, not to obscure the discussion.

We have removed “RES” from the revised manuscript. We have also removed DD, STG, V1, mPFC, EEG, ECoG, and fMRI, and spelled them out in full.

179 - 182: I’m not at all convinced when you have three data points, and are making conclusions on the one data point at the end of your data set. Increased the number randomly presented deviant lengths: 75, 100, 150 are not sufficient by themselves.

We agree with the reviewer’s comment that more data points are needed. We have removed

this experiment because it is not relevant to the main results of the paper.

188: Clearer and faster aren't objective observations. There needs to be statistical analysis to back it up.

We have rewritten this sentence as follows (lines 230–232):

“Deviance detection was clearly detected in both L1 and L2/3. The first timepoint with significant deviance detection was detected 100–200 ms from the tone onset in the neuropils, but was around 300 ms from the tone onset in the L2/3 neuronal somata”.

218: Was there an electrophysiological response? With the low temporal resolution of the imaging, no response doesn't necessarily mean there is no response. Even the imaging resolution is 50ms, which I doubt, neurones could fire 40 times in that window, and not be captured.

As we described in our response to the second major comment, the number (or occurrence probability) of action potentials within a given time period can be estimated in calcium imaging because calcium responses have a decay and accumulate as the number of action potentials increases. However, we cannot exclude the possibility that a low firing rate in dlPFC that could not be detected with our imaging system could have affected the RPB activity. We have therefore weakened the sentence in question as follows (lines 278–280): “These results suggest that the dlPFC was not the origin for the generation of strong deviance detection in the current dMMN paradigm”.

244: largely reduced needs to be backed up with stats!

We used the Wilcoxon rank sum test (Fig. 7a), paired *t*-test (Fig. 7d), and one-sample *t*-test (Supplementary Fig. 8e) to confirm that the RPB inactivation eliminated the induction of statistically significant deviance detection, as shown in the corresponding figure legends.

We have rewritten the corresponding sentence as follows (line 307):

“RPB inactivation eliminated the statistically significant deviance detection”.

245: I like the idea of the many-standard paradigm, but with your limited (again I assume, there has been NO discussion of imaging temporal resolution) temporal resolution, you are very constrained by what you can and cannot observe.

We have added discussion on the limited time resolution of calcium imaging as follows (lines 397–401):

“However, although two-photon calcium imaging has a single-cell resolution and is also used to examine visual MMN^{33,34}, its time resolution is lower than that of electrical recording. It is therefore necessary to conduct electrical recording to obtain a fine-scale time sequence to reveal how the onset and offset responses interact with each other to detect a deviant tone and transmit the error signal across the core, belt, and parabelt”.

279: This argument seems plausible to me.

Thank you for this comment.

291: I am much less convinced by this.

In the paper cited in this discussion, duration MMN was simulated with a shorter duration stimulus as a deviant tone. In addition, this simulation did not explain how the RPB response was attenuated in both oddball and many-standard control paradigms. We have now removed this from the discussion.

Reviewer #2 (Remarks to the Author):

The manuscript by Obara et al discovered that prediction error signal was generated in the rostral parabelt (RPB) of higher-order auditory cortex. Such error signal was manifested as deviance detection (DD) in the neural signal, which was measured in the current study by using one-photon (1P) or two-photon (2P) calcium imaging or extracellular recordings in the auditory cortical areas of core, belt, and parabelt of awake common marmoset, during presentation of auditory oddball paradigm. DD has been reported in previous studies to be observed in a macroscopic ECoG signal as mismatch negativity (MMN) in the frontal and temporal lobes of the brain, which is an important biomarker in neuropsychiatric disorders. Here the authors examined DD in primary and high-order auditory cortical areas upon presentation of auditory duration oddball paradigm. Distinct types of tone responses in single neurons at these auditory cortical areas were characterized according to their response onsets/offsets. A pure error signal, or residual error signal (RES), was for the first time discovered in RPB neurons, especially for RPB neurons of certain response type, when RPB neurons showed very strong response to oddball tone stimulus and very weak response to control tone stimulus. Using pharmacology and optogenetic stimulation, the authors found evidences supporting that the error signal, here DD, was backpropagated from the higher-order auditory area RPB to primary auditory cortex.

I think this work is original and of significance for understanding cortical processing of error signal. The evidence for supporting main findings are clear, although more experiments and analysis could be performed to make the findings more thorough. I find the description and discussion of results are sometimes too concise. It would be helpful if the authors can be more explicit in the results description and elaborate the meaning of results. In my opinion, the manuscript can potentially be considered for publication after major revision.

We appreciate these positive comments. We have added some discussion to connect the current results with previous results (lines 350–381).

My main comments are as follows:

1) I find it surprising that there was no clear simple adaptation of responses to the standard tones in the 1P calcium signal upon duration oddball paradigm stimulation (Fig 2b). What would be the reason for this?

First, we have backed up the calcium response data with electrical response data. We re-analyzed the LFP response in the core to the 50-ms standard tone in the oddball paradigm and to the 50-ms tone in the many-standards control paradigm (corresponding to Supplementary Fig. 2d). They were not significantly different (Response Fig. 1a).

Response Figure 1. Averaged LFP responses in the core to the 50-ms tone in the oddball paradigm (orange) and 50-ms tone in the many-standards control paradigm (blue) ($n = 13$ penetrations from the three animals that were used in Fig. 2f). There was not any time point at which the response amplitude was significantly different ($P < 0.05$, Wilcoxon rank sum test, FDR-adjusted). The averaged amplitudes during 0–200 ms or 200–400 ms after the tone onset were also not significantly different (0–200 ms, $P = 0.38$; 200–400 ms, $P = 0.23$, t -test). **b**, The left-most blue dot shows the averaged LFP amplitude to the 50-ms tone in the many-standards control paradigm in the core during 0–200 ms after the tone onset. In the other plots, the averaged LFP amplitudes in the core to the i th 50-ms tone (labeled as S_i , $i = 1, 2$, to 7) after the deviant tone stimulus in the oddball paradigm are shown ($n = 13$ penetrations from the same three animals as in **a**). Vertical lines indicate SEMs. The baseline amplitude was subtracted. **c**, Averaged LFP amplitude (during 0–200 ms after the tone onset) in the standard paradigm with SOA of 550 ms (orange) or 350 ms (green). $n = 18$ and 16 sessions, respectively, from two animals. SOA 550 ms: $n = 2$ at 1 kHz, 3 at 2 kHz, 3 at 4 kHz, 6 at 8 kHz, and 4 at 16 kHz. SOA 350 ms: $n = 2$ at 1 kHz, 3 at 2 kHz, 2 at 4 kHz, 5 at 8 kHz, and 4 at 16 kHz. * $P < 0.05$, the first tone vs. the second and subsequent tones (Wilcoxon signed-rank test, FDR-adjusted).

Next, we examined whether the LFP response to the standard 50-ms tone decreased in the repetition after the deviant tone (corresponding to Supplementary Fig. 2e). The amplitude of LFP in the core during 0–200 ms after the tone onset did not apparently decrease as the repetition increased (Response Fig. 1b).

Then, we examined whether the tone response adapted throughout the oddball paradigm, in which the SOA was 550 ms and the same tone frequency was used. As a new paradigm, we introduced repetitive (10 times) 50-ms standard tones with SOA of 550 ms (a standard-tone block). We repeated the standard-tone block with an inter-block interval of 10 s (standard paradigm). During this standard paradigm, we conducted LFP recording in the core. If the response were to show adaptation in the first few trials in the oddball paradigm of each session and remain at a steady-state low level, the response to the first tone in the standard-tone block would be higher than the responses to the following tones in the block. However, they were not significantly different (orange line in Response Fig. 1c).

Another possibility is that the adaptation threshold in the marmoset is high compared with other species, and that adaptation did not occur in our current condition. To test this, we shortened the SOA from 550 ms to 350 ms in the standard paradigm to make adaptation by repeated stimulation more likely to occur. In fact, papers that studied stimulus-specific adaptation in the auditory cortex of rodents frequently used an interstimulus interval or SOA of 300 ms (Natan et al., *eLife* 4, e09868, 2015; Nieto-Diego and Malmierca, *PLoS. Biol.* 14, e1002397, 2016). Shortening the SOA decreased the core responses to the second and subsequent tones (green line in Response Fig. 1c). Thus, the reason why core response adaptation was not apparent in the current study was because the stimulation condition was too weak to induce it in the awake marmoset. We have added these results to Fig. 2g–i.

By contrast, when we conducted LFP recording of the RPB in the standard paradigm, in which the tone duration was 50 ms and the SOA was 550 ms, response attenuation rapidly occurred (Response Fig. 7a). This was substantially different from the non-adaptive response in the core, as shown in Response Fig. 1c. Furthermore, we found that the RPB response to a 100-ms tone during the many-standards control paradigm adapted to a greater extent than the RPB response to a 100-ms tone in the absence of the preceding tone presentation (Response Fig. 7b). In both LFP recording and calcium imaging, the amplitude of the response to the 50-ms tone (averaged over 0–200 ms after the tone onset) did not differ between the oddball and many-standards control paradigms

(Response Fig. 7c, d). These results indicate that when a tone with the same frequency was repeated, whether or not the duration of the tone was constant, the RPB response rapidly adapted and remained at a low steady-state level. We have added these results to Fig. 3h and Supplementary Fig. 3g–i.

Response Figure 7. **a**, Averaged LFP amplitude (during 0–200 ms after the tone onset) from the RPB in the standard paradigm with SOA of 550 ms (orange) or 350 ms (green). $n = 15$ sessions from two animals that were the same as those in Response Fig. 1c. $*P < 0.05$, the first tone vs. the second and subsequent tones (Wilcoxon signed-rank test, FDR-adjusted, SOA 550 ms: $n = 15$; 4 at 2 kHz, 4 at 4 kHz, 6 at 8 kHz, and 2 at 16 kHz from two marmosets. SOA 350 ms: $n = 15$, 4 at 2 kHz, 4 at 4 kHz, 6 at 8 kHz, and 2 at 16 kHz). **b**, Averaged LFP amplitude (during 0–200 ms after the tone onset) from the RPB in the many-standards control paradigm with a first tone duration of 100 ms. On the left is the response to the first 100-ms tone and on the right is the average response to the 100-ms tones during the paradigm. $P < 0.001$, Wilcoxon signed-rank test ($n = 50$ sessions from one of those used in **a**; $n = 18$ at 1 kHz, 17 at 2 kHz, and 15 at 4 kHz). **c**, Averaged LFP amplitude in the RPB to the 50-ms tone in oddball (orange) and many-standards control (blue) paradigms ($n = 9$ from one animal that was used in Fig. 3g). The amplitude averaged over 0–200 ms after the tone onset was not significantly different ($P = 0.37$, t -test). **d**, Averaged amplitude of the calcium response in the RPB to the 50-ms tone in the oddball and many-standards control paradigms ($n = 25$ from two animals that were used in Fig. 3c). The amplitude averaged over 0–200 ms after the tone onset was not significantly different ($P = 0.13$, t -test).

If the response to the 100-ms tone was large in the oddball paradigm because there were neurons that specifically responded to the 100-ms tone but not to any other tones, and they were non-adapted during the repetition of 50-ms tones, then the amplitude of the response should be similar between the oddball and many-standards control paradigms because the presentation probability of the 100-ms tone was 0.10 in both paradigms and the 100-ms tone-specific neurons should not be adapted in either paradigm. However, the response to the 100-ms tone was lower in the many-standards control paradigm than in the oddball paradigm. Thus, it is rational to assume that the response to the 100-ms tone in the oddball paradigm was larger because the auditory cortex predicted the repetition of the 50-ms tones and detected the 100-ms tone as deviant (prediction error). Consistent with this

hypothesis, in the many-standards control paradigm, the responses of offset+ neurons were large for the tones with 10, 200, and 225 ms, which deviated considerably from the average sound duration (~100 ms) (Response Fig. 2). Thus, we speculate that the RPB activity is attenuated during predictable stimuli, and that when an unpredictable stimulus is detected, the attenuation is removed by disinhibition (Wacongne et al., *J. Neurosci.* 2012; Chien et al., *Biol. Cybernetics* 2019) and/or strong newly-generated excitatory inputs, and the RPB is able to respond. RPB then sends error signals back to the core. This signal may also be sent to some higher-order brain areas including PFC, which may trigger attention and some cognitive process. We have added these results to the Results section as follows:

“In contrast to the LFP response in the core, the LFP response in the RPB rapidly attenuated in the standard paradigm with the tone duration of 50 ms and SOA of 550 ms (Fig. 3h). In addition, the response was much reduced during the many-standards control paradigm (Supplementary Fig. 3g). The response to the 50 ms tone was similarly weak in the oddball and many-standards control paradigms in both LFP recording and calcium imaging, and the difference between these paradigms was not significant (Supplementary Fig. 3h,i). These results suggest that the repetitive stimulation with the same tone frequency strongly attenuated the RPB response to a low steady-state level regardless of the duration of sound presentation, and there was not apparent duration-specific adaptation as in the core at least under the present stimulus conditions”.

In the rat, mPFC neurons slowly respond to the deviant tone with a latency of 120–180 ms from the tone onset in the frequency MMN paradigm, and mPFC more rapidly shows attenuation to the repetitive tones than does the auditory cortex (Casado-Roman et al., *PLoS Biol.* 18, e3001019, 2020). Thus, mPFC in the rat shows pure deviance detection, which is similar to that in the RPB in the marmoset. We have cited this paper and discussed how the pure prediction error might be generated in the Discussion section, as follows (lines 335–349):

“The deviance detection in RPB neurons with BF of F_{odd} and offset responses was prominent, and its amplitude was much larger than that of the response to 100-ms tones in the many-standards control paradigm, even though the presentation probability of the 100-

ms tone was the same (0.1) in both paradigms. The RPB responses to 50-ms tones in both paradigms were similarly suppressed to a great extent, although the presentation probability of the 50-ms tone differed substantially between these paradigms (0.9 vs. 0.1). Thus, it is rational to assume that the auditory cortex detected that the 100-ms tone deviated from the repetitive 50-ms tone, and that it was not from random-order tones with 10 different tones (10–225 ms). Consistent with this, the RPB neurons significantly responded to those tones whose durations deviated from the averaged tone duration in the many-standards control paradigms. In the rat, attenuation of the auditory response to the repetitive stimulus is stronger in the higher-order auditory cortex than in A1, and stronger in the medial prefrontal cortex than in the higher-order auditory cortex^{35,47}. Thus, it may be a cross-species commonality that the higher the hierarchy of the cortical areas, the weaker the response to repeated stimuli. However, it remains unknown how the tone duration-independent attenuation occurred in RPB but not in the core, and how the deviant tone resumed the suppressed activity”.

2) The authors chose to calculate DD as the difference in calcium signal between responses to oddball tone in oddball paradigm and the responses to the same duration tone in the many-standards paradigm. How would the DD be when calculated in a more conventional way, i.e., comparing the calcium responses to oddball tone in the oddball paradigm and those to the standard tone in oddball paradigm?

The calcium responses to the deviant tone were larger than those to the standard tone in the oddball paradigm in both the core and RPB (Response Fig. 8). However, we used the oddball paradigm to detect the duration mismatch negativity, in which the standard tone duration was 50 ms and the deviant tone duration was 100 ms, so that any difference was likely to be simply due to the difference in tone duration. Thus, we compared the 100-ms tone in the oddball paradigm and the 100-ms tone in the many-standards control paradigm.

Response Figure 8. Averaged time courses of the responses to the deviant tone (100 ms; orange) and 50-ms standard tone (black) immediately before the deviant tone in the oddball paradigm in the core (a) and RPB (b). The black horizontal lines indicate the period over which the deviance detection was significant ($P < 0.05$, Wilcoxon rank sum test, FDR-adjusted, the data are the same as in Fig. 2e and 3f).

3) Would the DD be still present when swapping the tone duration in oddball paradigm, i.e., using shorter duration of 50 ms for oddball tone and longer duration of 100 ms for standard tone and compare with responses to the 50-ms tone in the many-standards paradigm?

We have not conducted this paradigm. In a reversal of the conditions of the duration MMN paradigm (the duration of the standard and deviant tones was 150 ms [or 125 ms] and 50 ms, respectively) in the rat, evidence for whether MMN occurs or not was not consistent across studies (Roger et al., *Psychophysiology*, 46, 1028–1032, 2009; Nakamura et al., *Front. Psychol.* 2, 367, 2011; Farley et al., *J. Neurosci.* 30, 16475–16484, 2010). Thus, although we agree that an experiment with the reversed condition is important to comprehensively understand deviance detection and duration mismatch negativity, we would need to examine different combinations of tone durations to confirm whether the deviance detection occurs or not. We think that this is beyond the scope of this study and is a suitable topic for future work.

4) Although the authors found that tonotopy was not clear in lateral belt and parabelt regions when using 1P calcium imaging (Fig 3b, line 154), they identified best frequencies (BF) of single neurons in these areas when using 2P calcium imaging (Ext Fig 3b-3d, line 158). This seems contradictory. Are neurons of different BF intermingled in these areas?

The proportion of the nearest neighbors that had the same BF was slightly higher in the core and ML/CL than in RPB and CPB (core, $44.5\% \pm 2.1\%$ $n = 32$ fields; ML/CL, $45.1\% \pm 3.9\%$, $n = 14$; RPB, $37.7\% \pm 4.3\%$, $n = 11$; CPB, $42.5\% \pm 4.8\%$, $n = 11$), but the differences were not significant ($P > 0.05$, unpaired t -test with Bonferroni correction). We suspect that the response detected from one-photon imaging included the responses from the neuropiles (axons and dendrites), as well as those from the neuronal somata in the superficial layers. The neuropiles with different BF might be intermingled more in the parabelt and belt than in the core. We have added these analytical results and interpretations to the Results section (lines 191–198).

5) The authors further characterized different response types according to the 2P calcium signal from neurons in different auditory cortical areas (Ext Table 1). Some offset responses, e.g., those in Ext Fig 3b-3d, look to me more like onset responses of longer latencies. I also wonder whether responses of onset+/offset+ are of two peaks or sustained responses. It would be helpful to provide exemplary traces and latencies for responses of onset+/offset-, onset+/offset-, onset-/offset+, and onset-/offset-.

We have added the response type and latency (defined as the time from the tone onset to the point when the response exceeded half of the maximal amplitude in the pure-tone paradigm) of each neuron to Fig. 1g,h and Supplementary Fig. 3b–d. We simply classified the neurons to the four types according to a statistical test between the time-averaged activity. Thus, whether onset+ offset+ neurons showed two peaks or sustained responses depended on the peak timings of the activities during the tone presentation period and during the period after the end of the tone presentation. We have added this description to the Methods section (lines 899–901).

6) The calculation of selectivity index (SI) is not clear. It would be helpful if the authors can provide exemplary traces showing similar tuning widths across core, ML/CL, RPB, and CPB (Ext Table 1).

We calculated the fluorescence intensity during 3 s after the tone onset minus the fluorescence intensity at baseline (1 s before the tone onset) for each of five sound frequencies in the pure-tone paradigm, and then calculated the selectivity index as the sum of the five relative intensities obtained by scaling the response to the best frequency to a value of 1. The maximum value of the range was 5 when the neuron responded equally to all frequencies, and the minimum value was 1 when the neuron responded to only one frequency (there was no negative response). We have added the SI value to each neuron shown in Fig. 1g,h and Supplementary Fig. 3b–d. For example, in neuron 1 in Fig. 1g and neuron 6 in Supplementary Fig. 3b, which showed strong responses to only one tone frequency, SI was low at ~1.6. By contrast, in neurons 5, 7, and 9 in Supplementary Fig. 3b–d, which showed similar responses to multiple tone frequencies, SI was high at 2.80–3.04. Thus, although the SI calculation was simple, we think that the SI reflected the basic property of the tuning width of individual neurons.

7) The authors observed very strong responses to oddball tone in the oddball paradigm and very weak responses to the control stimulus in the many-standards paradigm using 1P calcium imaging in RPB, suggesting the presence of pure error signal (Fig 3d-de). As RPB neurons can be responsive to pure-tones of longer duration of 500 ms (Ext fig 31-3d, using 2P calcium imaging), I wonder if such pure error signal in RPB exists only for shorter tone duration. Precisely, I wonder what would be the range of oddball tone duration for eliciting pure error signal, or RES, in RPB.

When we examined the amplitude of the response in offset+ neurons whose BF was F_{odd} to each tone duration during ~100–300 ms after the tone end in the many-standards control paradigm, the amplitude of these neurons in the core increased as the tone duration increased (Response Fig. 2). By contrast, the amplitude was large with the 10-ms tone and rapidly increased from ~150 ms to 225 ms tone duration in the RPB (Response Fig. 2). The large response to the 10-ms tone might be related to detection of the tone omission because

the current tone stimulation condition was weak, as described in our response to the first comment. We have added these results to Fig. 5e–h. The dependency of the response to the tone duration in ML/CL appeared to be intermediate to those in the core and RPB, and such a dependency was not clear in CPB neurons. These results suggest that the offset+ neurons in the RPB had a different property for detecting the tone duration to those in the other auditory cortical areas. Thus, under the current condition, the tone duration for inducing the pure error signal in the RPB neurons with offset responses was 25–175 ms. The residual error signal or pure error signal does not mean that the sound response is not included at all, but that it is used as a response that has mostly error signal. Therefore, we have removed the term ‘residual error signal’ and weakened the use of ‘pure error signal’ throughout the paper.

Response Figure 2. Averaged response amplitudes (100–300 ms after the tone end) of the offset+ neurons whose BF was F_{odd} to each of the 10–225-ms tones in the many-standards control paradigm, and to the 100-ms and 50-ms tones in the oddball paradigm in each region (one-sample t -test, FDR-adjusted, * $P < 0.05$, • $P < 0.01$, •• $P < 0.001$, $n = 71$ for core, $n = 20$ for ML/CL, $n = 21$ for RPB, $n = 8$ for CPB).

In addition, the weak response in the RPB during the many-standards control paradigm was due to attenuation, as described in the response to reviewer #2’s first comment. We have clearly described this result in the main text.

8) The time course of DD appears to be faster when using 1P calcium imaging compared to when using 2P calcium imaging (e.g., Fig 3f vs Fig 4g, and perhaps Fig 4a vs fig 4b-4d). What would be the reason for this? It would be very helpful to label clearly how signal was obtained (e.g., using 1P or 2P imaging) when showing calcium traces.

The signal-to-noise ratio was higher in one-photon imaging than in two-photon imaging because the responses detected with one-photon imaging were the summed responses of many neurons (including neuropils) in the ROIs. In addition, the neuropil activity in L1 and L2/3 might be largely involved in the fluorescence intensity on one-photon imaging and showed relatively fast appearance compared with the L2/3 neuron activity. Therefore, the small difference in the response around 50–200 ms after the tone onset might be significantly detected in one-photon imaging. We have used the labels “1P” for one-photon imaging and “2P” for two-photon imaging in the corresponding calcium traces in Fig. 1–7 and Supplementary Fig. 2–7.

9) In Fig 5, the authors sought to state that DD was largest in RPB neurons who displayed offset responses and upon oddball stimulation of their BF. I think it has to be clearly put that BF of RPB neurons were determined using 2P calcium imaging upon pure-tone paradigm (Ext Fig 3a-3d, Ext Fig 5c), in which the tone duration and inter-stimulus interval were longer compared to oddball and many-standards paradigms. The onset/offset responses may not be obvious upon oddball and many-standards paradigms of shorter tone duration.

We have added text stating that the BF that was determined in the pure-tone paradigm was not necessarily the best for the shorter tone stimulation (lines 240 and 241). However, among the offset+ and offset- neurons in the RPB neurons whose BF was F_{odd} and the RPB neurons whose BF was not F_{odd} , the response to the 225-ms tone in the many-standards control paradigm was the largest (Response Fig. 9). This suggests that many of the neurons that had a certain tone frequency as their best frequency during the 500-ms presentation responded strongly to the same tone frequency during presentation of shorter stimulus durations, although the response to the 225-ms tone might also include some deviance detection component. We have added these traces to Supplementary Fig. 5g.

Response Figure 9. Response amplitude (averaged from 333 ms after the tone onset to 533 ms after the onset) of four types of RPB neurons to the 225-ms tone in the many-standards control paradigm. The neurons with and without offset responses were further divided into those whose BF was F_{odd} (F_{odd} Offset+ and F_{odd} Offset-) and those whose BF was not F_{odd} (non- F_{odd} Offset+ and non- F_{odd} Offset-). ** $P < 0.01$, unpaired t -test with Bonferroni correction. $d = 0.92$ for F_{odd} Offset+ vs. F_{odd} Offset-, $d = 0.91$ for F_{odd} Offset+ vs. non- F_{odd} Offset-, $n = 21$ for F_{odd} Offset+, 47 for F_{odd} Offset-, 12 for non- F_{odd} Offset+, and 56 for non- F_{odd} Offset- from two animals.

10) The authors found that DD in the core was reduced when RPB was inactivated by injecting muscimol there (Fig 7a-7d). This is a significant finding suggesting that RPB is necessary for DD at the core. In Fig 7b, there may be missing black lines for indicating DD. The description in line 244-245 is confusing; it seems that this result is derived according Fig 7d and it would be helpful that the authors can provide more detailed description.

In Fig. 7b, there was no significant difference between any time point. In this statistical analysis, the values were compared at each time point using a false discovery rate-adjusted p -value of < 0.05 . This test is less likely to be significant than the test that simply compared the averaged values over the 200–400 ms period shown in Fig. 7d because there was more measurement variation and more data points in the former test than in the latter test. We have added this interpretation to the legend of Fig. 7.

It is however still not clear whether the reduced deviance detection in the core was caused by reduced responses to 100-ms tone in the oddball paradigm or enhanced responses to 100-ms tone in the many-standards control paradigm. Since inactivating RPB by using muscimol

injection could be non-specific, would the effect of reduced DD be still present when using more specific manipulation approach, e.g. optogenetic inhibition, for inactivating RPB?

The core response to the 100-ms tone in the many-standards control paradigm was not significantly different among the conditions with RPB muscimol injection, CPB muscimol injection, and RPB saline injection (unpaired *t*-test with Bonferroni correction, $P = 0.85$ for RPB muscimol vs CPB muscimol, $P = 0.70$ for RPB muscimol vs RPB saline, $P = 0.85$ for RPB saline vs CPB muscimol). We have added these statistical analyses to the legend of Fig. 7d. Effective optogenetic inactivation of a cortical subarea in the marmoset is still under development, and was therefore difficult to apply in the current study.

11) The authors simulated oddball stimulation by presenting a sequence of standard tones and applied optogenetic activation in RPB right after tone stimulation for 10% of the standard tones. The authors found enhanced late responses (after ~250 ms from tone onset) to stimulation of simulated oddball stimulus, resembling DD (Fig 7e-7f, Ext Fig 8e). This is a very interesting finding. It is however not clear how this can be linked to the generation of DD at the core. I think it would be helpful for understanding how RPB activation may contribute to DD generation at the core by plotting the trace of photostimulation only in Fig 7e.

We have added the trace of the core response to only photostimulation to Supplementary Fig. 8i. In this calculation, we removed the core area that showed a significant response to only photostimulation (averaged over 0–500 ms after the stimulation onset), as shown in the legend of Fig. 7f. There is a possibility that GCaMP could be excited by the red light used for ChrimsonR stimulation and its fluorescence could increase. Additionally, contamination from reflected light might be present even when GCaMP was not excited. Therefore, even if the fluorescence in the core increased because of the photostimulation alone, we cannot conclude that it was neural activity in the core induced by the photoactivated RPB. As a result, we have excluded this region from the calculation.

Nevertheless, the tone response in the remaining core area was enhanced by the photostimulation (Fig. 7f,g). We have rewritten the sentences in the main text regarding the optogenetic experiment as follows (lines 319–323):

“This photostimulation enhanced the response in the core region in which the activity was not evoked by only photostimulation (Fig. 7f and Supplementary Fig. 8h,i), and the response remaining after the end of the photostimulation was significantly enhanced (Fig. 7g). Thus, RPB activation immediately after the tone end can enhance the tone response in core neurons in a non-linear manner”.

We have also conducted additional experiments on the optogenetic RPB activation in two animals to strengthen our results. On the basis of these experiments, we propose that even when the core was not sufficiently depolarized to exhibit action potentials by the RPB photostimulation alone, the core that had already been depolarized by the sound stimulation could be depolarized to exceed the action potential threshold by the RPB photostimulation. We have added a schema proposing how the RPB activation affects the deviance detection in the core as Fig. 8.

Here are some minor comments:

1) The scale bars of y-axis are missing in Fig 1f, Fig 2b, Extended Fig 3a.

We have added scale bars to Fig. 1f and 2b and Supplementary Fig. 3a.

2) The reference for different response types measured using electrophysiological recording in mice A1 in line 166 is missing.

We have changed the reference to Okazaki et al. (Hear. Res. 259, 107–116, 2010), a study that used guinea pigs.

3) It's not clear to me the difference between Fig 4b-4d vs Fig 4e.

The top panels of Fig. 4b–4d show representative images of the same field at different cortical depths in the core (b, L1; c and d, the same depth in L2/3). The bottom panels show the summed responses within the imaging field in L1 (b), within the L2/3 field without the detected active neuronal somata (c), and in the L2/3 neuronal somata detected as active neurons within the field (d). Fig. 4e is the session-averaged response in the core.

4) In Ext Fig 4a, for the first neuron (top) at the core, the tone-evoked response to standard tones were weak. Are the 2 neurons in Ext Fig 4a both showing DD? If so, it would be helpful to provide the numbers of neurons, showing and not showing DD, in each auditory cortical area; the authors can consider to put line 714-717 in Results.

We have added horizontal bars showing the time points with statistically significant deviance detection to each trace, and added the p-value for testing whether the amplitude of the deviance detection averaged over 200–400 ms after the tone onset was significant to the corresponding legend.

5) It is not clear whether the responses shown in Fig 5 were obtained using 1P or 2P calcium imaging (I suppose 2P imaging).

All data in Fig. 5 were from the neuron data detected with two-photon imaging. We have labeled “2P” in Fig. 5b,c, as described above.

6) It is not clear what are the grey and pink shades in Fig 7e.

Schematically, light blue shading indicates the blue laser illumination for excitation of GCaMP6f and pink shading indicates the red-light illumination for excitation of ChrimsonR. We have made these colors a little darker and added an explanation of the colors to the legend of Fig. 7e.

REVIEWER COMMENTS

Reviewer #1 (Remarks to the Author):

all my comments have been addressed

Reviewer #2 (Remarks to the Author):

Obara et al performed new analyses and conducted new experiments in the revised manuscript and the rebuttal letter. The authors provided new results to address adaption in responses to tones in many-standard and oddball paradigms, e.g., revealing different time courses of adaptation in core and RPB areas, frequency-selectivity of deviance detection, the spread of muscimol injection, which was used for inactivating activity in parabelt regions. They further proposed a schematic to illustrate possible activity flow related to deviance detection in auditory cortical areas. In my opinion, the manuscript can potentially be considered for publication after revision.

Here are my comments to the authors' responses.

Response to main comment 1)

The authors performed several new analyses and new experiments to address the issue of adaptation in responses to repeating standard tones in oddball paradigm and its relation to deviance detection in the core and RPB areas.

I'd like to comment on their responses in 2 parts.

Part 1:

To address the issue of reduced adaptation to the repeating 50-ms ('standard') tones in the current oddball paradigm in the core area, first, the authors compared the LFP responses to 50-ms tones in oddball paradigm and to 50-ms tones in many-standard paradigm in core. They did not find significant difference between these responses in the 2 paradigms (Response Fig 1a), confirming the results found in calcium signal (Fig 2b, Fig S2d,e) and suggesting, if any, similar attenuation levels in responses to 50-ms tones in the current oddball and many-standard paradigms. Second, the authors compared the averaged LFP amplitudes to the 50-ms tones in many-standard paradigm with those to the repeating 50-ms tones in oddball paradigm in core. They found no significant difference in those responses (Response Fig R1b), confirming the absence of adaptation. Third, the authors conducted new experiments of LFP measurements using repeating 50-ms tones in 'standard paradigm' of 350-ms SOA and 550-ms SOA in core. They found that averaged LFP amplitudes did not attenuate with the repeated tone presentation in standard paradigm of 550 ms SOA, but adapted significantly in standard paradigm of 350-ms SOA (Response Fig1c). The authors concluded that adaptation could occur in core and that the 550-ms SOA in oddball paradigm was too long to induce apparent simple adaptation in core.

The authors further performed similar analyses and experiments to investigate adaptation in the RPB area. While similar attenuation levels were also reported in LFP responses to the 50-ms tones in oddball and many-standard paradigms (Response Fig 7c, 7d), significant and rapid adaptation was found in LFP responses to 50-ms tones in standard paradigm presented at both 350-ms SOA and 550-ms SOA (Response Fig 7a).

These results confirm no apparent adaptation in responses to standard tones in core upon stimulation of current oddball paradigm (550-ms SOA), and suggest fast-adapting responses to repeating standard tones in RPB.

I think that the new LFP measurements using standard paradigms of 50-ms tones of 350-ms SOA and 550-ms SOA in core and RPB provide direct evidence for the different time courses in adaptation in core and in RPB. These results seem to be different from those in previous studies such as Ulanovsky et al, Nat Neurosci, 2003, 2004, Taaseh et al, PloS One, 2011, and Fishman et al., J Neurosci, 2012, in which stimulus-specific adaptation was reported in primary auditory cortex using an inter-stimulus interval up to 2000 ms. I suggest to add a few lines to discuss about the different time courses of adaptation and their neuronal mechanisms (e.g., synaptic depression). In addition, I think that additional measurements of responses to 100-ms tones in many-standard and oddball paradigms of 350-ms SOA could help to clarify whether the presence of adaptation affects deviance detection in core.

Part 2:

The authors further analyzed responses to tones of different durations in the current many-standard paradigm in RPB. They observed that averaged LFP amplitudes to 100-ms tones that were not first presented attenuated more compared to those to the first presented 100-ms tones (Response Fig 7b), suggesting that adaptation could also occur when the same-duration tones reoccurred but not repetitively. They also found that offset(+) neurons in RPB showed significantly increased responses to tone durations that were very different from the average tone duration in many-standard paradigm, i.e., 10, 200, and 225 ms (Response Fig 2), seemingly suggesting detection of 'deviant' tone durations in these neurons. According to these results, the authors proposed that upon stimulation of many-standard paradigm, both deviance detection and adaptation could take place in RPB.

I find these results very interesting, but I think one has to be cautious with the interpretation because deviance detection was not actually measured by comparing responses to same-duration tones in many-standard and oddball paradigm (please see main comment 7)). For example, it remains unclear to me what would separate 10, 200, 225 ms from the rest of the tone durations in many-standard paradigm as 'sort of deviant' durations.

Here are some more comments:

a) I think it would be more comprehensive if the authors could plot LFP responses in RPB in a similar graph as in Response Fig1b, e.g., from the results in Fig S4.

b) In Response Fig 1a, I wonder if only the response to the standard tone was chosen for the standard tones immediately before the oddball tones or all standard tones.

Response to main comment 2)

According to a framework involving adaptation, the difference between the responses to 100-ms oddball tones and those to the 50-ms standard tones, i.e., duration mismatch negativity (dMMN) (Response Fig 8), may reflect not only deviance detection but also the different tuning to tone durations and adaptation component (Koshiyama et al., *Schizophrenia Bulletin*, 2020). I agree that it is reasonable that the authors assessed the 'deviant detection component' by comparing the 100-ms tones in oddball and many-standard paradigms.

In the response to my main comment 1), the authors identified the 'adaptation component' via the difference between the responses to 50-ms tones in many-standard paradigm and those to 50-ms tones in oddball paradigm (Response Fig 1a, Fig S2d, Response Fig 7c,7d), and they found that this component was not apparent, or similar attenuation levels in responses, in core and RPB. I understand that the authors focused on deviance detection but not dMMN in the current study, but I am still curious about the 'tone difference component' by comparing the responses to the 2 tones durations (here 50-ms and 100-ms), which were used in oddball paradigm, in the many-standard paradigm. This way, one could estimate its contribution to dMMN and may provide hints for selecting tone durations for inducing dMMN (e.g., if the adaptation component is not apparent, dMMN may not occur when the tone difference component is of opposite sign and larger amplitude compared to deviance detection component).

Response to main comment 3)

It seems that dMMN may not necessarily be induced when using a shorter tone duration for oddball tones compared to standard tones. I understand that the focus of the current study is to apply a duration oddball paradigm which can reliably induce deviance detection in auditory cortical areas, and I agree that additional experiments by swapping the tone durations for oddball and standard tone are not the most relevant. I however think that these experiments could provide insights into the generation of dMMN and deviance detection. For example, if dMMN could still occur when using oddball tones of a shorter tone duration, this may hint at additional mechanism for deviance detection such as sensory memory (Schroeger, *Journal of Psychophysiology*, 2007). I think that it would be helpful if the authors could add a sentence to explain the choice to tone durations for oddball and standard tones in the current study.

Response to main comment 4)

As the probability of neighboring neurons displaying the same BF was moderate in ML/CL, RPB, and CPB regions (~40%), neuronal somata of the same BF may not be tightly clustered. I agree that one-photon imaging was likely to collect calcium signal also from axons and dendrites, and such neuropil signal may confound the spiking activity in cell somata, especially when they were

not tightly clustered, obscuring potential tonotopic organization in lateral belt and parabelt areas. I think that tonotopy can be more reliably measured by using electrophysiological recording. I suggest to explain more clearly the frequency representation in these areas in the sentence at line 182 with the reference using electrophysiological recording.

Response to main comment 5)

The authors properly addressed my comment.

Response to main comment 6)

The authors provided more information about calculating selectivity index here in the response letter: '... We calculated the fluorescence intensity 3 s after tone onset minus the fluorescence intensity at baseline (1 s before the tone onset) for each of five sound frequencies in the pure-tone paradigm... a value of 1...' I think it would be more clear to replace the sentence in line 890-892 with this one. Adding SI values to neurons shown in Fig 1g,h and Fig S3b-d helps to make clear neurons of different selectivity to the 5 tone frequencies.

Response to main comment 7)

Indeed, pure error signal would be more likely to occur when a) strong responses to oddball tones in oddball paradigm and b) weak responses to the same duration tones in many-standard paradigm were induced. The authors performed new analyses to assess the tone durations that matched the a) and b) criteria. In RPB, they found that tone durations of 25-175 ms in many-standard paradigm did not induce significant responses in offset+ neurons whose BF=F_{Odd} (Response Fig 2, RPB), possibly due to adaptation (response to my main comment 1)). Hence, if tone durations of 25-175 ms of oddball tones in oddball paradigm would induce significant responses in these neurons, this tone duration range could potentially induce pure error signal. While I find these results very interesting, I however think that one has to be careful with the interpretation because deviance detection was not measured using tone duration within this range and other than 100 ms. For example, I notice significant responses to 50-ms standard tones in oddball paradigm in these neurons. Would this imply the generation of pure error signal upon presentation of standard tones in oddball paradigm in these neurons?

I find it intriguing that 10-ms tones in many-standard paradigm evoked significant responses in these neurons. Did the presentation of many-standard paradigm always start with 10-ms tones? Also, I wonder if the tone presentation in many-standard paradigm was randomized.

Response to main comment 8)

The authors properly addressed my comment. To be consistent, I suggest to label '2P' for the traces in Fig 1g, 1h and Fig S3b-d.

Response to main comment 9)

Neurons in RPB were classified into 4 types according to their offset responses and the matching between their BF and F_{Odd}. Comparing the tone responses to the longest duration tones, i.e. of 225 ms, in many-standard paradigm for the 4 types of neurons, the author reported that the tone responses were the largest when the neurons were offset+ and the presented tone frequencies, the same as F_{Odd}, were their BF (Response Fig 9). They concluded that largest responses would still be elicited when tones of BF were presented with shorter duration compared to that in pure-tone paradigm, possibly including some deviance detection component as 225 ms was very different from the average tone duration in many-standard paradigm (please see response to my main comment 1)).

In Response Fig 9, I wonder if there was significant difference between responses to 225-ms tones in F_{Odd}Offset+ neurons and those in non-F_{Odd}Offset+. I also wonder whether the responses to 225-ms tones in RPB neurons whose BF=F_{Odd}, both offset+ and offset-, were significantly larger compared to those in RPB neuron whose BF≠F_{Odd}, both offset+ and offset-.

The authors suggest that a deviance detection component may be included in the responses to 225-ms tones in many-standard paradigm. It remains unclear to me how 225-ms, along with 10-ms and 200-ms, tone duration was classified as 'deviant' durations in many-standard paradigm. I think that similar analyses as in Response Fig 9 for tone durations of 25-175 ms in many-standard paradigm, which may not induce deviance detection, could help clarify whether neurons show selective responses to BF when tones of shorter duration were used.

Response to main comment 10)

It is clear to me now why no black lines were indicated in Fig 7b and significant difference between responses to oddball tone and those to the same duration tones in many-standard paradigm indicated in Fig 7d.

To address whether the reduced deviance detection in core was caused by reduced responses to 100-ms tones in oddball paradigm or enhanced responses to 100-ms tones in many-standard paradigm, the authors compared the responses to 100-ms tones in many-standard paradigm with RPB muscimol, CPB muscimol, and RPB saline, and found no significant difference among them. This result indicated that muscimol injection in RPB and CPB would not significantly affect the response to the 100-ms many-standard tones compared to control manipulation of RPB saline, suggesting that the reduced deviance detection with RPB muscimol may be resulted from decreased responses to the 100-ms oddball tones. I think that a more direct way to address this question could be to compare the responses to 100-ms oddball tones and 100-ms many-standard tones before and after pharmacological manipulations, because this way potential compound effects on the tone responses due to injection of muscimol or saline could be excluded.

It is clear to me now effective optogenetic inactivation of RPB is still under development and not yet ready to be applied in the current study.

Response to main comment 11)

The authors explained that subareas in the core showed artifactual signal upon optogenetic stimulation of RPB (probably as shown in Fig S8h(3)) and therefore were removed for calculating the core responses upon photostimulating RPB (Fig S8i). I think that justification of this procedure should be provided in the manuscript. For example, I expect that artifactual signal upon photostimulating RPB may display different temporal profiles compared to tone-evoked responses. It remains yet unclear how RPB activation contributes to deviance detection in core. I think it'd be helpful if the authors could discuss about potential mechanisms according to their preliminary results described in the response letter.

Response to minor comment 1)

The authors properly addressed my comment.

Response to minor comment 2)

The authors properly addressed my comment.

Response to minor comment 3)

I find the bottom panels of Fig 4b-d still not clear. I think it'd be helpful to indicate their sample sizes and SEM. Fig 4e is clear to me now.

Response to minor comment 4)

Adding black lines indicating significant deviance detection in the left-most panels of Fig S4a, S4b and the p-values for testing deviance detection in averaged responses in these panels addressed my comment.

I still find line 908-911 in the revised manuscript should be moved to Results or Discussion.

Response to minor comment 5)

The authors properly addressed my comment.

Response to minor comment 6)

The authors properly addressed my comments.

Other comments

a) Sometimes data from just one animal were included (e.g., Response Fig7c). I wonder whether variation between animal could affect conclusion.

b) The blue and oranges traces in Fig 3f,3g, Fig 6d, Fig7a-d, Fig S6e, Fig S7d, S7e were not described in figure legends.

c) Line 122: in ref 32 no many-standard paradigm was used.

We thank all the reviewers for their careful consideration of our manuscript and for making helpful comments. Our detailed responses (in black) to the reviewer's comments (in blue) are provided below:

Reviewer #2 (Remarks to the Author):

Obara et al performed new analyses and conducted new experiments in the revised manuscript and the rebuttal letter. The authors provided new results to address adaption in responses to tones in many-standard and oddball paradigms, e.g., revealing different time courses of adaptation in core and RPB areas, frequency-selectivity of deviance detection, the spread of muscimol injection, which was used for inactivating activity in parabelt regions. They further proposed a schematic to illustrate possible activity flow related to deviance detection in auditory cortical areas. In my opinion, the manuscript can potentially be considered for publication after revision.

We thank Reviewer #2 for the positive comment on our study.

Here are my comments to the authors' responses.

Response to main comment 1)

The authors performed several new analyses and new experiments to address the issue of adaptation in responses to repeating standard tones in oddball paradigm and its relation to deviance detection in the core and RPB areas.

I'd like to comment on their responses in 2 parts.

Part 1:

To address the issue of reduced adaptation to the repeating 50-ms ('standard') tones in the current oddball paradigm in the core area, first, the authors compared the LFP responses to 50-ms tones in oddball paradigm and to 50-ms tones in many-standard paradigm in core. They did not find significant difference between these responses in the 2 paradigms (Response Fig 1a), confirming the results found in calcium signal (Fig 2b, Fig S2d,e) and suggesting, if any, similar attenuation levels in responses to 50-ms tones in the current oddball and many-standard paradigms. Second, the authors compared the averaged LFP amplitudes to the 50-ms tones in many-standard paradigm with those to the repeating 50-ms tones in oddball paradigm in core. They found no significant difference in those responses (Response Fig R1b), confirming the absence of adaptation. Third, the authors conducted new experiments of LFP measurements using repeating 50-ms tones in 'standard paradigm' of 350-ms SOA and 550-ms SOA in core. They found that averaged LFP amplitudes did not attenuate with the repeated tone presentation in standard paradigm of 550 ms SOA, but adapted

significantly in standard paradigm of 350-ms SOA (Response Fig1c). The authors concluded that adaptation could occur in core and that the 550-ms SOA in oddball paradigm was too long to induce apparent simple adaptation in core.

The authors further performed similar analyses and experiments to investigate adaptation in the RPB area. While similar attenuation levels were also reported in LFP responses to the 50-ms tones in oddball and many-standard paradigms (Response Fig 7c, 7d), significant and rapid adaptation was found in LFP responses to 50-ms tones in standard paradigm presented at both 350-ms SOA and 550-ms SOA (Response Fig 7a).

These results confirm no apparent adaptation in responses to standard tones in core upon stimulation of current oddball paradigm (550-ms SOA), and suggest fast-adapting responses to repeating standard tones in RPB.

I think that the new LFP measurements using standard paradigms of 50-ms tones of 350-ms SOA and 550-ms SOA in core and RPB provide direct evidence for the different time courses in adaptation in core and in RPB. These results seem to be different from those in previous studies such as Ulanovsky et al, Nat Neurosci, 2003, 2004, Taaseh et al, PloS One, 2011, and Fishman et al., J Neurosci, 2012, in which stimulus-specific adaptation was reported in primary auditory cortex using an inter-stimulus interval up to 2000 ms. I suggest to add a few lines to discuss about the different time courses of adaptation and their neuronal mechanisms (e.g., synaptic depression).

In addition, I think that additional measurements of responses to 100-ms tones in many-standard and oddball paradigms of 350-ms SOA could help to clarify whether the presence of adaptation affects deviance detection in core.

Ulanovsky et al. (2003) used a tone duration of 230 ms in halothane-anesthetized rats. Fishman and Steinschneider (2012) used a tone duration of 200 ms in awake macaques. We think it is rational that a long tone duration might induce strong adaptation. By contrast, Taaseh et al. (2011) used a tone duration of only 30 ms. However, they used halothane-anesthetized rats. Halothane anesthesia might be related to the strong adaptation partly because halothane increases inhibitory synaptic transmission (Kotani & Akaike, Brain Res. Bull. 93, 69–79, 2013). However, we believe that it is too speculative to infer the reasons why the adaptation differed between these studies and our study because the species and experimental conditions are markedly different, and we did not compare the auditory responses between different animals or between awake and anesthetized conditions.

However, we agree that it is important to discuss the different time courses of adaptation and their neuronal mechanisms. We have added the following sentences (lines 168–170):

“...the adaptation time course might be slower in the marmoset core than in other examined species^{36,37}. This might be due to some different properties of short-term synaptic depression and suppression of excitatory neurons by inhibitory neurons^{20,34}”.

We agree that it is ideal to clarify whether the presence of adaptation affects the core deviance detection. However, the major aim of the current study is to show the occurrence of the error feedback from the RPB to the core in the change detection. It is unknown whether the properties of the deviance detection in the core, belt, and parabelt and the feedback from the RPB to the core when the adaptation occurs in the core are the same as those in the current condition. To determine this, we need to perform all experiments in the condition with SOA of 350 ms. We believe this is beyond the scope of the current study. Instead, we have added the following sentence as a limitation of the current study (line 434–437):

“It should be clarified whether similar deviance detection is detected in the core and RPB when the adaptation occurs in the core (for example, the SOA of 350 ms is used), and if so, whether the back-projection of the error signal from the RPB is important for the core deviance detection”.

Part 2:

The authors further analyzed responses to tones of different durations in the current many-standard paradigm in RPB. They observed that averaged LFP amplitudes to 100-ms tones that were not first presented attenuated more compared to those to the first presented 100-ms tones (Response Fig 7b), suggesting that adaptation could also occur when the same-duration tones reoccurred but not repetitively. They also found that offset(+) neurons in RPB showed significantly increased responses to tone durations that were very different from the average tone duration in many-standard paradigm, i.e., 10, 200, and 225 ms (Response Fig 2), seemingly suggesting detection of ‘deviant’ tone durations in these neurons. According to these results, the authors proposed that upon stimulation of many-standard paradigm, both deviance detection and adaptation could take place in RPB.

I find these results very interesting, but I think one has to be cautious with the interpretation because deviance detection was not actually measured by comparing responses to same-duration tones in many-standard and oddball paradigm (please see main comment 7)). For example, it remains unclear to me what would separate 10, 200, 225 ms from the rest of the tone durations in many-standard paradigm as ‘sort of deviant’ durations.

As we described in our response on pages 8 and 9, we do not intend to insist that the responses to 10-, 200-, and 225-ms tones in the many-standards control paradigm were the deviance detection. We have added our interpretation as follows (lines 367–372):

“This property of RPB neurons might also be related to the finding that they significantly responded to 10-, 200-, and 225-ms tones whose durations were relatively different from the averaged tone duration (113.5 ms) in the many-standards control paradigm. To understand the generic role of the

RPB in dMMN, it is necessary to examine the different combinations of deviant and standard-tone durations including short deviant and long standard tones^{7,11,48}.

Here are some more comments:

a) I think it would be more comprehensive if the authors could plot LFP responses in RPB in a similar graph as in Response Fig1b, e.g., from the results in Fig S4.

We have added a corresponding graph of the LFP responses in the RPB as Supplementary Fig. 3j. Apparent adaptation after the deviant tone presentation was not observed. We have added the related description in lines 229–231. Rather, the response appeared to be enhanced. However, it might be affected by the small sample size (nine penetrations).

Supplementary Figure 3j. The left-most blue dot shows the averaged LFP amplitude to the 50-ms tone in the many-standards control paradigm in the RPB over 0–200 ms after the tone onset ($n = 9$ penetrations from one animal that was used in Fig. 3g). The other plots show the averaged LFP amplitudes in the RPB to the i th 50-ms tone (labeled S_i , $i = 1–7$) after the deviant tone stimulus in the oddball paradigm ($n = 9$ penetrations from the same animal). Vertical lines indicate SEMs. The baseline amplitude was subtracted.

b) In Response Fig 1a, I wonder if only the response to the standard tone was chosen for the standard tones immediately before the oddball tones or all standard tones.

We apologize for the unclear explanation. In Response Fig. 1a (the same as Fig. 2g) and Supplementary Fig. 3h in the previous revised manuscript, only the response to the tone stimulus immediately before the deviant tone stimulus in the oddball paradigm was chosen. We describe this explanation in the legends of Fig. 2g and Supplementary Fig. 3h in the current revised manuscript.

Response to main comment 2)

According to a framework involving adaptation, the difference between the responses to 100-ms oddball tones and those to the 50-ms standard tones, i.e., duration mismatch negativity (dMMN) (Response Fig 8), may reflect not only deviance detection but also the different tuning to tone durations and adaptation component (Koshiyama et al., Schizophrenia Bulletin, 2020). I agree that it

is reasonable that the authors assessed the ‘deviant detection component’ by comparing the 100-ms tones in oddball and many-standard paradigms.

In the response to my main comment 1), the authors identified the ‘adaptation component’ via the difference between the responses to 50-ms tones in many-standard paradigm and those to 50-ms tones in oddball paradigm (Response Fig 1a, Fig S2d, Response Fig 7c,7d), and they found that this component was not apparent, or similar attenuation levels in responses, in core and RPB. I understand that the authors focused on deviance detection but not dMMN in the current study, but I am still curious about the ‘tone difference component’ by comparing the responses to the 2 tones durations (here 50-ms and 100-ms), which were used in oddball paradigm, in the many-standard paradigm. This way, one could estimate its contribution to dMMN and may provide hints for selecting tone durations for inducing dMMN (e.g., if the adaptation component is not apparent, dMMN may not occur when the tone difference component is of opposite sign and larger amplitude compared to deviance detection component).

To clarify the contributions of the deviance detection, tone difference, and adaptation components to dMMN (Koshiyama et al., 2020; Response Fig. 1a), we used one-photon imaging data in the core and RPB to calculate the amplitudes (averaged over 200–400 ms after the tone onset) of these components (Response Fig. 1b, c). Although the tone difference component was approximately 50% in the core and approximately 10% in the RPB, a clear deviance detection component was detected in both areas. We have added these results as Supplementary Fig. 2h, 2i, and 3k.

Response Figure 1. a, dMMN consisting of the deviance detection, tone difference, and adaptation components. **b, c**, Contributions of the three components in the core (**b**) and RPB (**c**). The data are the same as those in Fig. 2e (core, $n = 25$ sessions from three animals) and Fig. 3f (RPB, $n = 25$ sessions from two animals). For each session, each component amplitude (for 200–400 ms after the tone onset) was normalized to the dMMN amplitude. $**P < 0.01$, paired t -test with Bonferroni correction.

We have also added the following descriptions to the Results section:

“The amplitude of dMMN is defined as the response to the deviant (100 ms) tone minus the response to the standard (50 ms) tone in the oddball paradigm. dMMN consists of the deviance detection component, tone difference component (the response to the 100-ms tone in the many-standards control paradigm minus the response to the 50-ms tone in the many-standards control paradigm), and adaptation component¹³ (Supplementary Fig. 2h). In the core dMMN, the deviance detection component was similar to the tone difference component (Supplementary Fig. 2i)” (lines 171–176). “In dMMN in the RPB, the deviance detection component was much larger than the tone difference component (Supplementary Fig. 3k)” (lines 234–236).

Response to main comment 3)

It seems that dMMN may not necessarily be induced when using a shorter tone duration for oddball tones compared to standard tones. I understand that the focus of the current study is to apply a duration oddball paradigm which can reliably induce deviance detection in auditory cortical areas, and I agree that additional experiments by swapping the tone durations for oddball and standard tone are not the most relevant. I however think that these experiments could provide insights into the generation of dMMN and deviance detection. For example, if dMMN could still occur when using oddball tones of a shorter tone duration, this may hint at additional mechanism for deviance detection such as sensory memory (Schroeger, *Journal of Psychophysiology*, 2007).

I think that it would be helpful if the authors could add a sentence to explain the choice to tone durations for oddball and standard tones in the current study.

The major reason why we used the 100-ms tone as a deviant tone and the 50-ms tone as a standard tone was because we followed the paradigm used by Koshiyama et al. (2020), who showed dMMN in humans, and Tada et al. (*Front. Psychiatry* 11, 874, 2020), who showed dMMN in macaques. We also followed Avissar et al. (*Schizophr. Res.* 191, 25–34, 2018), who showed that the effect size of MMN deficits in schizophrenia is larger in long duration deviants than in shorter duration deviants, although this result should be interpreted with caution because the level of deviance may also influence the effect size. We have added the following sentences to the Methods section (lines 767–771).

“The major reason why we used the 100-ms tone as a deviant tone and the 50-ms tone as a standard tone was because we followed recent studies that showed dMMN in humans and macaques with the same combination of the tone duration^{13,33}. We also followed a meta-analysis study that showed that the effect size of MMN deficits in schizophrenia is larger in long duration deviants than in shorter duration deviants¹⁶”.

We have also emphasized the significance of swapping the tone duration as follows (lines 370–372):

“To understand the generic role of the RPB in dMMN, it is necessary to examine the different combinations of deviant and standard-tone durations including short deviant and long standard tones^{7,11,48}”.

Response to main comment 4)

As the probability of neighboring neurons displaying the same BF was moderate in ML/CL, RPB, and CPB regions (~40%), neuronal somata of the same BF may not be tightly clustered. I agree that one-photon imaging was likely to collect calcium signal also from axons and dendrites, and such neuropil signal may confound the spiking activity in cell somata, especially when they were not tightly clustered, obscuring potential tonotopic organization in lateral belt and parabelt areas. I think that tonotopy can be more reliably measured by using electrophysiological recording. I suggest to explain more clearly the frequency representation in these areas in the sentence at line 182 with the reference using electrophysiological recording.

We thank the reviewer for this important point. We have calculated the proportion of neurons for each BF. The highest proportion of neurons had a BF of 8 kHz in the RPB (45.1%) and 1 kHz in the CPB (37.7%). This difference was consistent with the result of one-photon imaging (Fig. 3b). We have added this result to Supplementary Table 2. We have also cited a paper with electrophysiological recording in the macaque parabelt (Kajikawa et al., J. Neurosci. 35, 4140–4150, 2015) that showed the rostral-caudal gradient of high-to-low BF in the RPB and low-to-high BF in the CPB as follows:

“In the imaging fields, the highest proportion of neurons had a BF of 8 kHz in the RPB (45.1%) and 1 kHz in the CPB (37.7%) (Supplementary Table 2). This difference was consistent with the result of one-photon imaging (Fig. 3b). Although the rostral-caudal gradient of high-to-low BF in the RPB and low-to-high BF in the CPB has been reported in electrical recording of macaques³⁸, we did not detect a low-to-high BF gradient in the CPB” (lines 198–202).

“To clarify the tone frequency representation in more detail, it is necessary to record the BF of individual neurons in superficial and deep layers over a broad area including the belt and parabelt” (lines 209–211).

Response to main comment 5)

The authors properly addressed my comment.

Response to main comment 6)

The authors provided more information about calculating selectivity index here in the response letter: ‘... We calculated the fluorescence intensity 3 s after tone onset minus the fluorescence intensity at baseline (1 s before the tone onset) for each of five sound frequencies in the pure-tone paradigm... a value of 1...’ I think it would be more clear to replace the sentence in line 890-892 with this one. Adding SI values to neurons shown in Fig 1g,h and Fig S3b-d helps to make clear neurons of different selectivity to the 5 tone frequencies.

According to the reviewer’s suggestion, we have replaced that sentence in lines 890–892 (in the previous manuscript) with the sentence that we used in the previous response letter (lines 942–946). We have also corrected the time windows to the fluorescence intensity 1 s after tone onset minus the fluorescence intensity at baseline (0.5 s before the tone onset).

Response to main comment 7)

Indeed, pure error signal would be more likely to occur when a) strong responses to oddball tones in oddball paradigm and b) weak responses to the same duration tones in many-standard paradigm were induced. The authors performed new analyses to assess the tone durations that matched the a) and b) criteria. In RPB, they found that tone durations of 25-175 ms in many-standard paradigm did not induce significant responses in offset+ neurons whose $BF = FO_{\text{odd}}$ (Response Fig 2, RPB), possibly due to adaptation (response to my main comment 1)). Hence, if tone durations of 25-175 ms of oddball tones in oddball paradigm would induce significant responses in these neurons, this tone duration range could potentially induce pure error signal.

While I find these results very interesting, I however think that one has to be careful with the interpretation because deviance detection was not measured using tone duration within this range and other than 100 ms. For example, I notice significant responses to 50-ms standard tones in oddball paradigm in these neurons. Would this imply the generation of pure error signal upon presentation of standard tones in oddball paradigm in these neurons?

We are sorry that our response to the reviewer’s comment 7 in the previous revised version, “under the current condition, the tone duration for inducing the pure error signal in the RPB neurons with offset responses was 25–175 ms” was misleading. If the current many-standards control paradigm with 10–255-ms tones is used, the duration of the standard tone in the oddball paradigm to induce the pure error signal in the RPB should be at least 25–175 ms because RPB responses to these tones were very weak. However, we agree that with any of these standard tones, there is no guarantee that the pure error signal is induced in the RPB. We think that the response to the 50-ms tone in the oddball paradigm in RPB neurons was the sound response, rather than the pure error signal. The 50-

ms tone was presented much more in the oddball paradigm than in the many-standards control paradigm with 10 different tone presentations. Thus, the noise in each trial was canceled out by the averaging process in each session so that the weak positive auditory response might be detected at a significant level. In this graph, the test is whether there was a significant difference from zero, not whether there was a significant difference from the response to the 50-ms tone in the many-standards control paradigm. The response to the 50-ms tone did not significantly differ between the many-standards control and oddball paradigms ($P = 0.17$, $n = 25$, t -test). The pure error signal does not mean that the sound response is not included at all. As we described on page 3, we do not intend to insist that the responses to the 10-, 200-, and 225-ms tones in the many-standards control paradigm were the deviance detection. We have added our interpretation as follows (lines 367–372): “This property of RPB neurons might also be related to the finding that they significantly responded to 10-, 200-, and 225-ms tones whose durations were relatively different from the averaged tone duration (113.5 ms) in the many-standards control paradigm. To understand the generic role of the RPB in dMMN, it is necessary to examine the different combinations of deviant and standard-tone durations including short deviant and long standard tones^{7,11,48}”.

I find it intriguing that 10-ms tones in many-standard paradigm evoked significant responses in these neurons. Did the presentation of many-standard paradigm always start with 10-ms tones? Also, I wonder if the tone presentation in many-standard paradigm was randomized.

In the many-standards control paradigm, the tones with different durations were presented in a different pseudo-random order in each session. Thus, the first tone was not the same across the sessions. We have added this information to the Methods section as follows (line 773): “These tones were pseudo-randomly presented for each session”.

Response to main comment 8)

The authors properly addressed my comment. To be consistent, I suggest to label ‘2P’ for the traces in Fig 1g, 1h and Fig S3b-d.

We have added the label “2P” to the traces in Fig. 1g, 1h, and Supplementary Fig. 3b–d, accordingly.

Response to main comment 9)

Neurons in RPB were classified into 4 types according to their offset responses and the matching between their BF and FOdd. Comparing the tone responses to the longest duration tones, i.e. of 225

ms, in many-standard paradigm for the 4 types of neurons, the author reported that the tone responses were the largest when the neurons were offset+ and the presented tone frequencies, the same as F_{Odd}, were their BF (Response Fig 9). They concluded that largest responses would still be elicited when tones of BF were presented with shorter duration compared to that in pure-tone paradigm, possibly including some deviance detection component as 225 ms was very different from the average tone duration in many-standard paradigm (please see response to my main comment 1)). In Response Fig 9, I wonder if there was significant difference between responses to 225-ms tones in F_{Odd}Offset+ neurons and those in non-F_{Odd}Offset+.

There was no significant difference between them even when the Bonferroni correction was not applied ($P = 0.068$). We do not use “significantly” in this description in the main text. However, when we compared F_{Odd} onset- offset+ neurons ($n = 7$) and non-F_{Odd} onset- offset+ neurons ($n = 11$), the difference was significant ($P = 0.011$, Wilcoxon rank sum test). These slightly different results might be because of the imbalance of the numbers of onset+ offset+ and onset- offset+ neurons between F_{Odd} and non-F_{Odd} neurons (14 and 7 for F_{Odd} neurons, respectively, and 1 and 11 for non-F_{Odd} neurons, respectively).

I also wonder whether the responses to 225-ms tones in RPB neurons whose BF=F_{Odd}, both offset+ and offset-, were significantly larger compared to those in RPB neuron whose BF≠F_{Odd}, both offset+ and offset-.

There was no significant difference between them (0.088 ± 0.015 vs. 0.057 ± 0.012 , $n = 68$, $P = 0.099$, unpaired t -test). We have added this result to the legend of Supplementary Fig. 5g.

The authors suggest that a deviance detection component may be included in the responses to 225-ms tones in many-standard paradigm. It remains unclear to me how 225-ms, along with 10-ms and 200-ms, tone duration was classified as ‘deviant’ durations in many-standard paradigm.

I think that similar analyses as in Response Fig 9 for tone durations of 25-175 ms in many-standard paradigm, which may not induce deviance detection, could help clarify whether neurons show selective responses to BF when tones of shorter duration were used.

We have calculated the responses of RPB neurons to 25–175-ms tones in the many-standards control paradigm (Response Fig. 2). Except for the response to the 125-ms tone, the responses did not differ among the four types of neurons. The mean response to the 125-ms was small, ranging from -0.02 to 0.03 . Thus, it was difficult for us to conclude whether the 125-ms tone-specific significant differences were particularly meaningful. Overall, there was no common difference in the responses

between F_{odd} and non- F_{odd} neurons and the responses were small. Thus, the repeated stimulation in the many-standards control paradigm probably suppressed the responses to these tone durations. It is necessary to present short tones for a sufficiently long SOA to reveal the BF to short tones in individual neurons. This is noted in the main text as follows (lines 261 and 262):

“Although the BF determined in the pure-tone paradigm was not necessarily the best frequency for the shorter tone stimulation”.

As we described in our responses on pages 3, 8, and 9, we do not intend to insist that the responses to the 10-, 200-, and 225-ms tones in the many-standards control paradigm were the deviance detection. We have added our interpretation as follows (lines 367–372):

“This property of RPB neurons might also be related to the finding that they significantly responded to 10-, 200-, and 225-ms tones whose durations were relatively different from the averaged tone duration (113.5 ms) in the many-standards control paradigm. To understand the generic role of the RPB in dMMN, it is necessary to examine the different combinations of deviant and standard-tone durations including short deviant and long standard tones^{7,11,48}”.

Response Figure 2. Averaged response amplitudes (100–300 ms after the tone end) of four types of RPB neurons to the 25-, 50-, 75-, 100-, 125-, 150-, and 175-ms tones in the many-standards control paradigm. The neurons with and without offset responses were further divided into those whose BF was F_{odd} (F_{odd} Offset+ and F_{odd} Offset-) and those whose BF was not F_{odd} (non- F_{odd} Offset+ and non- F_{odd} Offset-). * $P < 0.05$, ** $P < 0.01$, unpaired t -test with Bonferroni correction. $n = 21$ for F_{odd} Offset+, 47 for F_{odd} Offset-, 12 for non- F_{odd} Offset+, and 56 for non- F_{odd} Offset- from two animals.

Response to main comment 10)

It is clear to me now why no black lines were indicated in Fig 7b and significant difference between responses to oddball tone and those to the same duration tones in many-standard paradigm indicated in Fig 7d.

To address whether the reduced deviance detection in core was caused by reduced responses to 100-ms tones in oddball paradigm or enhanced responses to 100-ms tones in many-standard paradigm, the authors compared the responses to 100-ms tones in many-standard paradigm with RPB muscimol, CPB muscimol, and RPB saline, and found no significant difference among them. This result indicated that muscimol injection in RPB and CPB would not significantly affect the response to the 100-ms many-standard tones compared to control manipulation of RPB saline, suggesting that the reduced deviance detection with RPB muscimol may be resulted from decreased responses to the 100-ms oddball tones. I think that a more direct way to address this question could be to compare the responses to 100-ms oddball tones and 100-ms many-standard tones before and after pharmacological manipulations, because this way potential compound effects on the tone responses due to injection of muscimol or saline could be excluded.

It is clear to me now effective optogenetic inactivation of RPB is still under development and not yet ready to be applied in the current study.

We agree that it would be ideal to measure the responses before and after drug administration within the same day. However, the drug administration procedure includes fixing the animal's head in the surgical apparatus, opening the glass window, injecting muscimol with a glass pipette, putting the glass window, and returning the animal to the cage. We waited 2 h after the injection before starting the imaging experiment. If the imaging experiment was conducted before the injection, it might have affected the animal's state after the injection. The animal's state might differ to some extent before and after the muscimol injection. Thus, we think it is appropriate to use saline injection, which takes into account the effect of the injection procedure, as a control in the current experimental condition.

Response to main comment 11)

The authors explained that subareas in the core showed artifactual signal upon optogenetic stimulation of RPB (probably as shown in Fig S8h(3)) and therefore were removed for calculating the core responses upon photostimulating RPB (Fig S8i). I think that justification of this procedure should be provided in the manuscript. For example, I expect that artifactual signal upon photostimulating RPB may display different temporal profiles compared to tone-evoked responses.

We have added the procedure to exclude the core region that responded to only photostimulation as follows (lines 344–347):

“In this calculation, we removed the core region in which the fluorescence intensity was increased by only photostimulation to remove the contribution of some red laser-induced light artifacts and RPB activation-dependent, but tone response-independent, activity (Supplementary Fig. 8h,j,k)”.

We have also added the fluorescence trace in these pixels in Supplementary Fig. 8j, k. As the reviewer predicted, the temporal profile of the response to the tone followed by photostimulation was markedly different from that of the response to the tone only (Supplementary Fig. 8j). The temporal profile of the difference between these responses was similar to that of the response to photostimulation only (Supplementary Fig. 8k).

Supplementary Figure 8j, k.

j, Calcium responses to the tone followed by photostimulation (red) and to only the tone (blue) in the core region that showed significant responses to the tone followed by photostimulation and to only photostimulation. The black horizontal line indicates the period during which the amplitude significantly differed between the red and blue traces ($P < 0.05$, Wilcoxon rank sum test, FDR-adjusted). $n = 46$ from three animals; 14 for 1 kHz, 11 for 2 kHz, 11 for 4 kHz, 9 for 8 kHz, and 1 for 16 kHz. The reason why this session number (46) was smaller than that in Fig. 7f (68) was because no core region showed significant responses to both the tone followed by the photostimulation and to only the photostimulation in 22 sessions. **k**, Black, calcium responses to only photostimulation in the core region that showed significant responses to the tone followed by the photostimulation and to only the photostimulation. Green, the red response minus the blue response in **j**. There was no timepoint at which the black and green traces significantly differed ($P > 0.05$, Wilcoxon rank sum test, FDR-adjusted). For the black response, $n = 54$ from three animals. The reason why the session number differed between the black and green responses was because the number of sessions with only photostimulation and the number of sessions with the tone followed by the photostimulation were different.

It remains yet unclear how RPB activation contributes to deviance detection in core. I think it'd be

helpful if the authors could discuss about potential mechanisms according to their preliminary results described in the response letter.

Feedback projection to layer 1 can enhance the dendritic excitability of pyramidal neurons (Larkum, *Trend in Neurosci.* 36, 141–151, 2013). Lesion of the extrastriate cortex decreases V1 neural responses to texture elements (Lamme et al., *Curr Opin Neurobiol* 8, 529–535, 1998). Thus, the higher-order cortical area can modulate the activity of the lower-order cortical area. We have discussed its possible mechanism as follows (lines 415–424):

“Based on the optogenetic experiment, even when the core was not sufficiently depolarized to exhibit action potentials upon RPB photostimulation alone, if the core had already been depolarized by the sound stimulation, it could be depolarized to exceed the action potential threshold by the RPB photostimulation. Feedback projection into L1 can increase the dendritic excitability of excitatory neurons⁵¹. Thus, the top-down feedback from RPB may further depolarize the core and belt neurons that are already depolarized during the tone presentation. In turn, some of these neurons in the local region might trigger action potentials ~200–400 ms after the tone onset, and then amplify RPB activity with offset response in the feedforward direction to maintain deviance detection signaling within the auditory cortex”.

We have also cited Lamme et al. (1998) as reference #5 in line 48.

Response to minor comment 1)

The authors properly addressed my comment.

Response to minor comment 2)

The authors properly addressed my comment.

Response to minor comment 3)

I find the bottom panels of Fig 4b-d still not clear. I think it'd be helpful to indicate their sample sizes and SEM. Fig 4e is clear to me now.

We apologize for the unclear explanation. All traces are the mean (trial-average), and shading indicates SEM. In b, the number of trials with the 100-ms tone in the oddball and many-standards control paradigms was 30 and 14, respectively. In c and d, the number of trials with the 100-ms tone in the oddball and many-standards control paradigms were 29 and 10, respectively. In d, the number of tone-responsive neurons was 64. Their responses were averaged in each trial, and then the neuron-averaged responses were averaged over trials. In b–d, F_{odd} was 4 kHz. We have added this information to the corresponding legend.

Response to minor comment 4)

Adding black lines indicating significant deviance detection in the left-most panels of Fig S4a, S4b and the p-values for testing deviance detection in averaged responses in these panels addressed my comment.

I still find line 908-911 in the revised manuscript should be moved to Results or Discussion.

We have moved this sentence to lines 242–245. We have also added a simple definition of these neurons before this sentence. We have rechecked the cell counts and found some errors. We have corrected them.

Response to minor comment 5)

The authors properly addressed my comment.

Response to minor comment 6)

The authors properly addressed my comments.

Other comments

a) Sometimes data from just one animal were included (e.g., Response Fig7c). I wonder whether variation between animal could affect conclusion.

We first used three animals for each of one-photon calcium imaging, two-photon calcium imaging, and LFP recording to confirm the reliability of measurement of the deviance detection in the core. The results were basically consistent across the three measurements. Thus, we think that our three types of measurements were reliable. Although the LFP recording in the ML/CL, RPB, and CPB was conducted in only one animal, the result of the deviance detection was consistent with the results obtained from one-photon and two-photon calcium imaging, in each of which two animals were used. Although one-photon calcium imaging of RPB axons in the core was conducted in one animal, the deviance detection was clearly detected across sessions and the result was consistent with the other one-photon imaging results. We used one animal for one-photon calcium imaging of the dlPFC. The finding that the deviance detection was not significant in one animal does not allow us to conclude that the dlPFC did not cause the deviance detection. However, marmosets are valuable animals, and we believe that presentation of this result as supplementary information is meaningful to many readers. Our main conclusions are from one-photon calcium imaging of the core and RPB in the dMMN paradigm, two-photon calcium imaging of the core and RPB in the dMMN paradigm, one-photon calcium imaging of the core dMMN response after muscimol injection into the RPB, and

one-photon calcium imaging of the core tone response during optogenetic activation of the RPB. Each of these experiments was repeated in two or more animals, and the results did not qualitatively differ between the animals. Therefore, although we do not exclude some effects of animal-to-animal variability on the results, we do not think that it greatly affected the current major conclusions.

While we admit that the number of animals is small for each type of experiment, we also think that the number of non-human primates used in invasive experiments must be as low as possible. The numbers of animals and sessions are described for each experimental result, allowing readers to interpret the current results based on these counts.

b) The blue and oranges traces in Fig 3f,3g, Fig 6d, Fig7a-d, Fig S6e, Fig S7d, S7e were not described in figure legends.

We have added an explanation to the legends of these figures.

c) Line 122: in ref 32 no many-standard paradigm was used.

We apologize for this mistake. We have changed it to reference #13 (Koshiyama et al., Schizophr. Bull. 46, 937–946, 2020).

REVIEWERS' COMMENTS

Reviewer #2 (Remarks to the Author):

I find the authors' replies to my comments satisfactory. I have only a few minor comments.

About main comment 1)

Some more comment a):

I think that it is more comprehensive now that the authors provided Figure S3j, as a comparison with Figure 2h. I however find the added lines 229-231 a bit confusing. I think that the authors implied that the LFP responses to 50-ms tones in oddball paradigm after the deviant tone in RPB were already adapted (Figure 3h, line 225-229) instead of not showing adaptation.

About main comment 2)

I think that it is more comprehensive now that the authors provided the illustration and results in Response Figure 1 as Supplementary Figure 2h, 2i, 3k.

Regarding Response Figure 1a, I think it may be more clear to indicate 'deviant tone duration 100 ms in oddball paradigm' instead of '100 ms in oddball' and 'standard tone duration 50 ms in oddball paradigm' instead of '50 ms in oddball'.

Regarding Response Figure 1b,1c, I think that it'd be more clear if the authors can discuss about no apparent adaptation component in core and RPB, e.g., the former was likely due to no adaptation in responses to 50-ms tones in oddball and many-standard paradigm, while the latter was likely due to similarly strong adaptation in responses to 50-ms tone in oddball and many-standard paradigm.

Other comment:

line 193-196: It is not clear to me how the percentages of offset+ and onset+ neurons were derived from Supplementary Table 2.

We thank the reviewer for his/her careful consideration of our manuscript and for making helpful comments. Our detailed responses (in black) to the reviewer's comments (in blue) are provided below:

Reviewer #2 (Remarks to the Author):

I find the authors' replies to my comments satisfactory. I have only a few minor comments.

We thank Reviewer #2 for the positive comment on the revised version of our manuscript.

About main comment 1)

Some more comment a):

I think that it is more comprehensive now that the authors provided Figure S3j, as a comparison with Figure 2h. I however find the added lines 229-231 a bit confusing. I think that the authors implied that the LFP responses to 50-ms tones in oddball paradigm after the deviant tone in RPB were already adapted (Figure 3h, line 225-229) instead of not showing adaptation.

We apologize for the unclear explanation. We have changed this sentence as follows (lines 233–235):

“and the attenuated amplitude of the LFP response to the 50-ms stimulus did not recover immediately after the deviant tone nor did it then adapt again until the next deviant tone in the oddball paradigm”.

About main comment 2)

I think that it is more comprehensive now that the authors provided the illustration and results in Response Figure 1 as Supplementary Figure 2h, 2i, 3k.

Regarding Response Figure 1a, I think it may be more clear to indicate ‘deviant tone duration 100 ms in oddball paradigm’ instead of ‘100 ms in oddball’ and ‘standard tone duration 50 ms in oddball paradigm’ instead of ‘50 ms in oddball’.

We have changed them in Supplementary Figure 2h accordingly. In the same figure panel, we have also changed “100 ms in many-standards” and “50 ms in many-standards” to “Tone duration 100 ms in many-standards control paradigm” and “Tone duration 50 ms in many-standards control paradigm”, respectively.

Regarding Response Figure 1b, 1c, I think that it'd be more clear if the authors can discuss about no apparent adaptation component in core and RPB, e.g., the former was likely due to no adaptation in responses to 50-ms tones in oddball and many-standard paradigm, while the latter was likely due to similarly strong adaptation in responses to 50-ms tone in oddball and many-standard paradigm.

We have added the following discussion in lines 239–242.

“No apparent adaptation component in the core was likely due to no apparent adaptation in responses to the 50-ms tone in the oddball and many-standard control paradigms, while that in the RPB was likely due to similarly strong adaptation in responses to the 50-ms tone in both paradigms”.

Other comment:

line 193-196: It is not clear to me how the percentages of offset+ and onset+ neurons were derived from Supplementary Table 2.

We apologize this mistake. We have changed these sentences as follows (lines 197–200):

“We found that 14–23% of tone-responsive neurons in the core, lateral belt, and parabelt showed such offset responses (offset+; Supplementary Table 2). Among offset+ neurons, 50–59% also showed responses during the tone presentation (onset+ offset+; Supplementary Table 2)”.